# INFERRING THE INVISIBLE: NEURO-SYMBOLIC RULE DISCOVERY FOR MISSING VALUE IMPUTATION

**Wendi Ren**[1*], **Ke Wan**[2*], **Junyu Leng**[3*], **Shuang Li**[1†]
[1] The Chinese University of Hong Kong, Shenzhen
[2] University of Virginia    [3] Texas A&M University
wendiren@link.cuhk.edu.cn, tbn5pj@virginia.edu
levileng@tamu.edu, lishuang@cuhk.edu.cn

## ABSTRACT

One of the central challenges in artificial intelligence is reasoning under partial observability, where key values are missing but essential for understanding and modeling the system. This paper presents a neuro-symbolic framework for latent rule discovery and missing value imputation. In contrast to traditional latent variable models, our approach treats missing grounded values as latent predicates to be inferred through logical reasoning. By interleaving neural representation learning with symbolic rule induction, the model iteratively discovers—both conjunctive and disjunctive rules—that explain observed patterns and recover missing entries. Our framework seamlessly handles heterogeneous data, reasoning over both discrete and continuous features by learning soft predicates from continuous values. Crucially, the inferred values not only fill in gaps in the data but also serve as supporting evidence for further rule induction and inference—creating a feedback loop in which imputation and rule mining reinforce one another. Using a staged block-coordinate gradient descent, the system learns these rules end-to-end by iteratively optimizing over parameter blocks in an alternating fashion. Experiments on both synthetic and real-world datasets demonstrate that our method effectively imputes missing values while uncovering meaningful, human-interpretable rules that govern system dynamics.

## 1 INTRODUCTION

Neural-symbolic reasoning integrates neural representation learning with symbolic structure, combining the noise tolerance of embeddings with the interpretability of symbolic reasoning (Hitzler & Sarker, 2022; Yang et al., 2024). This hybrid paradigm enables AI systems to detect complex patterns in unstructured data while reasoning about them in a structured and explainable manner.

Traditional rule induction methods extract explicit patterns from observed data but often fail when *some observations are missing or incomplete* (Campero et al., 2018; Claire Glanois, 2022). These approaches can effectively learn surface-level rules, yet their ability to fully explain the underlying system is limited when essential data points are absent. For example, in healthcare diagnostics, critical measurements may be missing, making accurate imputation necessary for reliable reasoning. Probabilistic models such as Markov Logic Networks (MLNs) (Richardson & Domingos, 2006) handle missing data by treating unobserved facts as latent predicates. However, they typically rely on a *fixed* rule base and *expensive joint inference*, limiting scalability and adaptability in large or heterogeneous datasets (Oltramari et al., 2020). In contrast, we propose a neuro-symbolic system that *co-learns rules and imputations* in a single differentiable loop, enabling fast forward-chaining inference and end-to-end learning.

We view each missing table entry as an unobserved grounded fact. Starting from the observed facts, our differentiable forward-chaining engine applies the current rule set to infer soft truth values for missing predicates. We then update rule embeddings, together with soft predicates for continuous attributes, by minimizing a masked reconstruction objective on the observed entries, treating the partially observed table as weak supervision. Better rules yield better imputations, and improved

---

*Equal contribution
†Corresponding author

imputations provide stronger evidence for refining rules in subsequent passes, producing an iterative coupling between completion and induction.

Many targets require compositional explanations, such as multi-hop chains and disjunctive rule sets, and the data are often heterogeneous with both discrete and continuous attributes. Our framework supports expressive rules over mixed data types by learning soft predicates for continuous features (sigmoid thresholds and slopes) and combining them with discrete predicates through smooth logical operators. In particular, we use soft-min as a differentiable surrogate for AND and soft-max as a surrogate for OR, enabling uniform forward chaining without pre-discretization.

Learning in this expressive space must also remain tractable. To achieve this, we use a staged, block-coordinate optimization scheme, where we update one rule or clause at a time while holding others fixed. This approach reduces interference between candidate explanations and helps manage search complexity as the rule set grows. For disjunctive heads, a common challenge is that individual clauses may be underrepresented due to missing data. To address this, we first generate a diverse set of clauses through sequential covering, then fine-tune them together using a soft-OR aggregator (LogSumExp), which captures interactions between clauses. This procedure scales efficiently to complex disjunctive rule sets, even under high missingness.

**Contributions.** (*i*) We introduce a differentiable neuro-symbolic framework that *co-learns rules and imputations* through an iterative feedback loop, instead of treating imputation as a separate pre-processing step. (*ii*) We develop a staged training strategy based on asynchronous block-coordinate updates, sequential covering for discovering diverse clauses, and joint fine-tuning with a soft-OR (LogSumExp) aggregator for disjunctive heads. (*iii*) We design a unified differentiable forward-chaining engine for heterogeneous tables, combining discrete predicates with learned soft predicates for continuous attributes (sigmoid thresholds and slopes) via smooth logical operators (soft-min for AND, soft-max for OR). (*iv*) We empirically validate the approach on synthetic chain and disjunction benchmarks and on real-world datasets (Birds, Heart, SPECT), demonstrating accurate imputation, strong downstream prediction, and recovery of human-interpretable rules.[1]

## 2 RELATED WORK

Our work combines neuro-symbolic Inductive Logic Programming (ILP) with missing value imputation.

**Neural Embedding-based ILP.** Embedding-based models such as TransE (Bordes et al., 2013), TransH (Wang et al., 2014), and TransR (Lin et al., 2015) are widely used for Knowledge Base (KB) completion, learning vector representations for entities and relations. Complex (Trouillon et al., 2016) extends these with complex-valued embeddings to handle asymmetric relations, and multi-hop methods like Guu et al. (2015) use path-based embeddings for reasoning over knowledge graphs. While successful, these methods are limited in their ability to discover complex logical rules. Recent advances in ILP integrate symbolic logic with neural networks, including Rocktäschel & Riedel (2017)'s Neural Theorem Proving (NTP), a differentiable backward-chaining method, and Campero et al. (2018)'s forward-chaining rule induction network. However, both rely on hand-designed templates, limiting flexibility. Claire Glanois (2022) improve flexibility with a hierarchical structure for more adaptive rule induction, but these models are still designed primarily for fully-observed data and struggle with missing values.

**Interpretable Rule Learning.** Learning interpretable logical rules has long been a goal in AI. Dash et al. (2018) propose BRCG, an integer programming approach that efficiently searches the space of candidate clauses, balancing accuracy with simplicity. Wang et al. (2021) introduce RRL, which uses Gradient Grafting to learn non-fuzzy rule lists in a deep learning framework, ensuring scalability. Qiao et al. (2021) present DR-NET, which learns decision rules in Disjunctive Normal Form (DNF) by optimizing both rule generation and weight learning. More recently, Barbiero et al. (2022) propose LEN, a differentiable neuro-symbolic method that extracts concise First-Order Logic (FOL) explanations from neural networks using an entropy-based criterion. In contrast to these methods, which focus primarily on classification tasks with complete data and binary features, our framework integrates rule learning directly with missing value imputation, enabling both tasks to reinforce each other in a feedback loop.

**Rule-Based Missing Value Imputation.** Traditional missing data imputation methods, ranging from statistical techniques such as MICE (Multivariate Imputation by Chained Equations) (Van Buuren & Groothuis-Oudshoorn, 2011), MissForest (Random Forest-based) (Stekhoven & Bühlmann,

---

[1]Our code is available at `https://github.com/conniemessi/infer_missing`.

2012), and SOFT-IMPUTE (Mazumder et al., 2010), to deep learning approaches like GAIN (GAN-based) (Yoon et al., 2018), diffusion-based models (Ouyang et al., 2023), mDAE (DAE-based) (Dupuy et al., 2024), and VAE-based methods (Veldkamp et al., 2025), generally rely on statistical patterns to fill in missing values. However, these models do not explicitly use logical rules to govern inter-variable relationships (see Appendix A for a detailed overview).

In recent years, there has been increasing interest in combining rule-based reasoning with missing value imputation. Chen et al. (2023) apply interpretable machine learning to imputation, but their methods do not explicitly use rule-based reasoning. MINTY (Stempfle & Johansson, 2024) employs a rule-based model, yet it lacks neuro-symbolic reasoning to learn complex relationships between observed and missing values. In contrast, our framework integrates neural representation learning with symbolic reasoning, enabling flexible discovery of interdependencies between observed and missing data. Non-neural approaches like (Wang et al., 2017) address missing values but lack the adaptability and representation learning of neural networks. Our method stands out by forming a feedback loop where rule discovery and imputation mutually reinforce each other.

## 3 BACKGROUND

**Predicates and Latent Predicates.** A predicate is a Boolean-valued relation that describes a property of an entity or a relationship between entities. Grounding a predicate with concrete entities yields an atomic fact, which may be observed (true/false) or missing. For example, $Has\_Fever(p)$ indicates whether patient $p$ has a fever, and $Use\_Drug(p)$ indicates whether a drug is used for patient $p$. We refer to a missing grounding of a predicate for a particular entity (or entity tuple) as a *latent predicate instance*; importantly, the same predicate may be observed for other entities. In our differentiable setting, such instances can be assigned soft truth values to represent uncertainty.

**Rules and Forward Chaining.** We model logical structure with rules in an OR-of-ANDs form:
$$Q_k \leftarrow (P_1 \wedge P_2) \vee (P_3 \wedge P_4) \vee \cdots,$$
so that $Q_k$ is inferred when any clause is satisfied. Such rule sets support one-step inference when clause bodies involve only observed predicates, and multi-step inference when intermediate predicates are inferred first, e.g.,
$$Q_1 \leftarrow P_1 \wedge P_2, \quad Q_2 \leftarrow P_3 \wedge P_4, \quad Q_k \leftarrow (Q_1 \wedge P_5) \vee (Q_2 \wedge P_6).$$

Given an evidence set of observed facts $\mathcal{E}$, we perform *forward chaining*: whenever a clause is (approximately) satisfied by facts in $\mathcal{E}$, it produces its head predicate and adds the newly inferred fact back into $\mathcal{E}$. Reusing inferred facts as evidence enables multi-hop reasoning through cascades of rule applications.

## 4 MODEL: NEURO-SYMBOLIC FORWARD CHAINING NETWORK (NS-FCN)

Consider problems where some information or features are incomplete. Given the observed predicate (denoted by $\mathbf{X}$) and the missing predicate (denoted by $\mathbf{U}$), our goal is to learn a set of logical rules and use them to impute the missing ones (as illustrated in Figure 1).

| $X_0$ | $X_1$ | $X_2$ | $X_3$ | $X_4$ | $X_5$ | $X_6$ | $X_7$ |
|---|---|---|---|---|---|---|---|
| 1 | 0 | 0 | 0 | nan | nan | 1 | 0 |
| 0 | 0 | 0 | 0 | 0 | 0 | 1 | 1 |
| 0 | 1 | 0 | nan | nan | 0 | 0 | 0 |
| 0 | 1 | 1 | nan | 0 | nan | 1 | 0 |
| 1 | 0 | 1 | nan | nan | nan | 0 | 0 |

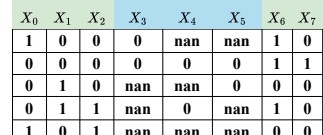
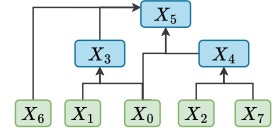
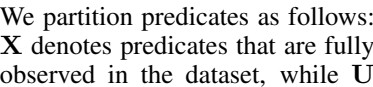

Figure 1: Example of missing variables imputation with rule discovery. Feature with *nan* corresponds to the predicates with missing information (belong to $\mathbf{U}$), which can be inferred by the logic rules and other observed features.

We partition predicates as follows: $\mathbf{X}$ denotes predicates that are fully observed in the dataset, while $\mathbf{U}$ denotes predicates that have missing values for at least one entity (i.e., partially observed predicates). Thus, a predicate belongs to $\mathbf{U}$ as long as some of its groundings are missing, even though other entities may have observed values. The predicate space considered by the model is the union $\mathbf{X} \cup \mathbf{U}$.

To summarize, our model learns logical rules to infer missing instances of latent predicates $\mathbf{U}$ by discovering hidden structures within data. This rule induction process identifies logical relationships among observed or partially observed predicates. By explicitly learning these structures, our approach enhances both inference capability and interpretability, offering clear insights into complex, otherwise hidden dependencies. The key method pipeline is summarized in Figure 2, with details presented in the following sections.

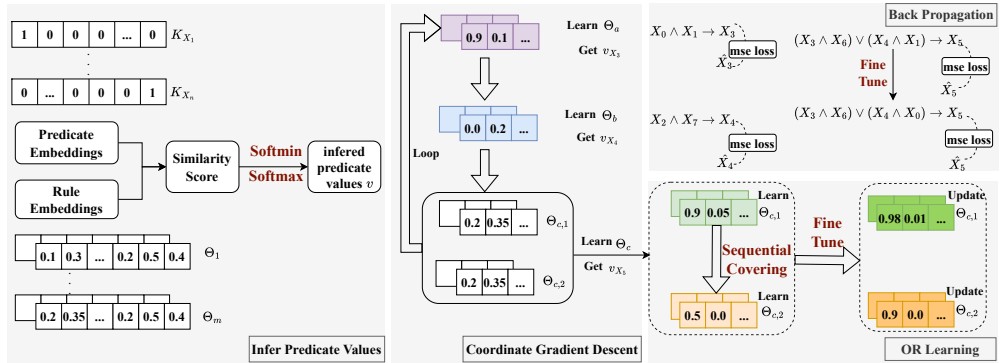

Figure 2: Model framework. Rule embeddings $\Theta$ are optimized using coordinate gradient descent. In each learning step, predicate values are inferred via the Softmin-Softmax operation (Eqs. 2, 3, and 4). For disjunctive (OR) rule learning, sequential hard covering is applied, followed by fine-tuning of the learned rule embeddings (Section 5.2). Errors are back-propagated using MSE loss between inferred predicate values and the small portion of observed latent predicate samples, constituting a weak-supervision setting.

## 4.1 MODEL PREPARATION: PRETRAINED PREDICATE EMBEDDINGS

We begin by defining two sets of predicates: $\mathbf{X} = \{X_1, \ldots, X_n\}$ represents the set of *fully observed* predicate variables, and $\mathbf{U} = \{U_1, \ldots, U_m\}$ denotes the set of predicate variables with *missing information* that the model aims to discover and define. Our framework is designed to handle both binary (categorical) and continuous features within a unified logical structure. Binary features are treated as standard logical predicates. For continuous features, we introduce a mechanism to derive a "soft" truth value, effectively creating learnable predicates from them. This allows the model to reason over heterogeneous data types, as detailed in Section 4.2.

We initialize a fixed, unique embedding for each predicate, whether observable or missing. For example, these embeddings can be instantiated as one-hot vectors within an embedding space of dimension $d$. We denote the collection of embeddings for observable predicates as $\mathbf{K}_X$ and incomplete predicates as $\mathbf{K}_U$. These *predicate embeddings* remain **frozen** throughout the rule learning phase and serve as a foundational dictionary, enabling the interpretation of the composition of learned rules by relating rule components back to specific predicates.

With the predicate representations defined, we next describe the core of our model: the representation of logical rules and the mechanism by which inferences are drawn.

## 4.2 MODEL BACKBONE: RULE REPRESENTATION AND INFERENCE

In our NS-FCN framework, logical rules are parameterized by learnable rule embeddings, which serve as the main trainable variables. We optimize these embeddings using an asynchronous block-coordinate descent scheme, updating one rule (or clause) at a time while holding the others fixed. This learning strategy is well-suited for discovering expressive rule sets, including multi-hop compositions (where one inferred predicate appears in the body of another rule) and disjunctive heads (where a predicate can be supported by multiple alternative clauses).

### 4.2.1 SPECIFICATION OF RULE EMBEDDINGS $\Theta$

Let $\mathcal{F}$ be the set of rules/clauses, and let $\Theta = \{\Theta_f\}_{f \in \mathcal{F}}$ be their embeddings. Each $\Theta_f$ encodes the content of a rule whose head predicate is $U_k \in \mathbf{U}$, and a single head predicate $U_k \in \mathbf{U}$ can be supported by multiple rules (an OR-of-ANDs structure).

*Conjunctive Rule Embedding.* Suppose $U_k$ is supported by a single conjunctive clause (e.g., $U_k \leftarrow X_a \wedge X_b$), we associate the clause with a rule embedding matrix $\Theta_f = [\theta_1, \ldots, \theta_h] \in \mathbb{R}^{d \times h}$. Here, $h$ is the number of body predicates in the conjunction (e.g., $h = 2$ for $X_a \wedge X_b$) and $d$ is the embedding dimension. Each column $\theta_j \in \mathbb{R}^d$ is trained to match the embedding of one body predicate, so that the learned $\Theta_f$ provides an interpretable representation of the clause.

*Disjunctive Rule Embeddings.* Suppose $U_k$ is defined by a disjunction of $R_k$ distinct conjunctive clauses (e.g., $U_k = \bigvee_{r=1}^{R_k} (\text{clause}_r)$), it will be associated with a set of $R_k$ distinct rule embed-

dings, denoted $\{\Theta_{k,1}, \ldots, \Theta_{k,R_k}\}$. Each individual rule embedding $\Theta_{k,r}$ is itself an $d \times h_r$ matrix, representing the $r$-th conjunctive clause, where $h_r$ is the arity of that specific clause.

All rule embeddings are initialized randomly prior to training and are subsequently optimized as described in Section 5. Given these rule embeddings, the model infers the values (or continuous approximations thereof) of latent predicates through a carefully defined inference mechanism.

### 4.2.2 INFERRING PREDICATE VALUES

We infer latent predicate values from the current predicate state and the learned rule embeddings $\Theta$, using a differentiable forward-chaining procedure.

**Predicate Matching.** Each column $\theta_j \in \mathbb{R}^d$ of a clause embedding $\Theta_f = [\theta_1, \ldots, \theta_h]$ is intended to select one body predicate. Let $\boldsymbol{K} = \boldsymbol{K}_X \cup \boldsymbol{K}_U$ be the set of frozen predicate embeddings. We match by cosine similarity and recover the predicate index:

$$K_j^* = \arg\max_{K \in \boldsymbol{K}} \cos(K, \theta_j), \qquad i_j = I(K_j^*) \in \{1, \ldots, n+m\},$$

where $I(\cdot)$ maps an embedding back to its predicate index.

**Forward-Chaining Update.** Let $\boldsymbol{v}^t \in [0,1]^{n+m}$ denote the predicate state after $t$ forward-chaining steps, containing observed values and current (soft) estimates for missing ones. We initialize $\boldsymbol{v}^0$ from the dataset (e.g., set unobserved entries to 0.5). Let $\boldsymbol{M} \in \{0,1\}^{n+m}$ indicate which entries are observed. Denoting one-step inference by $g(\boldsymbol{v}^t; \Theta)$, we update only unobserved entries:

$$\boldsymbol{v}^{t+1} = \boldsymbol{M} \circ \boldsymbol{v}^t + (1 - \boldsymbol{M}) \circ g(\boldsymbol{v}^t; \Theta), \qquad t = 0, 1, \ldots, T \tag{1}$$

where $\circ$ is elementwise multiplication. In practice, we set $T = 2 \sim 5$ steps rather than iterating to a fixed point to avoid gradient saturation. We next define $g(\boldsymbol{v}^t; \Theta)$.

**Clause Evaluation (Soft-AND).** Given the matched indices $\{i_j\}_{j=1}^h$, the $j$-th body literal uses the current value $v_{i_j}^t$ and its matching score $s_j = \cos(K_j^*, \theta_j)$. For a conjunctive clause $f$, we compute its value via a differentiable AND implemented by a soft-min over the $2h$ scalars $\{s_1, \ldots, s_h, v_{i_1}^t, \ldots, v_{i_h}^t\}$:

$$v_f^{t+1} = \mathrm{softmin}\left(s_1, \ldots, s_h, v_{i_1}^t, \ldots, v_{i_h}^t; \tau\right), \tag{2}$$

with

$$\mathrm{softmin}(x_1, \ldots, x_{2h}; \tau) = -\frac{1}{\tau} \log\left(\frac{1}{2h} \sum_{i=1}^{2h} \exp(-x_i/\tau)\right), \tag{3}$$

where $\tau > 0$ is a temperature parameter and $\mathrm{softmin} \to \min$ as $\tau \to 0$.

**Disjunction over Clauses (Soft-OR).** If a latent head predicate $U_k$ is supported by $R_k$ conjunctive clauses, $U_k \leftarrow \bigvee_{r=1}^{R_k} f_{k,r}$, we aggregate their values $\{v_{f_{k,1}}^{t+1}, \ldots, v_{f_{k,R_k}}^{t+1}\}$ using LogSumExp:

$$g_k(\boldsymbol{v}^t; \Theta) = \frac{1}{\mu} \log \sum_{r=1}^{R_k} \exp\left(\mu\, v_{f_{k,r}}^{t+1}\right), \tag{4}$$

where $\mu > 0$ controls the softness and Eq. 4 approaches $\max_r v_{f_{k,r}}^{t+1}$ as $\mu \to \infty$. After $T$ forward-chaining steps, we take $\boldsymbol{v}^T$ as the final inferred predicate values, where observed entries remain fixed and unobserved entries are filled by the rule-based updates.

The model's ability to discover meaningful rules and infer latent predicate states accurately hinges on an effective learning procedure. We now outline the training methodology employed to optimize the rule embeddings $\Theta$.

## 5 MODEL LEARNING

Given rule embeddings $\Theta$, the inference procedure in Section 4 produces a predicate state $\boldsymbol{v}^T$, where observed entries remain fixed, and missing entries are filled by forward chaining. Model learning updates $\Theta$ so that the inferred values agree with the observed entries. Concretely, we minimize a masked reconstruction loss (weak supervision): for each latent predicate, we compare its inferred values to the subset of instances that are observed, and ignore missing ones via a binary mask. To make optimization stable and scalable in expressive rule spaces (multi-hop and disjunctive rules), we adopt a staged learning strategy: block-coordinate updates for rule embeddings, sequential covering to initialize disjunctive clauses, followed by joint fine-tuning.

## 5.1 BLOCK-COORDINATE GRADIENT DESCENT

We partition parameters by head predicate. For each latent predicate $U_k$, let $\Theta_k$ denote the set of embeddings associated with rules whose head is $U_k$ (a single clause in the conjunctive case, or multiple clauses for disjunctive heads). We optimize $\Theta$ by block-coordinate gradient descent: in each cycle, we iterate through latent predicates in a random order and update one block $\Theta_k$ while keeping all other blocks fixed.

During the update of block $\Theta_k$, we run the inference procedure to obtain the current prediction vector $v_{U_k}$ for all instances of $U_k$. Let $U_{k,\text{obs}}$ be the observed values and let $M_k \in \{0,1\}^{n+m}$ indicate which instances are observed. We minimize the masked MSE loss

$$\mathcal{L}_{U_k}(\Theta_k) = \text{mean}\Big( \big( (v_{U_k} - U_{k,\text{obs}}) \odot M_k \big)^2 \Big), \tag{5}$$

and update $\Theta_k$ using gradients from $\mathcal{L}_{U_k}$ (Adam details are deferred to Appendix C.1). This implements a Gauss–Seidel style block-coordinate method on the overall smooth objective $\mathcal{L}(\Theta) = \sum_j \mathcal{L}_{U_k}(\Theta)$. A brief convergence discussion is provided in Appendix B.

For efficiency, if a block reaches a near-perfect fit on the observed entries (e.g., imputation accuracy $> 0.99$ and marginal loss improvement $< 10^{-3}$), we freeze $\Theta_j$ in subsequent cycles.

## 5.2 LEARNING DISJUNCTIVE HEADS VIA SEQUENTIAL COVERING AND FINE-TUNING

When a head predicate $U_k$ is supported by an OR-of-ANDs rule set with $R_k$ conjunctive clauses, $U_k \leftarrow \bigvee_{r=1}^{R_k} f_{k,r}$, we learn the clause embeddings $\{\Theta_{k,1}, \ldots, \Theta_{k,R_k}\}$ in two stages.

**Stage 1: Sequential covering (clause discovery).** We learn clauses one at a time to encourage diversity. Starting with $r = 1$, we optimize $\Theta_{k,r}$ using the same masked loss in Eq. 5, but restricted to the currently *uncovered* observed-positive instances of $U_k$. After training clause $r$, we mark an observed instance as covered if the clause output exceeds a confidence threshold (e.g., $v_{f_{k,r}} > 0.99$), and remove those covered instances from the active set. We repeat this process until $R_k$ clauses are obtained.

**Stage 2: Joint fine-tuning (interaction-aware refinement).** After initialization, we jointly optimize all clause embeddings for $U_k$ by backpropagating through the soft-OR aggregation (Eq. 4). Let $v_{U_k}$ denote the aggregated prediction and let $M_k$ indicate observed instances of $U_k$. We minimize

$$\mathcal{L}_{U_k,\text{finetune}} = \text{mean}\Big( \big( (v_{U_k} - U_{k,\text{obs}}) \odot M_k \big)^2 \Big), \tag{6}$$

which refines clause embeddings while accounting for their interactions under the soft-OR operator. Optimization hyperparameters and normalization details are provided in Appendix C.1.

## 6 DISCUSSION ON CONTINUOUS FEATURES

Our inference engine operates on predicate truth values in $[0,1]$. Binary attributes are already predicates, but a continuous feature must be converted into predicate instances (soft bins/conditions) before it can participate in rule evaluation. For each continuous feature $c$ and bin index $r \in \{1, \ldots, R_c\}$, we learn a threshold $\epsilon_{c,r}$ and slope $\beta_{c,r}$ and define

$$P_{c,r}(v_c) = \sigma\big( \beta_{c,r}(v_c - \epsilon_{c,r}) \big) \in [0,1], \tag{7}$$

where $v_c$ is the raw feature value for a given entity. During forward chaining, whenever a matched body predicate corresponds to $P_{c,r}$, we use its value from Eq. 7 in clause evaluation. Let $v_{P_{c,r}}$ denote the inferred values for predicate $P_{c,r}$ (the corresponding coordinates in $v^T$), and let $P_{c,r,\text{obs}}$ be the observed soft truth values computed from observed $v_c$ via Eq. 7. With a mask $M_c$ indicating entities where $v_c$ is observed, we use the masked reconstruction loss

$$\mathcal{L}_{P_{c,r}} = \text{mean}\Big( \big( (v_{P_{c,r}} - P_{c,r,\text{obs}}) \odot M_c \big)^2 \Big),$$

which is optimized jointly with the losses for latent predicates in Section 5. Gradients therefore update both rule embeddings and the soft-discretization parameters $\{\epsilon_{c,r}, \beta_{c,r}\}$.

## 7 EXPERIMENTS

We evaluate our framework on synthetic and real-world datasets to demonstrate its effectiveness in rule discovery and missing value imputation.

### 7.1 SYNTHETIC DATA EXPERIMENTS

We use synthetic datasets to evaluate our model's ability to learn chained and disjunctive rules under partial observability (Figure 3). Each dataset is built from observable Bernoulli variables, with miss-

ing predicates defined by ground truth rules and made partially available (10%-30% observability) under an MCAR setting. The task is to learn rule embeddings that capture the ground truth logic, evaluated by *Rule Discovery Accuracy* (i.e. the proportion of runs which learn the truth rules) and *Imputation Accuracy*.

Our method also demonstrates robustness under MAR and MNAR missingness mechanisms (Appendix E). Detailed hyperparameter configurations are provided in Appendix J.4. For the two key temperature parameters of soft operators—$\tau$ in Softmin (Eq. 3) and $\mu$ in Softmax (Eq. 4)—we conduct a sensitivity analysis in Appendix H. The results indicate that our framework **does not depend on careful hyperparameter tuning**: a moderately large $\mu$ for Soft-OR and a small $\tau$ for Soft-AND consistently achieve optimal performance. Accordingly, we fix $\tau = 0.1$ and $\mu = 10$ across all experiments.

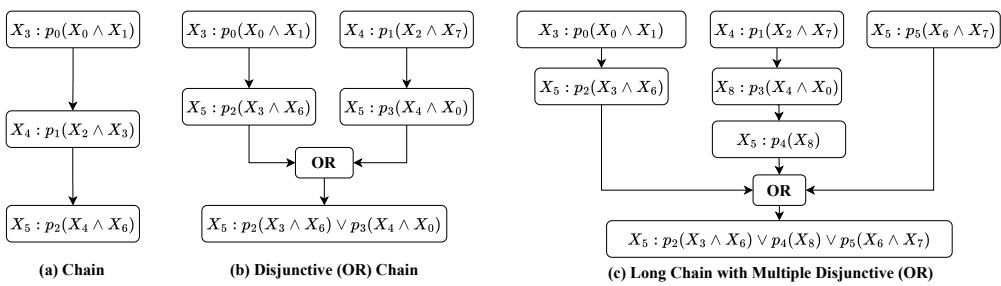

Figure 3: Example rule structures of synthetic experiments.

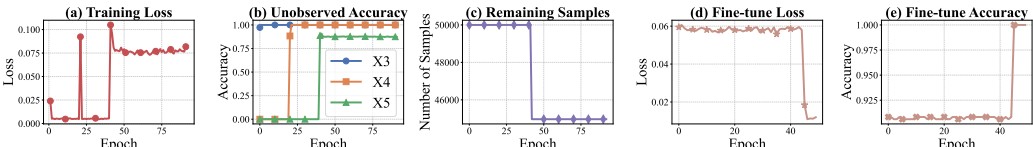

Figure 4: An example of loss and imputation accuracy during coordinate optimization (Obs. Ratio = 0.2, seed = 42). We assume the training order is $X_3$, $X_4$, $X_5$. Epochs 0–19 correspond to rule learning for $X_3$; epochs 20–39 for $X_4$; and epochs 40-end for $X_5$. Remaining samples identified how many samples are "well-explained" during the hard covering phase.

Table 1: Results for synthetic data example Figure 3(b) with an observation ratio of 0.2. Metrics are averaged over 30 random seeds on a dataset of 20,000 samples. Ground truth rules are underlined. Full results of observation ratio at 0.1 and 0.3 are in Tables 21-22 of Appendix J.1

|  | Imp. Acc. (Before FT) | Imp. Acc. (After FT) | Train Loss (Before FT) | Train Loss (After FT) | Learned Rules | Rule Acc. |
|---|---|---|---|---|---|---|
| $X_3$ | $1.000 \pm 0.000$ | / | $0.001 \pm 0.000$ | / | $\underline{X_0 \wedge X_1}$ | 1.00 |
| $X_4$ | $1.000 \pm 0.000$ | / | $0.001 \pm 0.000$ | / | $\underline{X_2 \wedge X_7}$ | 1.00 |
| $X_5$ | $0.893 \pm 0.072$ | $0.954 \pm 0.053$ | $0.087 \pm 0.045$ | $0.060 \pm 0.039$ | $(X_0 \wedge X_4) \vee (X_3 \wedge X_6)$ $\underline{(X_3 \wedge X_6)}$ $(X_0 \wedge X_4)$ | 0.53 |

**Results and Analysis.** We take setting (b) in Figure 3 as a representative example. The full results and corresponding analyses for settings (a) and (c) are provided in Appendix J.2 and Appendix J.3. Table 1 shows that our model achieves *perfect* recovery for simple conjunctive rules ($X_3$, $X_4$) and high imputation accuracy for the complex disjunctive rule ($X_5$). The top learned rule structures contain at least one correct clause, with the ground truth rule showing the highest accuracy.

Figure 4(a)-(b) illustrates stable training dynamics. For $X_5$, the model uses sequential covering (Figure 4(c)), with "well-explained" examples reducing the remaining set. The fine-tuning (FT)

Table 2: Ablation Study: Effect of Fine-tuning on $X_5$ (Disjunctive Rule) Learning

| Metric for $X_5$ | Before Fine-tuning | After Fine-tuning |
|---|---|---|
| **Recovered Rule Structure** | $(X_0 \wedge X_2) \vee (X_0 \wedge X_4)$ | $(X_3 \wedge X_6) \vee (X_0 \wedge X_4)$ |
| **Imputation Accuracy for $X_5$ (Unobserved)** | 0.8729 | 1.0 |

phase is followed, which corrects the rule structure and boosts accuracy (Figure 4(d)-(e)). The corresponding ablation study (Table 2) confirms that fine-tuning is critical for disjunctive rules, increasing unobserved imputation accuracy for $X_5$ from 0.87 to 1.00.

**Learning Efficiency.** We further investigated the impact of dataset sample size, varying it from 500 to 20,000 samples. As shown in Figure 9 of Appendix J.1, our method is data-efficient, recovering complex disjunctive rules for $X_5$ with as few as **1,000** samples. Besides, while coordinate descent requires different cycle numbers, Table 9 of Appendix F demonstrates minimal time and memory costs. Processing 50,000 samples in about 3 minutes demonstrates strong efficiency for CPU-based execution.

Table 3: Impact of rule optimization order on learning progress. Use the example (a) of Figure 3. Note: ✓ denotes successful learning for the respective predicate.

| Cycle | Metric | Run 1 | Run 2 | Run 3 |
|---|---|---|---|---|
| Cycle 1 | Optimization Order | $[X_5, X_4, X_3]$ | $[X_3, X_5, X_4]$ | $[X_3, X_4, X_5]$ |
| | Rule Accu. | $X_3$ ✓, $X_4$, $X_5$ | $X_3$ ✓, $X_4$ ✓, $X_5$ | $X_3$ ✓, $X_4$ ✓, $X_5$ ✓ |
| | Imputation Accu., Train Loss | $X_3 : 1.00, 0.005$ $X_4 : 0.87, 0.074$ $X_5 : 0.94, 0.053$ | $X_3 : 1.00, 0.005$ $X_4 : 1.00, 0.004$ $X_5 : 0.94, 0.035$ | $X_3 : 1.00, 0.005$ $X_4 : 1.00, 0.004$ $X_5 : 1.00, 0.003$ |
| Cycle 2 | Optimization Order | $[X_3, X_5, X_4]$ | $[X_5, X_3, X_4]$ | — |
| | Rule Accu. | $X_3$ ✓, $X_4$ ✓, $X_5$ | $X_3$ ✓, $X_4$ ✓, $X_5$ ✓ | — |
| | Imputation Accu., Train Loss | $X_3 : 1.00, 0.005$ $X_4 : 1.00, 0.004$ $X_5 : 0.94, 0.035$ | $X_3 : 1.00, 0.005$ $X_4 : 1.00, 0.004$ $X_5 : 1.00, 0.003$ | — |
| Cycle 3 | Optimization Order | $[X_3, X_4, X_5]$ | — | — |
| | Rule Accu. | $X_3$ ✓, $X_4$ ✓, $X_5$ ✓ | — | — |
| | Imputation Accu., Train Loss | $X_3 : 1.00, 0.005$ $X_4 : 1.00, 0.004$ $X_5 : 1.00, 0.003$ | — | — |

**Convergence Analysis of Asynchronous Coordinate Descent.** As discussed in Section 5.1, the learning order for sets of rule parameters, $\Theta_k$, can be varied. This is particularly relevant in real-world scenarios where the true dependency structure among rules is unknown. To illustrate this "order effect", we present exemplary results (corresponding to example (a) in Figure 3) in Table 3. This table highlights that different learning trajectories can affect training efficiency, while our method can learn the rule exactly. The complete results for example (a) and (b), including detailed learning curves with loss and gradient plots (Figures 8-11), are provided in Appendix J.2

We now analyze the convergence of our coordinate gradient descent design. Exact rule-set induction reduces to the minimum-set-cover problem (*NP-hard*), so like any practical rule learner, we do not claim global optimality. Instead, we frame search as asynchronous block-coordinate descent on a smooth surrogate loss: at each step, we update a single rule embedding in closed form, which guarantees the loss never increases yet keeps each move computationally cheap. To guard against poor local minima, we (i) freeze a rule only after this rule is perfectly learned, and (ii) launch diverse initializations. Across 30 runs on synthetic datasets (Appendix Tables 21-26), this strategy delivers tiny imputation performance variance, and the top-ranked learned rules consistently match ground truth rules. High Jaccard index scores ($> 0.6$) in Table 11 of Appendix G.1 further support that the learned rules are structurally stable. More theoretical discussions are provided in Appendix B.

## 7.2 REAL-WORLD DATA EXPERIMENTS

We validate our approach on three real-world datasets, comparing it with *(i) statistical models* (**MICE** (Van Buuren & Groothuis-Oudshoorn, 2011), **MissForest** (Stekhoven & Bühlmann, 2012)), *(ii) deep generative models* (**MLP**, **GAIN** (Yoon et al., 2018), **MissDiff** (Ouyang et al., 2023),

**mDAE** (Dupuy et al., 2024), **VAE** (Veldkamp et al., 2025)) and *(iii) rule-based interpretable models* (**BRCG** (Dash et al., 2018), **RRL** (Wang et al., 2021), **DR-NET** (Qiao et al., 2021), **LEN** (Barbiero et al., 2022)). For each dataset, we randomly miss some features. We then evaluated the models on their ability to impute these missing values, as well as their performance on a downstream target classification task. Preprocessing and baselines details are provided in Appendix D.2 and D.3.

For *logical reasoning*, we used the Birds dataset (Tafjord et al., 2021) with a 90% missing ratio for two key predicates. As shown in Table 4, under some random seeds, NS-FCN achieves perfect imputation accuracy (1.00) and, crucially, **perfectly recovers the ground truth logical rules**, highlighting its superior capability in deciphering underlying logical structures. Table 6 compares our approach with non-interpretable baselines. While an MLP achieves optimal performance given the simplicity of the Birds dataset, our model remains highly competitive; more importantly, it demonstrates robustness across diverse random initializations, successfully recovering the correct ground-truth rules in the majority of cases. Table 20 further shows that a few hundred samples are sufficient for the model to converge to the correct logical truth.

Table 4: Comparison of imputation accuracy and learned rules on the Birds dataset.

| Method | Imp Acc. | Learned Rules Example |
|---|---|---|
| LEN | 0.57 | $abnormal\_bird \leftarrow (ostrich \land \neg wounded) \lor (bird \land wounded)$ |
| | 0.55 | $can\_fly \leftarrow (bird \land \neg ostrich) \lor (\neg ostrich \land \neg wounded)$ |
| RRL | 0.53 | $abnormal\_bird \leftarrow (bird \land \neg wounded) \lor (bird \land ostrich)$ |
| | 0.51 | $can\_fly \leftarrow (\neg ostrich \land \neg wounded) \lor (bird \land \neg ostrich)$ |
| BRCG | 0.50 | $abnormal\_bird \leftarrow bird \land ostrich$ |
| | 0.47 | $can\_fly \leftarrow bird \land \neg abnormal\_bird$ |
| DR-NET | 0.56 | $abnormal\_bird \leftarrow (bird \land \neg ostrich \land wounded) \lor (bird \land ostrich \land \neg wounded)$ |
| | 0.53 | $can\_fly \leftarrow (bird \land \neg ostrich \land \neg abnormal\_bird) \lor (bird \land \neg ostrich \land \neg wounded)$ |
| **NS-FCN** | **1.00** | $abnormal\_bird \leftarrow ostrich \lor (bird \land wounded)$ |
| | **1.00** | $can\_fly \leftarrow bird \land \neg abnormal\_bird$ |

Table 5: Comparison of imputation accuracy and learned rules on the Heart Disease dataset.

| Method | Imp Acc. | Learned Rules Example |
|---|---|---|
| LEN | 0.65 | $trestbps\_high \leftarrow (\neg st\_mild \land cp\_atypical\_angina) \lor (chol\_low \land cp\_asymptomatic)$ |
| | 0.53 | $chol\_high \leftarrow (sex\_female \land ca\_2) \lor (bp\_normal \land cp\_asymptomatic)$ |
| | 0.62 | $hr\_high \leftarrow (cp\_asymptomatic \land target) \lor (chol\_low \land ca\_1)$ |
| | 0.70 | $st\_severe \leftarrow (cp\_non\_anginal \land \neg fbs\_normal) \lor (age\_old \land chol\_low)$ |
| RRL | 0.28 | $trestbps\_high \leftarrow (sex\_female \land \neg cp\_typical\_angina) \lor (exang\_yes \land \neg thal\_normal)$ |
| | 0.33 | $hr\_high \leftarrow (age\_middle \land sex\_male) \lor (\neg restecg\_stt\_abnormality \land slope\_upsloping)$ |
| | 0.33 | $thalach \leftarrow (age < 60) \land (restecg = 0)$ |
| | 0.32 | $st\_severe \leftarrow (\neg exang\_yes \land \neg slope\_flat) \lor (chol\_low \land cp\_asymptomatic)$ |
| BRCG | 0.53 | $trestbps\_high \leftarrow \neg age\_young \land \neg ca\_4$ |
| | 0.35 | $chol\_high \leftarrow \neg age\_young \land \neg restecg\_hypertrophy$ |
| | 0.33 | $hr\_high \leftarrow \neg cp\_typical\_angina \land \neg ca\_4$ |
| | 0.32 | $st\_severe \leftarrow \neg age\_young \land \neg slope\_upsloping$ |
| DR-NET | 0.53 | $trestbps\_high \leftarrow (chol\_low \land \neg hr\_low \land \neg fbs\_high) \lor (slope\_flat \land ca\_1 \land thal\_normal)$ |
| | 0.33 | $chol\_high \leftarrow sex\_male \land slope\_upsloping \land ca\_3$ |
| | 0.33 | $hr\_high \leftarrow \neg age\_old \land \neg cp\_typical\_angina \land fbs\_high$ |
| | 0.32 | $st\_severe \leftarrow hr\_high \land \neg sex\_male \land \neg fbs\_normal$ |
| **NS-FCN** | **0.86** | $trestbps\_high \leftarrow (age > 60) \land (chol > 250)$ |
| | **0.85** | $chol\_high \leftarrow (sex = 1 \land age > 55) \lor (trestbps > 150)$ |
| | **0.90** | $hr\_high \leftarrow (trestbps > 145) \lor (age > 57 \land cp = 3)$ |
| | **0.76** | $st\_severe \leftarrow (slope = 2) \land (thalach < 150)$ |

In *medical diagnosis*, we use Heart Disease (Detrano et al., 1989) and SPECT Heart (Kurgan et al., 2001) datasets, introducing 30% missingness. We also vary the observation ratio from 0.3 to 0.9, and the results in Tables 18 and 19 show comparable performance with only 30% of the data observed.

On the Heart Disease dataset, with its mix of continuous and categorical features, NS-FCN's direct handling of continuous values led to superior imputation (e.g., 90% accuracy for `thalach`) and

Table 6: Imputation accuracy of missing feature value comparison across Heart Disease and Bird datasets on non-interpretable baselines. Results are over 10 random seeds.

| Method | Heart Disease | | | | Birds | |
| --- | --- | --- | --- | --- | --- | --- |
| | *trestbps* | *chol* | *thalach* | *oldpeak* | *abnormal_bird* | *can_fly* |
| MICE | 0.84±0.016 | 0.83±0.014 | 0.88±0.011 | 0.87±0.015 | 0.88±0.006 | 0.86±0.011 |
| MissForest | **0.88±0.015** | 0.84±0.012 | **0.91±0.004** | 0.88±0.016 | 0.38±0.123 | 0.68±0.086 |
| MLP | **0.88±0.009** | **0.85±0.016** | 0.88±0.014 | 0.80±0.025 | **0.96±0.059** | **0.99±0.003** |
| GAIN | 0.85±0.022 | 0.84±0.011 | 0.90±0.014 | **0.89±0.014** | 0.83±0.102 | 0.82±0.083 |
| MissDiff | 0.82±0.017 | 0.83±0.019 | 0.89±0.018 | 0.84±0.030 | 0.83±0.020 | 0.86±0.007 |
| mDAE | **0.88±0.011** | 0.84±0.012 | 0.90±0.015 | 0.87±0.015 | 0.87±0.002 | 0.87±0.004 |
| VAE-based | 0.85±0.015 | 0.84±0.021 | 0.90±0.015 | 0.86±0.015 | 0.62±0.006 | 0.87±0.004 |
| **NS-FCN** | 0.87±0.025 | **0.85±0.017** | 0.88±0.014 | 0.78±0.020 | 0.95±0.064 | 0.95±0.064 |

Table 7: Medical diagnosis after missing value imputation. Results are over 10 random seeds.

| Method | Heart Disease | | SPECT | |
| --- | --- | --- | --- | --- |
| | *Accuracy* | *F1* | *Accuracy* | *F1* |
| MICE (Van Buuren & Groothuis-Oudshoorn, 2011) | 0.83±0.010 | 0.81±0.012 | 0.78±0.019 | 0.87±0.013 |
| MissForest (Stekhoven & Bühlmann, 2012) | 0.83±0.013 | 0.81±0.014 | 0.79±0.012 | 0.87±0.008 |
| MLP | 0.84±0.010 | 0.82±0.012 | **0.92±0.007** | 0.90±0.005 |
| GAIN (Yoon et al., 2018) | 0.84±0.004 | 0.82±0.006 | 0.76±0.019 | 0.85±0.013 |
| MissDiff (Ouyang et al., 2023) | 0.84±0.010 | 0.82±0.011 | 0.77±0.023 | 0.86±0.016 |
| mDAE (Dupuy et al., 2024) | 0.84±0.009 | 0.82±0.010 | 0.80±0.013 | 0.88±0.009 |
| VAE-based (Veldkamp et al., 2025) | 0.83±0.009 | 0.81±0.009 | 0.75±0.016 | 0.85±0.011 |
| BRCG (Dash et al., 2018) | 0.77±0.006 | 0.74±0.034 | 0.85±0.046 | 0.90±0.035 |
| RRL (Wang et al., 2021) | 0.78±0.002 | 0.80±0.003 | 0.90±0.005 | 0.94±0.005 |
| DR-NET (Qiao et al., 2021) | 0.85±0.005 | 0.82±0.005 | 0.89±0.025 | 0.92±0.017 |
| LEN (Barbiero et al., 2022) | 0.69±0.007 | 0.80±0.000 | 0.76±0.035 | 0.85±0.017 |
| **NS-FCN** | **0.89±0.018** | **0.89±0.018** | **0.92±0.009** | **0.96±0.009** |

the discovery of **clinically relevant rules with numerical thresholds** (e.g., $age > 60$, $chol > 250$), as shown in Tables 5 and 30. NS-FCN attains imputation accuracy comparable to the advanced statistical and generative baselines, yet distinguishes itself by offering full interpretability, a critical advantage over these black-box approaches. Compared with rule-based models, our evaluation highlights NS-FCN's unique ability to handle heterogeneous data types. A key distinction is that NS-FCN directly models continuous features, whereas **baseline methods are restricted to binary inputs**, forcing discretization (e.g., for `trestbps`, binning values into $< 120, 120 - 140, > 140$ mmHg as low, normal, and high).

On the binary SPECT dataset, we randomly miss all 22 features, thus we report the diagnosis accuracy after imputation. When the imputed features are used for diagnosis, NS-FCN outperforms all baselines on both Heart Disease and SPECT, as shown in Table 7. Unlike baseline models that train a classifier on previously imputed samples, where imputation errors inevitably propagate to the downstream task, our method jointly optimizes rule discovery and target inference. Furthermore, our use of soft-logic relaxation prevents the model from overfitting to noise (such as incorrect features), enabling it to capture dominant logical structures. This robustness is further supported by the comprehensive noise sensitivity analysis in Appendix I.1 (Tables 16 and 17), which demonstrates that the model learns valid rule approximations (e.g., capturing one correct clause) and maintains strong predictive performance even as noise levels increase. Detailed rules and LLM assessments are in Appendix Tables 28, 29, and 30.

## 8 CONCLUSION

Our NS-FCN framework effectively learns interpretable rules for missing value imputation, demonstrating strong performance across a diverse range of synthetic and real-world datasets. A key strength is its ability to seamlessly reason over heterogeneous data, handling both binary predicates (e.g., Birds) and continuous features in complex domains like medical diagnosis (e.g., SPECT and Heart Disease). It successfully handles missing data and learns hierarchical rule structures, offering significant potential for trustworthy diagnostics and transparent decision-making.

REPRODUCIBILITY STATEMENT

We have made extensive efforts to ensure the reproducibility of our results. The complete description of both the synthetic dataset generation and the real-world dataset preprocessing methods is illustrated in Appendix D.2, E, and J. Details of the computational setup, including hardware configuration and software environment, as well as the choice of hyperparameters, are documented in Appendix J.4 and K.3. We also release our code.

ACKNOWLEDGEMENTS

We thank the anonymous reviewers and the area chair for their constructive feedback, which helped improve the paper. This work was supported in part by the Key Program of the National Natural Science Foundation of China (NSFC) under Grant No. 72495131; the Shenzhen Stability Science Program 2023; the Shenzhen Science and Technology Program No. JCYJ20250604141038013; and the Longgang District Key Laboratory of Intelligent Digital Economy Security.

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

APPENDIX OVERVIEW

- **Section A** extends the related-work discussion and provides broader context on ILP and missing-data imputation.
- **Section B** gives a theoretical convergence view of our block coordinate optimization, including assumptions and stationarity statements.
- **Section C** summarizes model-supplement details, including optimization implementation notes (Section C.1).
- **Section D** describes datasets (Section D.1), preprocessing pipelines (Section D.2), and baseline configurations used in experiments (Section D.3).
- **Section E** reports performance under different missingness mechanisms (MCAR/MAR/MNAR).
- **Section F** provides runtime and memory analyses on synthetic and real-world settings.
- **Section G** evaluates structural stability of learned rules (e.g., Jaccard-based analyses in Section G.1).
- **Section H** presents sensitivity analysis for soft-operator temperatures.
- **Section I** reports robustness under label noise (Section I.1) and varying missing ratios (Section I.2).
- **Section J** provides additional synthetic experiment results and extended case studies.
- **Section K** provides additional real-world experiment analyses and supplementary rule assessments.
- **Section L** discusses limitations and broader impact considerations.
- **Section M** clarifies how LLMs are used in manuscript polishing and rule-assessment narration.

## A    RELATED WORK SUPPLEMENT

**Traditional Inductive Logic Programming (ILP) Methods.** Inductive Logic Programming learns logical rules from relational data. Cohen (1995) proposed RIPPER, a fast rule induction algorithm using separate-and-conquer strategy. Quinlan (1990) developed FOIL, which generates clauses iteratively. Dash et al. (2018) introduced Boolean decision rules using column generation. Wei et al. (2019) proposed GLRM, integrating decision rules into linear models. Cropper & Morel (2021) presented LFF implemented in Popper. These approaches rely on heuristics but may not guarantee optimal solutions. Pellegrina & Vandin (2024) proposed SamRuLe for near-optimal rule lists via sampling.

**Differentiable ILP Methods.** Traditional ILP models struggle with noisy data and scalability. Differentiable approaches address these issues by integrating continuous relaxation, which allows gradient descent for optimization. Shindo et al. (2021) proposed $\partial ILP$, which represents logic rules in a differentiable form and combines neural networks with symbolic logic. Manhaeve et al. (2018) introduced DeepProbLog, extending ProbLog with neural predicates. Neural Logic Machines (NLMs) (Dong et al., 2019) combine MLPs with logic programming to improve computational efficiency but reduce interpretability.

**Broader Missing Data Imputation Methods.** Missing data imputation methods range from global model-based techniques to localized and hybrid strategies, extending to deep and ensemble frameworks.

At the global end, nonparametric bootstrap methods (Efron, 1994) provide bias-corrected estimates via repeated sampling, while spectral regularization approaches like SOFT-IMPUTE (Mazumder et al., 2010) solve a nuclear-norm minimization through iterative soft-thresholded SVD. Classical multivariate imputation schemes such as MICE (Van Buuren & Groothuis-Oudshoorn, 2011) construct a sequence of conditional models for each variable with missingness and iteratively sample from these chained regressions until convergence, thereby approximating draws from the joint posterior and naturally propagating uncertainty across multiple imputations. Tree-based ensemble methods such as MissForest (Stekhoven & Bühlmann, 2012) adopt an iterative refinement strategy in which random forests are trained per variable using the currently imputed data as predictors, updating missing entries via out-of-bag predictions until changes stabilize, thus capturing complex nonlinearities and high-order interactions without requiring parametric distributional assumptions.

Moving toward local adaptation, decision tree–based EM (DMI) (Rahman & Islam, 2011) partitions complete cases via C4.5 and imputes within each leaf, and clustering-based random imputation (CRI) (Zhang et al., 2006) applies kernel-weighted estimation in the nearest k-means cluster. Hybrid similarity learners, such as KI and its fuzzy extension FCKI (Fouad et al., 2021), refine this idea by dynamically selecting neighborhood sizes before multivariate imputation. For high-dimensional or heterogeneous data, deep architectures like GAIN (Yoon et al., 2018) cast imputation as a generative adversarial game where a generator proposes imputations conditioned on an observed-mask vector and a discriminator learns to distinguish observed from imputed components, while VAE-based imputers (Veldkamp et al., 2025) treat the complete feature matrix as generated from low-dimensional latent variables and learn to reconstruct missing entries via amortized variational inference under a probabilistic encoder–decoder architecture. Building on denoising autoencoders, mDAE (Dupuy et al., 2024) modifies the reconstruction loss to ignore pre-imputed values at missing positions and couples this with an overcomplete hidden representation, which empirically improves RMSE over standard DAEs and several classical imputers across multiple UCI datasets (Dupuy et al., 2024). In the same spirit of generative modeling, MissDiff (Ouyang et al., 2023) trains a diffusion model on tabular data with missing values by injecting noise along a forward stochastic process and learning a reverse denoising process that is explicitly conditioned on the observed-mask pattern, thereby producing imputations through iterative refinement from pure noise. Models such as MMDL (Li et al., 2020) align stacked autoencoder embeddings across modalities to exploit cross-view correlations. Ensemble schemes like FIMUS (Rahman & Islam, 2014) combine co-appearance, correlation, and similarity in a weighted-voting framework. Despite their varied focuses—ranging from global inference to localized and multimodal learning—these methods uniformly rely on statistical patterns and *do not leverage explicit logical rules to govern inter-variable relationships.*

## B  Convergence Analysis of Coordinate Gradient Descent

For clarity, we analyze a simplified version of our learning algorithm in which each head predicate $U_k$ is associated with a single parameter block $\Theta_k$. Let $\Theta = (\Theta_1, \ldots, \Theta_m)$ collect all parameters. The global training objective is

$$\mathcal{L}(\Theta) = \sum_{k=1}^{m} \mathcal{L}_{U_k}(\Theta), \qquad \mathcal{L}_{U_k}(\Theta) = \text{mean}\big((v_{U_k}(\Theta) \odot \boldsymbol{M}_k - U_{k,\text{obs}} \odot \boldsymbol{M}_k)^2\big), \tag{8}$$

where $v_{U_k}(\Theta)$ is computed by forward chaining using the differentiable operators introduced in the main text (e.g., Eqs. 1, 2, 4).

### B.1  Assumptions

We make the following standard assumptions for smooth block coordinate descent (e.g., Tseng (2001); Bertsekas (1997); Nesterov (2013)).

**Assumption 1** *The objective $\mathcal{L} : \mathbb{R}^d \to \mathbb{R}$ is*

1. *bounded below:* $\inf_\Theta \mathcal{L}(\Theta) > -\infty$,
2. *continuously differentiable in $\Theta$, and*
3. *has block-wise Lipschitz-continuous gradients: for each $k$ there exists $L_k < \infty$ such that, for all $\Theta$ and all $h_k$,*
$$\big\|\nabla_{\Theta_k}\mathcal{L}(\Theta + e_k h_k) - \nabla_{\Theta_k}\mathcal{L}(\Theta)\big\| \leq L_k \|h_k\|, \tag{9}$$
*where $e_k h_k$ denotes the vector obtained by changing only block $k$.*

These conditions hold in our setting because $\mathcal{L}$ is built from smooth operations (e.g., linear maps, sigmoid, softmin, log-sum-exp) composed with a squared loss, and training is restricted to bounded level sets.

### B.2  Idealized Full-Batch Block Coordinate Gradient Descent

Consider the following idealized algorithm. At iteration $\tau$ we pick a block index $k_\tau \in \{1, \ldots, m\}$ (e.g., by cycling through $\{1, \ldots, m\}$) and perform a gradient step on that block only:

$$\Theta_{k_\tau}^{\tau+1} = \Theta_{k_\tau}^\tau - \eta \, \nabla_{\Theta_{k_\tau}}\mathcal{L}(\Theta^\tau), \tag{10}$$

$$\Theta_\ell^{\tau+1} = \Theta_\ell^\tau \qquad \text{for all } \ell \neq k_\tau, \tag{11}$$

where $\eta > 0$ is a step size. This matches the idealized version of the rule update in Section 5.1: when we update $U_{k_\tau}$, all other predicates $U_\ell$ are kept fixed.

**Lemma 1 (Monotone decrease for small steps)** *Suppose Assumption 1 holds. If the step size satisfies $0 < \eta \leq 1/L_{k_\tau}$ at iteration $\tau$, then*

$$\mathcal{L}(\Theta^{\tau+1}) \leq \mathcal{L}(\Theta^\tau) - \frac{\eta}{2}\left\|\nabla_{\Theta_{k_\tau}}\mathcal{L}(\Theta^\tau)\right\|^2. \tag{12}$$

*In particular, the sequence $\{\mathcal{L}(\Theta^\tau)\}_{\tau \geq 0}$ is monotonically non-increasing and convergent.*

**Proof 1 (Proof sketch)** *By block-wise Lipschitz continuity of $\nabla_{\Theta_{k_\tau}}\mathcal{L}$,*

$$\mathcal{L}(\Theta^{\tau+1}) = \mathcal{L}\big(\Theta^\tau + e_{k_\tau}(\Theta_{k_\tau}^{\tau+1} - \Theta_{k_\tau}^\tau)\big) \tag{13}$$

$$\leq \mathcal{L}(\Theta^\tau) + \big\langle \nabla_{\Theta_{k_\tau}}\mathcal{L}(\Theta^\tau), \Theta_{k_\tau}^{\tau+1} - \Theta_{k_\tau}^\tau \big\rangle + \frac{L_{k_\tau}}{2}\|\Theta_{k_\tau}^{\tau+1} - \Theta_{k_\tau}^\tau\|^2. \tag{14}$$

*Substituting the update $\Theta_{k_\tau}^{\tau+1} - \Theta_{k_\tau}^\tau = -\eta\,\nabla_{\Theta_{k_\tau}}\mathcal{L}(\Theta^\tau)$ and rearranging gives*

$$\mathcal{L}(\Theta^{\tau+1}) \leq \mathcal{L}(\Theta^\tau) - \eta\Big(1 - \frac{\eta L_{k_\tau}}{2}\Big)\big\|\nabla_{\Theta_{k_\tau}}\mathcal{L}(\Theta^\tau)\big\|^2. \tag{15}$$

*If $\eta \leq 1/L_{k_\tau}$, then $1 - \eta L_{k_\tau}/2 \geq 1/2$, yielding the claimed inequality.*

Lemma 1 implies that the loss decreases at every iteration and the gradients on updated blocks cannot stay large forever. Combined with a mild assumption that each block is selected infinitely often, we obtain convergence to a block-stationary point.

**Proposition 1 (Convergence to a block-stationary point)** *Assume Assumption 1 holds, the level set $\{\Theta : \mathcal{L}(\Theta) \leq \mathcal{L}(\Theta^0)\}$ is bounded, each block $k$ is selected infinitely often, and the step sizes satisfy $0 < \eta \leq \min_k 1/L_k$. Then any limit point $\Theta^\star$ of the sequence $\{\Theta^\tau\}$ generated by the above block coordinate gradient method is* block-stationary:

$$\nabla_{\Theta_k}\mathcal{L}(\Theta^\star) = 0 \quad \text{for all } k = 1, \ldots, m. \tag{16}$$

*Equivalently, no single block $\Theta_k$ can be perturbed to decrease $\mathcal{L}$ while all other blocks are fixed.*

**Proof 2 (Proof sketch)** *Summing the inequality from Lemma 1 over $\tau$ shows that*

$$\sum_{\tau=0}^{\infty}\big\|\nabla_{\Theta_{k_\tau}}\mathcal{L}(\Theta^\tau)\big\|^2 < \infty,$$

*so the block gradients must tend to zero along the subsequence where a given block $k$ is updated. Since each block is selected infinitely often and the iterates remain in a bounded level set, standard arguments for block coordinate descent (Tseng, 2001) imply that any limit point has zero gradient in every block.*

Thus, in the ideal full-batch setting with sufficiently small steps, our predicate-wise coordinate updates produce a non-increasing loss sequence $\{\mathcal{L}(\Theta^\tau)\}$ and converge to a point where no single predicate block $\Theta_k$ can further reduce the global objective.

### B.3 STOCHASTIC MINI-BATCH VARIANT AND ADAM

In practice, our implementation uses mini-batches and the Adam optimizer for each block update (as described in Section 5.1). In this case, the gradient $\nabla_{\Theta_k}\mathcal{L}$ is replaced by a stochastic estimate computed on a mini-batch, and the step uses Adam's adaptive preconditioning. This yields a *stochastic* block-coordinate gradient scheme: the loss is no longer guaranteed to decrease at every single update, but under standard assumptions, stochastic block-coordinate methods are known to approach a neighborhood of a stationary point in expectation (see, e.g., Richtárik & Takáč (2014); Wright (2015)).

## C MODEL SUPPLEMENT DESCRIPTION

### C.1 OPTIMIZATION DETAILS

Throughout all training stages, each rule embedding (or set of embeddings during joint fine-tuning) is optimized using the Adam optimizer. A crucial step following each gradient update is the normalization of the rule embeddings. This involves applying a Rectified Linear Unit (ReLU) activation to the embedding data (ensuring non-negative values, which can aid interpretability for positive predicate contributions) followed by $L_2$ normalization of each row vector within the rule embedding matrix. This normalization helps stabilize the training process and maintains consistent magnitudes for the embedding components.

# D  DATASETS AND BASELINES

## D.1  DATASETS

**Heart Disease.** We use the widely-cited Cleveland Clinic dataset from the UCI Heart Disease database (Detrano et al., 1989). This dataset contains 303 patient records, each with 13 features—a mix of continuous and categorical variables—such as age, cholesterol level, and resting blood pressure. The task is to predict the presence of heart disease, which is indicated by the target variable on a scale from 0 (absence) to 4 (severe). Following standard practice, we simplify this into a binary classification problem: predicting presence (values 1-4) versus absence (value 0).

**SPECT.** The SPECT (Single Proton Emission Computed Tomography) dataset presents a binary classification task to diagnose cardiac conditions (normal/abnormal) based on 22 binary patient features. The dataset describes the diagnosis of cardiac SPECT images. Each of the patients is classified into two categories: normal and abnormal. The 267 SPECT image sets (patients) database was processed to extract features that summarize the original SPECT images. As a result, 44 continuous feature patterns were created for each patient. The pattern was further processed to obtain 22 binary feature patterns. The CLIP3 algorithm was used to generate classification rules from these patterns (Kurgan et al., 2001). The CLIP3 algorithm generated rules that were 84.0% accurate (as compared with cardiologists' diagnoses). A key challenge in this domain is the prevalence of missing data, making it an ideal testbed for our model's imputation and rule-learning capabilities.

**Birds.** Bird's Rulebase is a well-known logic problem designed to assess an AI's ability to learn and reason with hierarchical logical rules that mimic common-sense knowledge (Tafjord et al., 2021). It has the ground truth single theory of six rules [2] as follows.

$$\text{can\_fly}(X) \leftarrow \text{bird}(X), \text{not abnormal\_bird}(X)$$
$$\text{bird}(X) \leftarrow \text{ostrich}(X)$$
$$\text{abnormal\_bird}(X) \leftarrow \text{ostrich}(X)$$
$$\text{not can\_fly}(X) \leftarrow \text{ostrich}(X)$$
$$\text{abnormal\_bird}(X) \leftarrow \text{bird}(X), \text{wounded}(X)$$
$$\text{not can\_fly}(X) \leftarrow \text{wounded}(X)$$

Figure 5 further illustrates the structure of these rules.

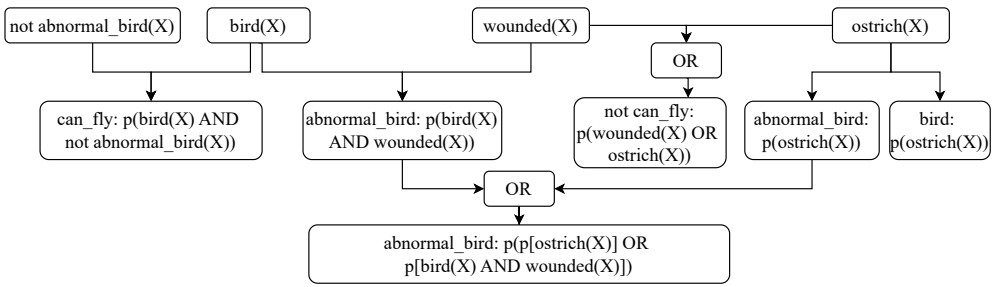

Figure 5: Ground truth rules for Bird dataset.

## D.2  PREPROCESSING OF DATASETS

**Heart Disease.** The UCI Heart Disease dataset contains a mix of 13 continuous and categorical features with 303 samples. To create a challenging imputation task, we introduced a 30% missing ratio independently into four key continuous variables: resting blood pressure (`trestbps`), cholesterol (`chol`), maximum heart rate (`thalach`), and ST depression (`oldpeak`). Following the protocol in MissDiff (Ouyang et al., 2023), we generate missing values under a Missing Completely At Random (MCAR) mechanism. Let $\mathbf{x} \in \mathbb{R}^d$ denote the complete data vector. We generate a binary mask vector $\mathbf{m} \in \{0, 1\}^d$, where $m_i = 1$ indicates that $x_i$ is observed, and $m_i = 0$ indicates it is missing. The observed data is represented as $\tilde{\mathbf{x}} = \mathbf{x} \odot \mathbf{m} + \text{na} \odot (1 - \mathbf{m})$, where $\odot$ denotes element-wise multiplication.

---

[2]https://www.doc.ic.ac.uk/ mjs/teaching/KnowledgeRep491/ExtendedLP 491-2x1.pdf, p5

For our NS-FCN framework, the task is to directly impute these missing continuous values. For deep learning baselines, continuous features are standardized using Z-score normalization, and categorical features are one-hot encoded. For tree-based and statistical baselines (MissForest, MICE), categorical variables are treated as factors. However, to accommodate the baseline models which only support binary inputs, we first discretized these four variables into three categorical bins based on clinical thresholds: blood pressure ($< 120, 120 - 140, > 140$), cholesterol ($< 200, 200 - 240, \geq 240$), max heart rate ($< 100, 100 - 160, \geq 160$), and ST depression ($\leq 1.0, 1.0 - 2.0, > 2.0$). The baselines were then tasked with imputing the correct category. Consequently, we evaluate the imputation accuracy on the discretized bins.

**SPECT Heart.** The dataset's 22 binary features were randomly masked with a 30% probability to simulate missing data. Our framework was then applied to a two-stage task: first, to impute the missing features, and second, to perform the final patient diagnosis based on the completed feature set. The diagnostic performance is compared against five baseline methods, including four rule-based approaches and an MLP.

**Birds.** Following the ground truth logical rules, we generated a dataset of 1,500 samples. To create a difficult logical reasoning challenge, we introduced a 90% missing ratio for two crucial latent predicates: can_fly and abnormal_bird. The task for all models was to impute these missing binary values based on the observed predicates. The imputation accuracy is compared against the same set of baselines.

### D.3 BASELINE MODELS

To rigorously evaluate performance, we compare our method against 11 established baselines, ranging from classical statistical methods to advanced deep generative models and interpretable models.

**Statistical Models.**

- **MICE** (Van Buuren & Groothuis-Oudshoorn, 2011)

  Multivariate Imputation by Chained Equations (MICE) is a widely used statistical method based on Fully Conditional Specification (FCS). It iteratively imputes missing values by modeling each feature with missing data as a function of other features using linear regression (for continuous variables) or logistic regression (for categorical variables). We generate $m = 5$ imputed datasets and report results from the first completion.

- **MissForest** (Stekhoven & Bühlmann, 2012)

  MissForest is a non-parametric method that handles mixed-type data using an iterative Random Forest approach. It treats the missing data problem as a prediction task, training a random forest on the observed parts of the data to predict the missing values. It is particularly effective at capturing non-linear interactions without explicit distributional assumptions.

**Deep Generative Models.**

- **MLP (Multilayer Perceptron)**

  We use a simple feed-forward neural network with fully connected layers and ReLU activations as a deterministic imputation baseline. Given an input vector $\mathbf{x} \in \mathbb{R}^d$ and a binary mask $\mathbf{m} \in \{0, 1\}^d$ indicating observed entries ($m_j = 1$ if $x_j$ is observed, 0 otherwise), we first obtain $\tilde{\mathbf{x}} = \mathbf{x} \odot \mathbf{m} + \mathrm{na} \odot (1 - \mathbf{m})$, and use the observed mask for input gating:
  $$\mathbf{h}_0 = \tilde{\mathbf{x}} \odot \mathbf{m}.$$
  The network $f_\theta$ takes $\mathbf{h}_0$ as input and outputs a reconstruction $\hat{\mathbf{x}} = f_\theta(\mathbf{h}_0)$. Training is performed under weak supervision by minimizing the Mean Squared Error (MSE) *only* on observed entries:
  $$\mathcal{L}_{\mathrm{MLP}} = \|(\hat{\mathbf{x}} - \mathbf{x}) \odot \mathbf{m}\|_2^2,$$
  so that gradients are propagated only through coordinates with ground-truth observations; at test time, the missing entries ($m_j = 0$) are imputed using the corresponding components of $\hat{\mathbf{x}}$.

- **VAE (Variational Autoencoder)**

  Our VAE-based imputer follows the amortized inference framework of Kingma & Welling (2013), adapted to incomplete tabular data as in recent work on VAE with missingness (e.g. Veldkamp et al. (2025)). Given $(\mathbf{x}, \mathbf{m})$, we construct a gated and masked input
  $$\tilde{\mathbf{x}} = \mathbf{x} \odot \mathbf{m} + \mathrm{na} \odot (1 - \mathbf{m}), \quad \mathbf{h}_0 = \tilde{\mathbf{x}} \odot \mathbf{m},$$

and feed the concatenated vector $[\mathbf{h}_0, \mathbf{1}-\mathbf{m}]$ into the encoder to obtain a Gaussian posterior

$$q_\phi(\mathbf{z} \mid \mathbf{x}, \mathbf{m}) = \mathcal{N}(\boldsymbol{\mu}_\phi, \mathrm{diag}(\boldsymbol{\sigma}_\phi^2)).$$

A latent sample $\mathbf{z}$ is drawn via the reparameterization trick and passed through a decoder $p_\theta(\mathbf{x} \mid \mathbf{z})$ to produce $\hat{\mathbf{x}}_\theta(\mathbf{z})$. The model is trained by maximizing the Evidence Lower Bound (ELBO), where the reconstruction term only involves *observed* entries:

$$\mathcal{L}_{\mathrm{VAE}} = \underbrace{\left\| \left( \hat{\mathbf{x}}_\theta(\mathbf{z}) - \mathbf{x} \right) \odot \mathbf{m} \right\|_2^2}_{\text{reconstruction on observed data}} + \underbrace{\mathrm{KL}\big( q_\phi(\mathbf{z} \mid \mathbf{x}, \mathbf{m}) \,\|\, p(\mathbf{z}) \big)}_{\text{KL regularization}}.$$

At inference time, missing values are imputed by the decoder output $\hat{\mathbf{x}}_\theta(\mathbf{z})$ at coordinates where $m_j = 0$.

- **DAE / mDAE (modified Denoising Autoencoder)**

  For the autoencoder baseline, we adopt a denoising autoencoder architecture with a modification of the loss function proposed in the mDAE (Dupuy et al., 2024). Given $(\mathbf{x}, \mathbf{m})$, we first perform a simple pre-imputation to obtain a complete input $\tilde{\mathbf{x}}$, and then apply masking noise with rate $\rho$ *only* on originally observed entries:

  $$\tilde{\mathbf{x}} = \mathbf{x} \odot \mathbf{m} + \mathrm{na} \odot (1 - \mathbf{m}), \quad \mathbf{c} \sim \mathrm{Ber}(\rho)^d, \quad \tilde{\mathbf{x}}^{(\mathrm{noisy})} = (\tilde{\mathbf{x}} \odot \mathbf{m}) \odot \left( 1 - \mathbf{c} \odot \mathbf{m} \right).$$

  The corrupted input $\tilde{\mathbf{x}}^{(\mathrm{noisy})}$ is fed into an encoder–decoder network $g_\psi$ that outputs a reconstruction $\hat{\mathbf{x}} = g_\psi(\tilde{\mathbf{x}}^{(\mathrm{noisy})})$. Crucially, following the modified-loss idea of mDAE (Dupuy et al., 2024), the reconstruction loss is computed *only on truly observed entries*, and pre-imputed missing values are ignored:

  $$\mathcal{L}_{\mathrm{mDAE}} = \left\| \left( \hat{\mathbf{x}} - \mathbf{x} \right) \odot \mathbf{m} \right\|_2^2.$$

  This prevents the autoencoder from overfitting arbitrary pre-imputed values at missing positions while still benefiting from denoising training; at test time, imputations for missing entries ($m_j = 0$) are taken from the corresponding components of $\hat{\mathbf{x}}$.

- **GAIN (Generative Adversarial Imputation Nets) (Yoon et al., 2018)**

  GAIN adapts the Generative Adversarial Network framework for imputation. The generator $G$ imputes missing components, while the discriminator $D$ attempts to distinguish between observed and imputed components. A hint mechanism is introduced to provide $D$ with partial information about the mask distribution, forcing $G$ to learn the true underlying data distribution. We utilize a hybrid loss function combining adversarial loss with MSE for continuous features and cross-entropy for categorical features.

- **MissDiff (Diffusion Imputation Nets) (Ouyang et al., 2023)**

  We employ a diffusion probabilistic model specifically adapted for tabular missing data. The model is trained to reverse a noise-adding process. During inference (imputation), we utilize the *guided sampling* or *conditioning* strategy: at each denoising step $t$, the known observed values $\mathbf{x}^{obs}$ ($\mathbf{x}^{obs} = \mathbf{x} \odot \mathbf{m} + \mathrm{na} \odot (1 - \mathbf{m})$) are re-injected into the sample to ensure consistency with the ground truth. The model effectively samples $\mathbf{x}^{imp}$ from the conditional distribution $p(\mathbf{x}^{miss} | \mathbf{x}^{obs})$.

**Interpretable Models.**

- **BRCG** (Dash et al., 2018) is an integer program designed to trade classification accuracy for rule simplicity. It uses column generation to search over an exponential number of candidate clauses efficiently.

- **LEN** (Barbiero et al., 2022) is an end-to-end differentiable method for extracting logical explanations from neural networks using First-Order Logic.

- **DR-NET** (Qiao et al., 2021) is a method for learning independent logical rules in disjunctive standard form as an interpretable model for classification.

- **RRL** (Wang et al., 2021) learns interpretable non-fuzzy rules for data representation and classification using a novel training method called Gradient Grafting.

## E  PERFORMANCE UNDER DIFFERENT MISSINGNESS MECHANISMS

We compare three general missingness mechanisms for dataset generation:

- **MCR (Missing Completely at Random)**: The probability of being missing is the same for all cases, which is the missingness mechanism in other experiments in our paper.

- **MAR (Missing at Random)**: Missingness depends on observed variables. We can indicate which observed variable to use for missingness; the default is $X_0$. Then, we set a higher probability of missing when the dependency variable is 1.
- **MNAR (Missing Not at Random)**: Missingness depends on unobserved variables or the missing values themselves. Take $X_3$ for example, we set it is more likely to be missing when $X_3 = 1$ (positive values are harder to observe).

We show an observation ratio = 0.2 and a sample size = 20,000 as a representative case in Table 8. We run 20 random seeds. Since the seeds are different from those used in Tables 21 and 22, the results are slightly different.

Table 8: Comparison of inference accuracy and rule accuracy under different missing mechanisms.

|  | MCAR | | MAR | | MNAR | |
|---|---|---|---|---|---|---|
|  | Imputation Accu. | Rule Accu. | Imputation Accu. | Learned Rules | Imputation Accu. | Rule Accu. |
| $X_3$ | 1.00 ± 0.00 | 1.0 | 1.00 ± 0.00 | 1.0000 | 1.00 ± 0.00 | 1.0000 |
| $X_4$ | 1.00 ± 0.00 | 1.0 | 1.00 ± 0.00 | 1.0000 | 1.00 ± 0.00 | 1.0000 |
| $X_5$ | 0.95 ± 0.07 | 0.6 | 0.95 ± 0.07 | 0.6000 | 0.93 ± 0.05 | 0.4000 |

The results show that MAR and MNAR show comparable results to MCAR, which demonstrates our method's effectiveness across the full spectrum of missing data scenarios.

## F RUNNING TIME AND MEMORY COST ANALYSIS

### F.1 SYNTHETIC DATASET

While coordinate descent requires different cycle numbers (Table 3), our method demonstrates efficient performance on standard CPU configurations. We conducted experiments using an Apple M4 chip with 10 cores and 16GB memory, taking observation ratio = 0.2 as an example. Results over 20 runs on setting (b) of Figure 3.

Table 9: Running time and memory cost of our model with varying sample sizes. Results over 20 seeds on the example (b) of Figure 3.

| Sample size | 2500 | 5000 | 10,000 | 25,000 | 50,000 | 100,000 |
|---|---|---|---|---|---|---|
| Running time (s) | 15.66±3.48 | 30.17±2.12 | 54.49±15.98 | 130.36±45.80 | 194.59±98.53 | 493.99±152.81 |
| Memory cost (MB) | 64.84±10.72 | 71.92±0.82 | 78.55±1.96 | 95.81±12.13 | 126.99±26.97 | 175.64±33.93 |

Overall, we observe **minimal time and memory costs**. Time complexity scales near-linearly with increasing sample size, while memory requirements remain modest even for large datasets. Processing 100,000 samples in under 9 minutes demonstrates strong efficiency for CPU-based execution.

### F.2 REAL-WORLD DATASET

Table 10: Comparison of running time and memory cost across different methods in SPECT dataset.

| Method | Running time (s) | Memory cost (MB) |
|---|---|---|
| MLP | 0.16 ± 0.02 | 158.60 ± 0.10 |
| LEN | 0.20 ± 0.00 | 102.78 ± 0.01 |
| RRL | 16.23 ± 0.01 | 132.59 ± 0.01 |
| BRCG | 2.65 ± 0.27 | 135.11 ± 0.08 |
| DR-NET | 89.01 ± 0.06 | 45.93 ± 0.30 |
| **NS-FCN (Ours)** | 10.34 ± 0.30 | 61.42 ± 1.02 |

We conducted a comparative analysis of our proposed NS-FCN model against baseline methods, focusing on computational efficiency. We take SPECT dataset as an example. The results in Table 10 demonstrate that NS-FCN achieves a competitive balance between performance and resource consumption. While methods like MLP and LEN offer the fastest execution times, they use higher

memory costs. Our NS-FCN, though not the fastest, maintains a considerably minimal memory cost and running time.

## G ASSESSMENT OF RULE QUALITY

### G.1 STRUCTURAL STABILITY.

To quantify the structural stability and reliability of the learned rules, we measure the consistency of rule predicates across different random seeds using the Jaccard index. For each rule, we treat the set of instances that satisfy its predicates in a given run as a binary mask, and compute pairwise Jaccard indices between runs obtained under different random seeds and observation probabilities. The **Jaccard index**, defined as the intersection over union of two predicate sets, provides a natural measure of similarity between rule structures learned across independent runs. High mean Jaccard scores (close to 1.0) indicate that the learned rules are structurally stable and robust to stochasticity in training and sampling, whereas lower scores reveal predicates whose semantics are more sensitive to noise or initialization.

**Synthetic Dataset (Figure 3 (b)).** As shown in Table 11, rules $X_3$ and $X_4$ achieve perfect Jaccard indices of 1.0 across all observation probabilities, demonstrating complete structural stability. In contrast, the aggregated $X_5$ rule exhibits more variability, reflecting the increased complexity of learning disjunctive rule structures. In this way, structural stability—measured via the Jaccard index of predicates across runs—provides a complementary notion of reliability that focuses on the consistency of the learned logical structure rather than solely on predictive performance.

Table 11: Jaccard index of learned rule predicates on synthetic data under different observation probabilities. Example (b) of Figure 3 with 20,000 samples over 30 random seeds. These results correspond to the rule structures reported in Table 22.

| Obs. Ratio | $X_3$ | $X_4$ | $X_5$ |
|---|---|---|---|
| 0.1 | $1.0000 \pm 0.0000$ | $1.0000 \pm 0.0000$ | $0.6707 \pm 0.2270$ |
| 0.2 | $1.0000 \pm 0.0000$ | $1.0000 \pm 0.0000$ | $0.6568 \pm 0.2831$ |
| 0.3 | $1.0000 \pm 0.0000$ | $1.0000 \pm 0.0000$ | $0.7124 \pm 0.2279$ |

Table 12: Jaccard Index of learned predicates across different sample sizes on the Birds dataset. Results over 10 seeds.

| Sample Size | abnormal_clause1 (*ostrich*) | abnormal_clause2 (*bird* $\wedge$ *wounded*) | can_fly (*bird* $\wedge \neg$ *abnormal_bird*) |
|---|---|---|---|
| 100 | $0.8000 \pm 0.2449$ | $0.5000 \pm 0.3162$ | $0.5000 \pm 0.3162$ |
| 500 | $0.8000 \pm 0.2449$ | $0.8000 \pm 0.2449$ | $0.7000 \pm 0.2449$ |
| 1000 | $0.8000 \pm 0.2449$ | $0.6000 \pm 0.3000$ | $0.6000 \pm 0.3000$ |
| 1500 | $0.8187 \pm 0.2404$ | $0.6868 \pm 0.3024$ | $0.7967 \pm 0.2670$ |
| 2000 | $0.8000 \pm 0.2449$ | $0.6000 \pm 0.3000$ | $0.6000 \pm 0.3000$ |

**Birds Dataset.** We analyze the consistency of learned rule structures in Birds Dataset (Figure 5). Table 12 presents the Jaccard indices across all pairwise comparisons between seeds for different sample sizes, where *abnormal_clause1* and *abnormal_clause2* correspond to the two conjunctive clauses in the disjunctive rule for *abnormal_bird*: *abnormal_bird* $\leftarrow$ *ostrich* $\vee$ (*bird* $\wedge$ *wounded*).

The results demonstrate that, with the exception of $n = 100$ where the sample size is insufficient, the model achieves good consistency (Jaccard index $> 0.60$) across all rules and sample sizes. Overall, $n = 1500$ yields the best consistency, with *abnormal_clause1* reaching 0.8187 and *can_fly* reaching 0.7967, indicating that this sample size provides an optimal balance between data availability and model stability.

**Heart Disease dataset.** For this real-world dataset, where ground-truth rules are unknown, we evaluate structural stability by computing the Jaccard index of selected features across all prediction rules learned under different random seeds. Table 13 shows that the model achieves moderate

Table 13: Structural stability of learned prediction rules on the Heart Disease dataset.

| Metric | Value |
|---|---|
| **Mean Pairwise Jaccard Index** | $0.4151 \pm 0.0994$ |
| **Most Frequently Selected Features** | |
| *restecg_1.0* (ST-T wave abnormality) | 9/10 runs |
| *thal_3.0* (normal thalassemia) | 8/10 runs |
| *ca_3.0* (3 major vessels colored) | 8/10 runs |
| *thalach* (maximum heart rate achieved) | 7/10 runs |

consistency (Jaccard index $0.4151 \pm 0.0994$), indicating that while different seeds may select varying feature combinations, there is substantial overlap in the most important features. The most frequently selected features include *restecg_1.0* (ST-T wave abnormality on resting electrocardiogram), *thal_3.0* (normal thalassemia, a blood disorder), *ca_3.0* (three major vessels colored by fluoroscopy, indicating severe coronary artery disease), and *thalach* (maximum heart rate achieved during exercise). These features align with established clinical risk factors for heart disease, suggesting that the model successfully identifies medically relevant features despite the lack of explicit rule supervision.

### G.2 RULE LENGTH ANALYSIS

To understand the sensitivity of our framework to the rule structure hyperparameters, we conduct ablation studies on the Heart Disease dataset, systematically varying the arity of conjunction ($h$) and the number of conjunctive clauses ($R_k$).

We find that both $h$ and $R_k$ show optimal performance in a wide range. For instance, $h \in [3, 9]$ and $R_k \in [5, 20]$, showing that except for very small $h$ and $R_k$, our model is able to capture the logic structure within the dataset. Besides, the number of disjunctive clauses is more critical than the arity of individual conjunctions for this dataset. This aligns with the intuition that complex real-world decision boundaries often require multiple alternative rules rather than highly complex single rules.

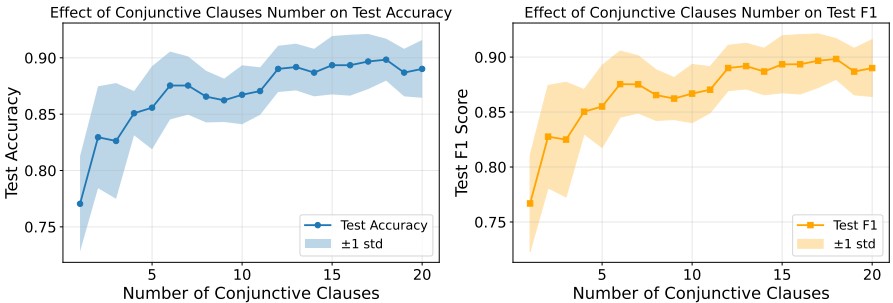

Figure 6: Classification accuracy for heart disease risk under the effect of the number of conjunction arity ($h$). Results are over 10 seeds.

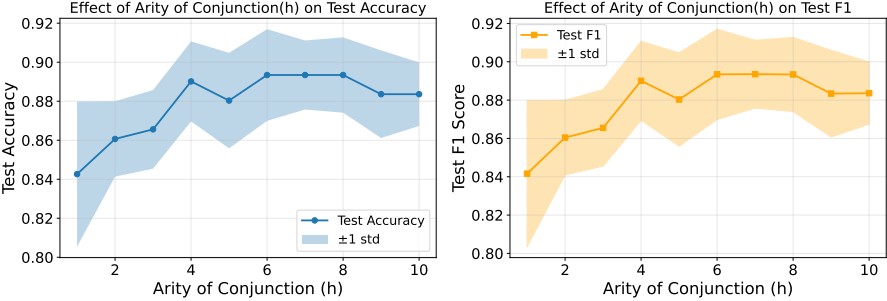

Figure 7: Classification accuracy for heart disease risk under the effect of the number of conjunctive Clauses ($R_k$). Results are over 10 seeds.

## H  ANALYSIS OF TEMPERATURE IN SOFT OPERATORS

To validate robustness, we conducted a sensitivity analysis on Figure 3(b) with 20,000 samples and 0.2 observation ratio.

Table 14: Imputation accuracy for latent predicates $X_3, X_4, X_5$ under different softmin temperatures (with fixed $\mu = 10$ over 20 random seeds).

| $\tau$ in Eq. 3 | Imputation Acc. X3 | Imputation Acc. X4 | Imputation Acc. X5 |
|---|---|---|---|
| 0.01 | $1.000 \pm 0.000$ | $1.000 \pm 0.000$ | $0.965 \pm 0.063$ |
| 0.02 | $1.000 \pm 0.000$ | $1.000 \pm 0.000$ | $0.987 \pm 0.034$ |
| 0.05 | $1.000 \pm 0.000$ | $1.000 \pm 0.000$ | $0.939 \pm 0.061$ |
| 0.10 | $1.000 \pm 0.000$ | $1.000 \pm 0.000$ | $0.958 \pm 0.076$ |
| 0.20 | $1.000 \pm 0.000$ | $1.000 \pm 0.000$ | $1.000 \pm 0.000$ |
| 0.50 | $1.000 \pm 0.000$ | $1.000 \pm 0.000$ | $0.858 \pm 0.038$ |
| 1.00 | $0.928 \pm 0.122$ | $0.892 \pm 0.134$ | $0.776 \pm 0.027$ |
| 2.00 | $0.751 \pm 0.002$ | $0.839 \pm 0.157$ | $0.786 \pm 0.008$ |
| 5.00 | $0.769 \pm 0.111$ | $0.750 \pm 0.003$ | $0.772 \pm 0.083$ |
| 10.00 | $0.752 \pm 0.003$ | $0.770 \pm 0.111$ | $0.802 \pm 0.064$ |
| 20.00 | $0.697 \pm 0.098$ | $0.733 \pm 0.046$ | $0.799 \pm 0.064$ |
| 50.00 | $0.733 \pm 0.049$ | $0.733 \pm 0.046$ | $0.798 \pm 0.077$ |
| 100.00 | $0.750 \pm 0.004$ | $0.698 \pm 0.064$ | $0.803 \pm 0.069$ |

Table 15: Imputation accuracy for latent predicates $X_3, X_4, X_5$ under different constant temperature values $\mu$ (with fixed $\tau = 0.1$ over 20 random seeds).

| $\mu$ of Eq. 4 | Imputation Acc. X3 | Imputation Acc. X4 | Imputation Acc. X5 |
|---|---|---|---|
| 0.1 | $1.000 \pm 0.000$ | $1.000 \pm 0.000$ | $0.218 \pm 0.002$ |
| 0.2 | $1.000 \pm 0.000$ | $1.000 \pm 0.000$ | $0.218 \pm 0.002$ |
| 0.5 | $1.000 \pm 0.000$ | $1.000 \pm 0.000$ | $0.218 \pm 0.002$ |
| 1 | $1.000 \pm 0.000$ | $1.000 \pm 0.000$ | $0.217 \pm 0.002$ |
| 2 | $1.000 \pm 0.000$ | $1.000 \pm 0.000$ | $0.870 \pm 0.117$ |
| 3 | $1.000 \pm 0.000$ | $1.000 \pm 0.000$ | $0.942 \pm 0.056$ |
| 4 | $1.000 \pm 0.000$ | $1.000 \pm 0.000$ | $0.866 \pm 0.057$ |
| 5 | $0.965 \pm 0.093$ | $1.000 \pm 0.000$ | $0.894 \pm 0.056$ |
| 10 | $0.965 \pm 0.093$ | $1.000 \pm 0.000$ | $0.915 \pm 0.040$ |
| 15 | $0.965 \pm 0.092$ | $1.000 \pm 0.000$ | $0.899 \pm 0.072$ |
| 20 | $0.966 \pm 0.091$ | $1.000 \pm 0.000$ | $0.928 \pm 0.057$ |
| 25 | $0.965 \pm 0.092$ | $1.000 \pm 0.000$ | $0.886 \pm 0.060$ |
| 30 | $1.000 \pm 0.000$ | $1.000 \pm 0.000$ | $0.899 \pm 0.088$ |
| 35 | $0.965 \pm 0.093$ | $1.000 \pm 0.000$ | $0.880 \pm 0.076$ |
| 40 | $1.000 \pm 0.000$ | $1.000 \pm 0.000$ | $0.889 \pm 0.075$ |
| 45 | $0.964 \pm 0.094$ | $0.965 \pm 0.093$ | $0.909 \pm 0.051$ |
| 50 | $1.000 \pm 0.000$ | $1.000 \pm 0.000$ | $0.917 \pm 0.064$ |
| 100 | $0.965 \pm 0.094$ | $1.000 \pm 0.000$ | $0.916 \pm 0.064$ |
| 200 | $1.000 \pm 0.000$ | $1.000 \pm 0.000$ | $0.908 \pm 0.070$ |

**Soft-AND ($\tau$).** Table 14 shows that the model maintains high accuracy when $\tau$ is small (e.g., $\tau \in [0.01, 0.20]$). This is expected because as $\tau \to 0$, Softmin approximates the hard $\min$ logic required for strict conjunctions. Performance degrades only when $\tau$ becomes too large ($\tau \geq 1.0$), where the operator becomes too "soft" to capture the decisive logical boundaries. Thus, a small constant temperature (e.g., $\tau = 0.1$) is a safe and effective default.

**Soft-OR ($\mu$).** Table 15 reports the imputation accuracy under varying constant $\mu$ values. The results demonstrate that our model is highly robust to $\mu$: it achieves near-perfect accuracy for all latent predicates ($X_3, X_4, X_5$) across a wide range, specifically for $\mu \geq 3$. This aligns with the theoretical

property that as $\mu \to \infty$, the LogSumExp function approximates the hard $\max$ operator. In practice, any sufficiently large provides a strong gradient signal for discrimination while maintaining differentiability.

**Conclusion. Our framework does not rely on careful hyperparameter tuning.** A moderate to large $\mu$ for Soft-OR and a small $\tau$ for Soft-AND consistently yield optimal results. Thus, we use $\tau = 0.1$ and $\mu = 10$ as temperature parameters for all our experiments. Furthermore, complex scheduling strategies like cosine annealing can be employed if constant temperatures are not good enough.

# I SENSITIVITY ANALYSIS WITH LABEL NOISE AND MISSING RATIO

## I.1 ROBUSTNESS ANALYSIS WITH LABEL NOISE

To assess the robustness of our framework against data inconsistencies and imperfect logical dependencies, we conducted experiments by injecting label noise into the latent predicates.

Specifically, we first generate the ground-truth latent predicates $X_3, X_4, X_5$ following the perfect logical rules (e.g., $X_3 = X_0 \land X_1$). Then, we introduce stochasticity by flipping the binary labels of these latent predicates with a probability $p_{noise} \in \{0.0, 0.1, 0.2, 0.3, 0.4, 0.5\}$. **This setup simulates real-world scenarios where logical rules may have exceptions or where the observed data contains errors, directly challenging the model's ability to distill consistent symbolic rules from noisy supervision.** Tables 16 and 17 present the learned rule structures and their corresponding imputation accuracies under varying noise ratios. We use Figure 3 (b) as the representative example with an observation ratio of 0.3 and sample sizes of 20,000.

In the noise-free setting ($p_{noise} = 0.0$), our model perfectly recovers the ground-truth rules for the simpler conjunctive predicates $X_3$ and $X_4$ (with rule accuracy 1.00), achieving perfect imputation accuracy (1.000). For the more complex disjunctive rule $X_5$, the model achieves a rule accuracy of 0.50 and an imputation accuracy of 0.955 after fine-tuning, indicating that while the exact ground-truth structure is harder to isolate, the learned approximations maintain strong predictive performance.

Remarkably, the model demonstrates strong robustness at low-to-moderate noise levels ($p_{noise} \leq 0.3$). At $p_{noise} = 0.1$ and $0.2$, the ground-truth rules (underlined in the table) for $X_3$ and $X_4$ are perfectly recovered (rule accuracy 1.00) with near-perfect imputation accuracies; for the complex multi-hop rules of $X_5$, the ground-truth rules frequently emerge as the dominant learned structures (with rule accuracy above 0.5). Even at $p_{noise} = 0.3$, the model maintains high rule accuracy (0.85) for both $X_3$ and $X_4$, with imputation accuracies above 0.95; for $X_5$, the rule accuracy decreases to 0.2 at $p_{noise} = 0.3$, but the imputation accuracy remains at 0.828, suggesting that **the model learns valid approximations (e.g., capturing one correct disjunctive branch) that preserve predictive power.**

As noise increases beyond 0.3, the performance degrades more significantly. At $p_{noise} = 0.4$, rule accuracies drop to 0.85 and 0.6 for $X_3$ and $X_4$ respectively, while $X_5$ fails to recover the correct structure (rule accuracy 0.00). At $p_{noise} = 0.5$, the model struggles to learn meaningful rules, with rule accuracies in $[0.0, 0.1]$ for all predicates. However, the imputation accuracies remain above 0.70 even at these high noise levels, indicating that the learned approximations, while not perfectly matching the ground-truth rules, still provide useful predictive signals.

The imputation accuracy degrades gracefully as noise increases, rather than collapsing abruptly, indicating that **the soft-logic relaxation effectively prevents the model from overfitting to noise**, allowing it to capture the dominant logical signals within the data. The fine-tuning step for $X_5$ consistently improves imputation accuracy across all noise levels, demonstrating the effectiveness of the iterative refinement process.

## I.2 MISSING RATIO

In three synthetic datasets, we have varied the missing ratio in $\{0.7, 0.8, 0.9\}$ in the above results.

In real-world datasets, to assess the model's robustness under different levels of data scarcity, we evaluated its performance on the SPECT and Heart Disease dataset while varying the observation ratio from 0.3 to 0.9 (i.e. missing ratio from 0.1 to 0.7).

As shown in Tables 18 and 19, the model's accuracy remains acceptable and improves consistently as more data becomes available. Notably, in SPECT, even with only 30% of the data observed (a

Table 16: Impact of label noise on rule learning and missing value imputation performance. Results are over 20 random seeds.

| Noise Ratio | Avg. Imputation Accu. (Before Fine-tune) | Avg. Imputation Accu. (After Fine-tune) | Train Loss (Before Fine-tune) | Train Loss (After Fine-tune) |
|---|---|---|---|---|
| 0.0 | $X_3 : 1.000 \pm 0.000$ | / | $X_3 : 0.001 \pm 0.000$ | / |
|  | $X_4 : 1.000 \pm 0.000$ | / | $X_4 : 0.001 \pm 0.000$ | / |
|  | $X_5 : 0.907 \pm 0.050$ | $X_5 : 0.955 \pm 0.049$ | $X_5 : 0.089 \pm 0.035$ | $X_5 : 0.067 \pm 0.031$ |
| 0.1 | $X_3 : 1.000 \pm 0.000$ | / | $X_3 : 0.098 \pm 0.003$ | / |
|  | $X_4 : 1.000 \pm 0.000$ | / | $X_4 : 0.099 \pm 0.004$ | / |
|  | $X_5 : 0.948 \pm 0.042$ | $X_5 : 0.946 \pm 0.038$ | $X_5 : 0.168 \pm 0.026$ | $X_5 : 0.123 \pm 0.021$ |
| 0.2 | $X_3 : 0.975 \pm 0.076$ | / | $X_3 : 0.193 \pm 0.005$ | / |
|  | $X_4 : 0.987 \pm 0.057$ | / | $X_4 : 0.193 \pm 0.007$ | / |
|  | $X_5 : 0.894 \pm 0.050$ | $X_5 : 0.902 \pm 0.046$ | $X_5 : 0.260 \pm 0.008$ | $X_5 : 0.204 \pm 0.012$ |
| 0.3 | $X_3 : 0.950 \pm 0.103$ | / | $X_3 : 0.282 \pm 0.006$ | / |
|  | $X_4 : 0.987 \pm 0.056$ | / | $X_4 : 0.282 \pm 0.008$ | / |
|  | $X_5 : 0.822 \pm 0.046$ | $X_5 : 0.824 \pm 0.066$ | $X_5 : 0.320 \pm 0.006$ | $X_5 : 0.266 \pm 0.006$ |
| 0.4 | $X_3 : 0.863 \pm 0.127$ | / | $X_3 : 0.360 \pm 0.009$ | / |
|  | $X_4 : 0.862 \pm 0.128$ | / | $X_4 : 0.357 \pm 0.008$ | / |
|  | $X_5 : 0.792 \pm 0.078$ | $X_5 : 0.786 \pm 0.065$ | $X_5 : 0.370 \pm 0.006$ | $X_5 : 0.311 \pm 0.006$ |
| 0.5 | $X_3 : 0.745 \pm 0.085$ | / | $X_3 : 0.418 \pm 0.007$ | / |
|  | $X_4 : 0.725 \pm 0.077$ | / | $X_4 : 0.421 \pm 0.007$ | / |
|  | $X_5 : 0.761 \pm 0.057$ | $X_5 : 0.767 \pm 0.072$ | $X_5 : 0.421 \pm 0.007$ | $X_5 : 0.349 \pm 0.007$ |

Table 17: Learned rule structures under label noise. Ground truth rules are indicated with underlines. Results are over 20 random seeds.

| Noise Ratio | Learned Rule Structure | Rule Accu. |
|---|---|---|
| 0.0 | $X_3 : \underline{X_0 \wedge X_1}$ | $X_3 : 1.00$ |
|  | $X_4 : \underline{X_2 \wedge X_7}$ | $X_4 : 1.00$ |
|  | $X_5 : (\underline{\overline{X_0} \wedge \overline{X_4}}) \vee (X_3 \wedge X_6), (X_3 \wedge X_4) \vee (X_3 \wedge X_6), (X_0 \wedge X_4) \vee (X_0 \wedge X_4), (X_0 \wedge X_4) \vee (X_1 \wedge X_6), (X_1 \wedge X_3) \vee (X_3 \wedge X_6)$ | $X_5 : 0.50$ |
| 0.1 | $X_3 : \underline{X_0 \wedge X_1}$ | $X_3 : 1.00$ |
|  | $X_4 : \underline{X_2 \wedge X_7}$ | $X_4 : 1.00$ |
|  | $X_5 : (\underline{\overline{X_0} \wedge \overline{X_4}}) \vee (X_3 \wedge X_6), (X_3 \wedge X_4) \vee (X_3 \wedge X_6), (X_0 \wedge X_4) \vee (X_0 \wedge X_4), (X_0 \wedge X_4) \vee (X_2 \wedge X_3), (X_0 \wedge X_2) \vee (X_3 \wedge X_6)$ | $X_5 : 0.55$ |
| 0.2 | $X_3 : \underline{X_0 \wedge X_1}$ | $X_3 : 1.00$ |
|  | $X_4 : \underline{X_2 \wedge X_7}$ | $X_4 : 1.00$ |
|  | $X_5 : (\underline{\overline{X_0} \wedge \overline{X_4}}) \vee (X_3 \wedge X_6), (X_0 \wedge X_4) \vee (X_3 \wedge X_4), (X_0 \wedge X_1) \vee (X_3 \wedge X_6), (X_0 \wedge X_4) \vee (X_0 \wedge X_7), (X_0 \wedge X_4) \vee (X_0 \wedge X_6)$ | $X_5 : 0.60$ |
| 0.3 | $X_3 : \underline{X_0 \wedge X_1}, X_1 \wedge X_2, X_1 \wedge X_1, X_1 \wedge X_7$ | $X_3 : 0.85$ |
|  | $X_4 : \underline{X_2 \wedge X_7}, X_2 \wedge X_6, X_2 \wedge X_2$ | $X_4 : 0.85$ |
|  | $X_5 : (\underline{\overline{X_0} \wedge \overline{X_4}}) \vee (X_3 \wedge X_6), (X_0 \wedge X_7) \vee (X_3 \wedge X_6), (X_0 \wedge X_3) \vee (X_0 \wedge X_4), (X_3 \wedge X_6) \vee (X_4 \wedge X_7), (X_0 \wedge X_1) \vee (X_0 \wedge X_6)$ | $X_5 : 0.20$ |
| 0.4 | $X_3 : \underline{X_0 \wedge X_1}, X_1 \wedge X_1, X_1 \wedge X_2, X_0 \wedge X_6, X_0 \wedge X_0$ | $X_3 : 0.60$ |
|  | $X_4 : \underline{X_2 \wedge X_7}, X_7 \wedge X_7, X_0 \wedge X_2, X_0 \wedge X_0, X_2 \wedge X_6$ | $X_4 : 0.60$ |
|  | $X_5 : (\underline{\overline{X_0} \wedge \overline{X_2}}) \vee (X_4 \wedge X_6), (X_0 \wedge X_6) \vee (X_3 \wedge X_3), (X_0 \wedge X_2) \vee (X_2 \wedge X_3), (X_0 \wedge X_6) \vee (X_1 \wedge X_7), (X_1 \wedge X_4) \vee (X_4 \wedge X_4)$ | $X_5 : 0.00$ |
| 0.5 | $X_3 : X_1 \wedge X_7, X_2 \wedge X_2, X_6 \wedge X_6, X_0 \wedge X_2, \underline{X_0 \wedge X_1}$ | $X_3 : 0.10$ |
|  | $X_4 : X_6 \wedge X_6, X_0 \wedge X_6, X_0 \wedge X_7, X_6 \wedge X_7, \overline{X_1 \wedge X_2}$ | $X_4 : 0.05$ |
|  | $X_5 : (X_0 \wedge X_3) \vee (X_1 \wedge X_6), (X_0 \wedge X_2) \vee (X_2 \wedge X_7), (X_0 \wedge X_0) \vee (X_2 \wedge X_7), (X_0 \wedge X_3) \vee (X_4 \wedge X_7), (X_2 \wedge X_7) \vee (X_6 \wedge X_7)$ | $X_5 : 0.00$ |

70% missing ratio), the model maintains a high F1 score of 0.751, demonstrating its capability to learn meaningful diagnostic rules from highly incomplete datasets.

For the Birds Dataset, we fix the observation ratio as 0.1 (i.e. 90% missingness) and show results over different number of training samples. Results in Table 20 show that a few hundred samples are sufficient for the model to converge to the correct logical truth.

Table 18: Performance on the SPECT dataset with varying observation ratios.

| Observation Ratio | Imputation Acc. | Diagnosis Acc. | Diagnosis F1 |
|---|---|---|---|
| 0.3 | 0.501 | 0.679 | 0.751 |
| 0.5 | 0.630 | 0.765 | 0.808 |
| 0.7 | 0.763 | 0.920 | 0.958 |
| 0.9 | 0.791 | 0.929 | 0.960 |

Table 19: Imputation accuracy for Heart Disease under different observation ratios.

| Observation Ratio | Overall | trestbps | chol | thalach | oldpeak |
|---|---|---|---|---|---|
| 0.3 | 0.6444 | 0.7129 | 0.6304 | 0.7393 | 0.4950 |
| 0.5 | 0.7434 | 0.8053 | 0.7558 | 0.7954 | 0.6172 |
| 0.7 | 0.8432 | 0.8647 | 0.8482 | 0.9043 | 0.7558 |
| 0.9 | 0.9439 | 0.9439 | 0.9307 | 0.9769 | 0.9241 |

Table 20: Impact of training sample size on the imputation accuracy of latent predicates (`abnormal`, `fly`) in the Birds domain. Results are reported as mean $\pm$ std over 10 random seeds, evaluated with 10% observation probability.

| # Samples | Acc. Abnormal Bird | Acc. Can Fly |
|---|---|---|
| 100 | $0.896 \pm 0.058$ | $0.845 \pm 0.148$ |
| 500 | $0.976 \pm 0.054$ | $0.928 \pm 0.066$ |
| 1000 | $0.951 \pm 0.067$ | $0.928 \pm 0.066$ |
| 1500 | $0.949 \pm 0.070$ | $0.952 \pm 0.066$ |
| 2000 | $0.951 \pm 0.067$ | $0.928 \pm 0.066$ |

## J  ADDITIONAL SYNTHETIC EXPERIMENTS RESULTS

### J.1  MAIN RESULTS SUPPLEMENT OF EXAMPLE (B) OF FIGURE 3.

**Dataset Generation.** The base variables $\{X_0, X_1, X_2, X_6, X_7\}$ are independently generated from a Bernoulli distribution, each with $p = 0.5$. Subsequently, the values for $\{X_3, X_4, X_5\}$ are deterministically derived using the ground truth logical rules depicted in Figure 3. Specifically, these rules are:

$$X_3 \leftarrow X_0 \wedge X_1$$
$$X_4 \leftarrow X_2 \wedge X_7$$
$$X_5 \leftarrow (X_3 \wedge X_6) \vee (X_4 \wedge X_0)$$

Finally, to introduce missing information, a portion of the values for $X_3, X_4$, and $X_5$ are randomly masked. These masked variables become the targets for imputation. In our experiments, we vary the level of missingness, applying masking probabilities of 70%, 80%, and 90% to these target variables (corresponding to observation ratios of 30%, 20%, and 10%, respectively).

**Main Results.** As demonstrated in a previous case study (Table 3, which shows three runs using the same seed but different internal rule optimization orders), variations in the rule optimization sequence within a single seed can affect training efficiency. We thus show the coordinate descent training progress under a different random optimization order from Figure 4 here in Figure 8. In this run, the optimization order is $[X_5, X_4, X_3]$ for cycle 1 and $[X_4, X_5, X_3]$ for cycle 2. Given such different learning trajectories, our model still discovers the correct rules successfully.

Furthermore, random initialization across different seeds can lead to the discovery of varied rule sets, and occasionally, the model might converge to a local optimum. However, as the analysis of convergence before, performing multiple runs with different initializations enhances the probability of identifying the global optimal solution. Our results show that the model identifies global optima in more than half of the 30 random-seed runs (Tables 21 and 22).

**Learning Efficiency.** As the observation ratio decreases, the guidance signal becomes less informative, reducing both rule structure recovery and missing value imputation. We further investigated the impact of dataset sample size, varying it from 500 to 20,000 samples. As shown in Figure 9, the most efficient setting we can recover all the AND/OR rules for $X_3$, $X_4$, and $X_5$ is to use an observation ratio of 0.1 and a dataset of 1,000 samples.

### J.2  RESULTS OF EXAMPLE (A) OF FIGURE 3

**Dataset Generation.** The base variables $\{X_0, X_1, X_2, X_6\}$ are independently generated from a Bernoulli distribution, each with $p = 0.5$. Subsequently, the values for $\{X_3, X_4, X_5\}$ are deterministically derived using the ground truth logical rules depicted in Figure 3. Specifically, these rules

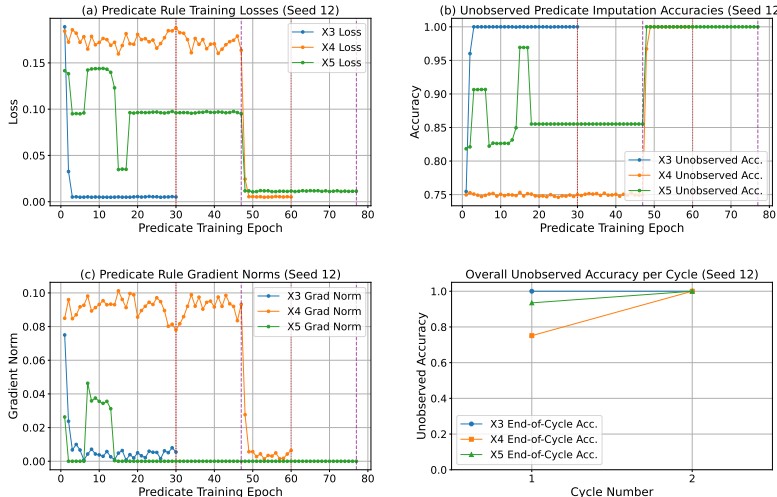

Figure 8: Training dynamics for a representative run (Obs. Ratio = 0.3) of Figure 3 (b). The optimization order: [ $X_5, X_4, X_3$ ] for Cycle 1; [$X_4, X_5, X_3$ ] for Cycle 2. Subplots display: (a) training losses, (b) unobserved imputation accuracies, and (c) gradient norms for rule embeddings; (d) overall imputation accuracies each cycle. Red dashed lines indicate the conclusion of training blocks for $X_3$ or $X_4$ (each allocated 30 epochs when active within a cycle). Purple dashed lines delineate training phases for $X_5$ (Rule 1, Rule 2, and Fine-tune); the epoch count for these $X_5$ phases can vary per cycle due to the dynamic nature of the hard covering mechanism. Correct rule structures were learned for $X_3$ by the end of Cycle 1, and for $X_4$ and $X_5$ by the end of Cycle 2.

Table 21: Summary of synthetic data experiment results for example (b) of the Figure 3. Evaluated on 20,000 samples and results are averaged over 30 random seeds.

| Obs. Ratio | Avg. Imputation Accu. (Before Fine-tune) | Avg. Imputation Accu. (After Fine-tune) | Train Loss (Before Fine-tune) | Train Loss (After Fine-tune) |
|---|---|---|---|---|
| 0.3 | $X_3 : 1.000 \pm 0.000$ | / | $X_3 : 0.001 \pm 0.000$ | / |
|  | $X_4 : 1.000 \pm 0.000$ | / | $X_4 : 0.001 \pm 0.000$ | / |
|  | $X_5 : 0.889 \pm 0.071$ | $X_5 : 0.956 \pm 0.052$ | $X_5 : 0.090 \pm 0.046$ | $X_5 : 0.060 \pm 0.037$ |
| 0.2 | $X_3 : 1.000 \pm 0.000$ | / | $X_3 : 0.001 \pm 0.000$ | / |
|  | $X_4 : 1.000 \pm 0.000$ | / | $X_4 : 0.001 \pm 0.000$ | / |
|  | $X_5 : 0.893 \pm 0.072$ | $X_5 : 0.954 \pm 0.053$ | $X_5 : 0.087 \pm 0.045$ | $X_5 : 0.060 \pm 0.039$ |
| 0.1 | $X_3 : 1.000 \pm 0.000$ | / | $X_3 : 0.001 \pm 0.000$ | / |
|  | $X_4 : 1.000 \pm 0.000$ | / | $X_4 : 0.001 \pm 0.000$ | / |
|  | $X_5 : 0.879 \pm 0.064$ | $X_5 : 0.950 \pm 0.050$ | $X_5 : 0.090 \pm 0.052$ | $X_5 : 0.066 \pm 0.039$ |

are:

$$X_3 \leftarrow X_0 \wedge X_1$$
$$X_4 \leftarrow X_2 \wedge X_3$$
$$X_5 \leftarrow X_4 \wedge X_6$$

Finally, to introduce missing information, a portion of the values for $X_3, X_4$, and $X_5$ are randomly masked. These masked variables become the targets for imputation. In our experiments, we vary the level of missingness, applying masking probabilities of 70%, 80%, and 90% to these target variables (corresponding to observation ratios of 30%, 20%, and 10%, respectively).

**Main Results.** We show the coordinate descent training progress under different random optimization order. Figure 10 demonstrates the convergence in two cycles, while Figure 11 requires three cycles to complete training.

We summarize the results for example (a) of the Figure 3 in Tables 23 and 24, which demonstrate both the effectiveness of our rule discovery approach and the precision of missing variables imputa-

Table 22: Summary of learned rule structures and accuracy for example (b) of Figure 3. Each observation ratio was evaluated using 20,000 samples, with results averaged over 30 random seeds. We present the top 3 learned rule structures in order of discovery accuracy. Rule accuracy indicates the percentage of 30 runs in which a rule was learned completely correctly.

| Obs.Ratio | Learned Rule Structure | Rule Accu. |
|---|---|---|
| 0.3 | $X_3 : \underline{X_0 \wedge X_1}$ 
 $X_4 : \underline{X_2 \wedge X_7}$ 
 $X_5 : \underline{(X_0 \wedge X_4) \vee (X_3 \wedge X_6)}, \overline{(X_3 \wedge X_4) \vee (X_3 \wedge X_6)}, (X_3 \wedge X_6)$ | $X_3 : 1.00$ 
 $X_4 : 1.00$ 
 $X_5 : 0.53$ |
| 0.2 | $X_3 : \underline{X_0 \wedge X_1}$ 
 $X_4 : \underline{X_2 \wedge X_7}$ 
 $X_5 : \underline{(X_0 \wedge X_4) \vee (X_3 \wedge X_6)}, \overline{(X_3 \wedge X_6)}, (X_0 \wedge X_4)$ | $X_3 : 1.00$ 
 $X_4 : 1.00$ 
 $X_5 : 0.53$ |
| 0.1 | $X_3 : \underline{X_0 \wedge X_1}$ 
 $X_4 : \underline{X_2 \wedge X_7}$ 
 $X_5 : \underline{(X_0 \wedge X_4) \vee (X_3 \wedge X_6)}, (X_3 \wedge \overline{X_4}) \vee (\overline{X_3} \wedge X_6), (X_0 \wedge X_4) \vee (X_3 \wedge X_4)$ | $X_3 : 1.00$ 
 $X_4 : 1.00$ 
 $X_5 : 0.43$ |

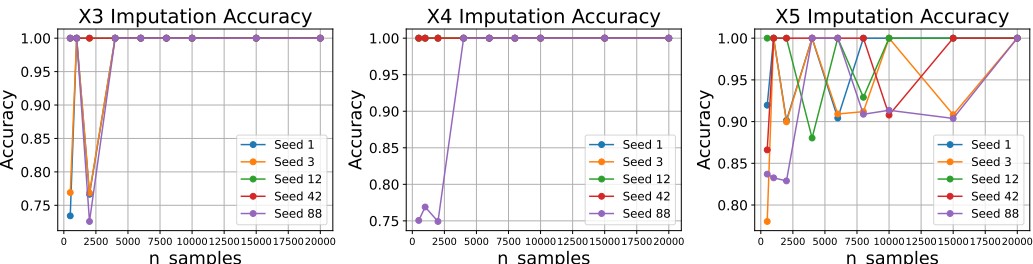

Figure 9: Imputation accuracy versus dataset sample size for Figure 3 (b). For these experiments, 10% of the data was observed (i.e., a 90% missing ratio) for predicates in $X_3$, $X_4$, and $X_5$.

tion. Our analysis reveals that learning the multi-step chain structure presents significant challenges, primarily because the algorithm uses inferred predicate values $v^t$ from previous steps to update the current values by Eq. 2. This creates a dependency chain where suboptimal rule embeddings learned at earlier optimization steps can propagate errors to subsequent steps, potentially degrading overall performance. Despite these challenges, our model successfully identifies the correct rules in the majority of experimental runs. This robustness indicates that with multiple random initializations, the algorithm reliably converges to the optimal rule structures like the results from Figures 10 and 11, which effectively overcome the inherent difficulties of sequential dependency learning in chain-like logical structures.

Table 23: Summary of synthetic data experiment results for example (a) of the Figure 3. Each observation ratio is evaluated using 20,000 samples and results are averaged over 30 random seeds. No fine-tune phase since we assume no disjunctive rules.

| Obs. Ratio | Avg. Imputation Accu. | Train Loss |
|---|---|---|
| 0.3 | $X_3 : 0.88 \pm 0.13$ 
 $X_4 : 0.92 \pm 0.06$ 
 $X_5 : 0.95 \pm 0.03$ | $X_3 : 0.06 \pm 0.07$ 
 $X_4 : 0.04 \pm 0.04$ 
 $X_5 : 0.03 \pm 0.02$ |
| 0.2 | $X_3 : 0.94 \pm 0.11$ 
 $X_4 : 0.95 \pm 0.06$ 
 $X_5 : 0.96 \pm 0.03$ | $X_3 : 0.03 \pm 0.06$ 
 $X_4 : 0.03 \pm 0.04$ 
 $X_5 : 0.03 \pm 0.02$ |
| 0.1 | $X_3 : 0.94 \pm 0.11$ 
 $X_4 : 0.94 \pm 0.06$ 
 $X_5 : 0.96 \pm 0.03$ | $X_3 : 0.03 \pm 0.06$ 
 $X_4 : 0.03 \pm 0.04$ 
 $X_5 : 0.03 \pm 0.02$ |

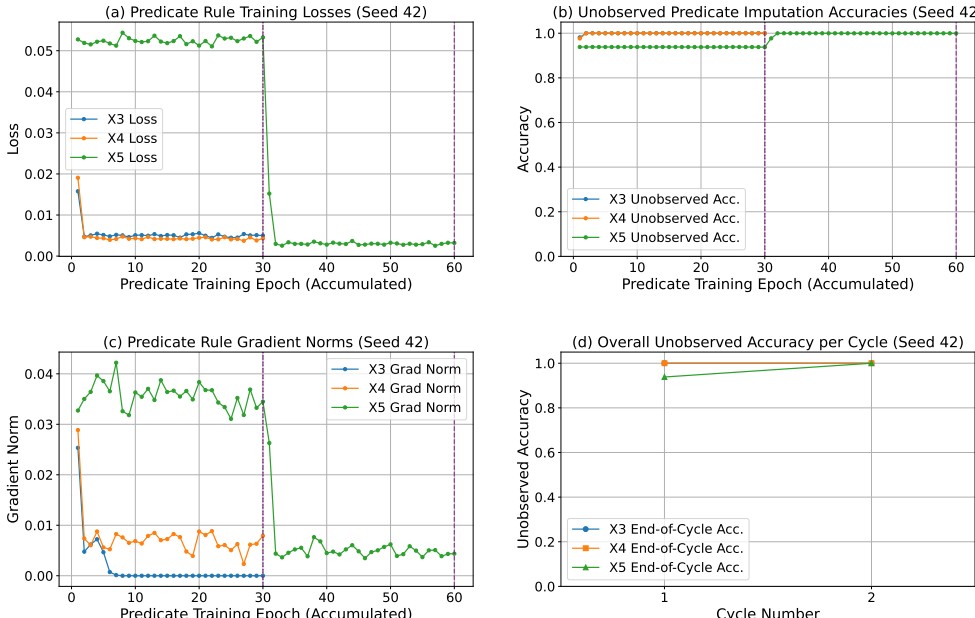

Figure 10: Training dynamics for a representative run (Observation Ratio = 0.2) of Figure 3 (a). Subplots display: (a) training losses, (b) unobserved imputation accuracies, and (c) gradient norms for rule embeddings; (d) overall imputation accuracies each cycle. Purple dashed lines indicate the conclusion of training blocks for one cycle (each allocated 30 epochs). The rule embedding optimization order: $[X_5, X_3, X_4]$ for Cycle 1; $[X_5, X_4, X_3]$ for Cycle 2. Correct rule structures were learned for $X_3$ and $X_4$ by the end of Cycle 1, for $X_5$ by the end of Cycle 2. The learned rules: $X_3 \leftarrow X_0 \wedge X_1, X_4 \leftarrow X_2 \wedge X_3, X_5 \leftarrow X_4 \wedge X_6$.

Table 24: Summary of learned rule structures and accuracy for example (a) of Figure 3. Each observation ratio is evaluated using 20,000 samples, with results averaged over 30 random seeds. We present the top 3 learned rule structures in order of discovery accuracy. The rules that are truth rules are indicated by underline. Rule accuracy indicates the percentage of 30 runs in which a rule was learned completely correctly.

| Obs. Ratio | Learned Rule Structure | Rule Accuracy |
|:---:|:---:|:---:|
| 0.3 | $X_3 : \underline{X_0 \wedge X_1}, X_1 \wedge X_4, X_0 \wedge X_4$ 
 $X_4 : \underline{X_2 \wedge X_3}, X_0 \wedge X_1, X_1 \wedge X_2$ 
 $X_5 : \underline{X_4 \wedge X_6}, X_3 \wedge X_6, X_0 \wedge X_4$ | $X_3 : 0.53$ 
 $X_4 : 0.40$ 
 $X_5 : 0.23$ |
| 0.2 | $X_3 : \underline{X_0 \wedge X_1}, X_1 \wedge X_5, X_1 \wedge X_4$ 
 $X_4 : \underline{X_2 \wedge X_3}, X_3 \wedge X_5, X_1 \wedge X_2$ 
 $X_5 : \underline{X_4 \wedge X_6}, X_1 \wedge X_4, X_3 \wedge X_6$ | $X_3 : 0.77$ 
 $X_4 : 0.60$ 
 $X_5 : 0.37$ |
| 0.1 | $X_3 : \underline{X_0 \wedge X_1}, X_1 \wedge X_4, X_0 \wedge X_4$ 
 $X_4 : \underline{X_2 \wedge X_3}, X_0 \wedge X_1, X_0 \wedge X_2$ 
 $X_5 : \underline{X_4 \wedge X_6}, X_0 \wedge X_4, X_3 \wedge X_4$ | $X_3 : 0.77$ 
 $X_4 : 0.53$ 
 $X_5 : 0.33$ |

J.3 RESULTS OF EXAMPLE (C) OF FIGURE 3

**Dataset Generation.** The base variables $\{X_0, X_1, X_2, X_6, X_7\}$ are independently generated from a Bernoulli distribution, each with $p = 0.5$. Subsequently, the values for $\{X_3, X_4, X_5, X_8\}$ are deterministically derived using the ground truth logical rules depicted in Figure 3. Specifically,

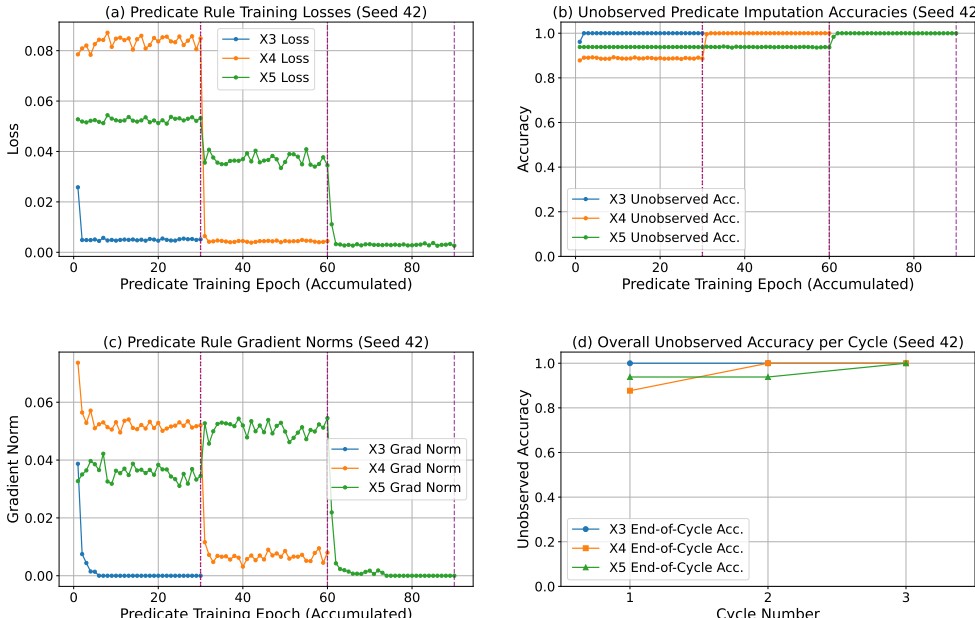

Figure 11: Training dynamics for a representative run (Observation Ratio = 0.2) of Figure 3 (a). Subplots display: (a) training losses, (b) unobserved imputation accuracies, and (c) gradient norms for rule embeddings; (d) overall imputation accuracies each cycle. Purple dashed lines indicate the conclusion of training blocks for one cycle (each allocated 30 epochs). The rule embedding optimization order: [ $X_5, X_4, X_3$ ] for Cycle 1,2; [$X_3, X_5, X_4$ ] for Cycle 3. Correct rule structures were learned for $X_3$ by the end of Cycle 1, for $X_4$ by the end of Cycle 2, and for $X_5$ by the end of Cycle 3. The learned rules: $X_3 \leftarrow X_0 \wedge X_1, X_4 \leftarrow X_2 \wedge X_3, X_5 \leftarrow X_4 \wedge X_6$.

these rules are:

$$X_3 \leftarrow X_0 \wedge X_1$$
$$X_4 \leftarrow X_2 \wedge X_7$$
$$X_8 \leftarrow X_4 \wedge X_0$$
$$X_5 \leftarrow (X_3 \wedge X_6) \vee (X_8) \vee (X_6 \wedge X_7)$$

Finally, to introduce missing information, a portion of the values for $X_3, X_4, X_8$ and $X_5$ are randomly masked. These masked variables become the targets for imputation. In our experiments, we vary the level of missingness, applying masking probabilities of 70%, 80%, and 90% to these target variables (corresponding to observation ratios of 30%, 20%, and 10%, respectively).

**Main Results.** We summarize the results for example (c) of the Figure 3 in the Tables 25 and 26, showcasing the effectiveness of rule discovery and the precision of missing variables imputation. We have random coordinate descent training order for rule optimization.

This task is more challenging due to the chain-like structure of the disjunctive rules, particularly with three clauses for $X_5$, resulting in lower learning accuracy than in example (b). Nonetheless, our method achieves the highest rule discovery accuracy for the ground-truth rules while maintaining acceptable imputation accuracy. For the most difficult prediction task ($X_5$) and the chain-derived predicate $X_8$, we obtain about 90% accuracy across all three observation ratios. Other predicate predictions reach almost perfect accuracy. For the learned rules in Table 26, we can find most body predicates are correct even for the complex three-clause rules governing $X_5$, which include the chain-derived predicate $X_8$. We also show the loss plot for one run in Figure 12.

### J.4 HYPER-PARAMETERS SETTING AND COMPUTING RESOURCE

Our model operates efficiently in a CPU environment utilizing the PyTorch library. The hyperparameters are configured as follows:

- Rule Embedding and Fine-tuning Optimizer: Adam, learning rate: 0.01.
- Temperature of softmin and softmax: 0.1 (for Eq. 3) and 10.0 for (Eq. 4).

Table 25: Summary of synthetic data experiment results for example (c) of the Figure 3. Each observation ratio is evaluated using 20,000 samples and results are averaged over 30 random seeds.

| Obs. Ratio | Avg. Imputation Accu. (Before Fine-tune) | Avg. Imputation Accu. (After Fine-tune) | Train Loss (Before Fine-tune) | Train Loss (After Fine-tune) |
|---|---|---|---|---|
| 0.3 | $X_3 : 0.98 \pm 0.06$ | / | $X_3 : 0.013 \pm 0.045$ | / |
| | $X_4 : 1.00 \pm 0.00$ | / | $X_4 : 0.001 \pm 0.000$ | / |
| | $X_5 : 0.89 \pm 0.08$ | $X_5 : 0.91 \pm 0.08$ | $X_5 : 0.117 \pm 0.077$ | $X_5 : 0.077 \pm 0.055$ |
| | $X_8 : 0.93 \pm 0.05$ | / | $X_8 : 0.053 \pm 0.037$ | / |
| 0.2 | $X_3 : 0.98 \pm 0.06$ | / | $X_3 : 0.013 \pm 0.046$ | / |
| | $X_4 : 1.00 \pm 0.00$ | / | $X_4 : 0.001 \pm 0.000$ | / |
| | $X_5 : 0.91 \pm 0.07$ | $X_5 : 0.93 \pm 0.08$ | $X_5 : 0.109 \pm 0.084$ | $X_5 : 0.061 \pm 0.052$ |
| | $X_8 : 0.91 \pm 0.05$ | / | $X_8 : 0.068 \pm 0.035$ | / |
| 0.1 | $X_3 : 0.97 \pm 0.08$ | / | $X_3 : 0.020 \pm 0.057$ | / |
| | $X_4 : 1.00 \pm 0.00$ | / | $X_4 : 0.001 \pm 0.000$ | / |
| | $X_5 : 0.90 \pm 0.08$ | $X_5 : 0.92 \pm 0.08$ | $X_5 : 0.103 \pm 0.081$ | $X_5 : 0.065 \pm 0.055$ |
| | $X_8 : 0.91 \pm 0.06$ | / | $X_8 : 0.072 \pm 0.047$ | / |

Table 26: Summary of learned rule structures and accuracy for example (c) of Figure 3. Each observation ratio is evaluated using 20,000 samples, with results averaged over 30 random seeds. We present the top 3 learned rule structures in order of discovery accuracy. The rules that are truth rules are indicated by underline. Rule accuracy indicates the percentage of 30 runs in which a rule was learned completely correctly.

| Obs. Ratio | Learned Rule Structure | Rule Accu. |
|---|---|---|
| 0.3 | $X_3 : \underline{X_0 \wedge X_1}, X_1 \wedge X_2, X_0 \wedge X_6$ | $X_3 : 0.93$ |
| | $X_4 : \underline{X_2 \wedge X_7}$ | $X_4 : 1.00$ |
| | $X_5 : \underline{(X_3 \wedge X_6) \vee X_8 \vee (X_6 \wedge X_7)}, \overline{(X_3 \wedge X_6)} \vee (X_4 \wedge X_6) \vee (X_6 \wedge X_7),$ | $X_5 : 0.27$ |
| | $(X_3 \wedge X_4) \vee (X_3 \wedge X_6) \vee (X_4 \wedge X_6)$ | |
| | $X_8 : X_3 \wedge X_4, \underline{X_0 \wedge X_4}, X_1 \wedge X_4$ | $X_8 : 0.30$ |
| 0.2 | $X_3 : \underline{X_0 \wedge X_1}, X_0 \wedge X_2, X_0 \wedge X_6$ | $X_3 : 0.93$ |
| | $X_4 : \underline{X_2 \wedge X_7}$ | $X_4 : 1.00$ |
| | $X_5 : \underline{(X_3 \wedge X_6) \vee X_8 \vee (X_6 \wedge X_7)}, \overline{(X_3 \wedge X_6)} \vee (X_4 \wedge X_6) \vee (X_6 \wedge X_7),$ | $X_5 : 0.37$ |
| | $(X_3 \wedge X_6) \vee X_8 \vee (X_4 \wedge X_6)$ | |
| | $X_8 : X_3 \wedge X_4, X_2 \wedge X_7, \underline{X_0 \wedge X_4}$ | $X_8 : 0.17$ |
| 0.1 | $X_3 : \underline{X_0 \wedge X_1}, X_0 \wedge X_6, X_0 \wedge X_2$ | $X_3 : 0.90$ |
| | $X_4 : \underline{X_2 \wedge X_7}$ | $X_4 : 1.00$ |
| | $X_5 : \underline{(X_3 \wedge X_6) \vee X_8 \vee (X_6 \wedge X_7)}, \overline{(X_3 \wedge X_4)} \vee (X_3 \wedge X_6) \vee (X_6 \wedge X_7),$ | $X_5 : 0.30$ |
| | $(X_3 \wedge X_6) \vee (X_4 \wedge X_6) \vee (X_6 \wedge X_7)$ | |
| | $X_8 : \underline{X_0 \wedge X_4}, X_3 \wedge X_4, X_0 \wedge X_7$ | $X_8 : 0.23$ |

- "Well-explained" Threshold: 0.99 (for sequential hard covering in disjunctive rule learning).

- Batch Size: 64.

# K  ADDITIONAL REAL WORLD DATA EXPERIMENTS RESULTS

## K.1  SPECT

We ask for an expertise from cardiovascular surgery of a hospital to give us domain knowledge, and then we try to explain the learned rules. We select several meaningful rules to demonstrate.

The domain knowledge are as follows.

- R1: The anterior wall and the septum of the left ventricle are adjacent and often simultaneously affected by the Left Anterior Descending artery (LAD). If both anterior wall and septum show infarction, it strongly suggests an issue with the LAD. If both apical anterior

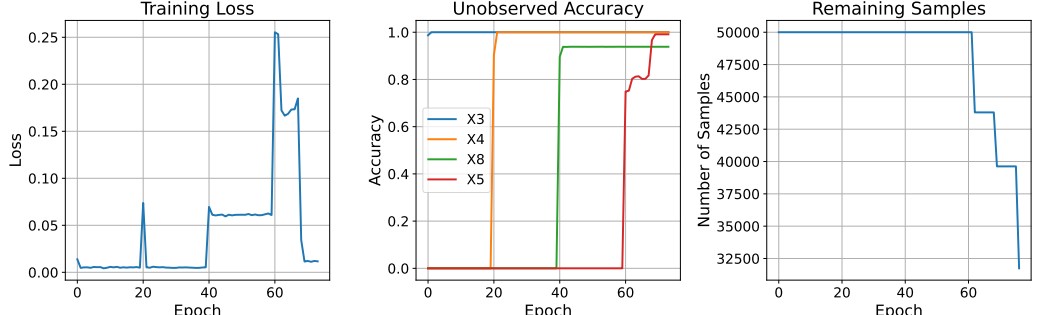

Figure 12: An example of loss and imputation accuracy during coordinate optimization (Obs. Ratio $= 0.1$, seed $= 88$, from example (c) of Figure 3). The training order is $[X_3, X_4, X_8, X_5]$. Epochs 0–19 correspond to rule learning for $X_3$; epochs 20–39 for $X_4$; epochs 40–59 for $X_8$, and epochs 60-end for $X_5$. Remaining samples identified how many samples are "well-explained" during hard covering phase. As the imputation accuracy for missing $X_5$ is 1.00, we do not go to the fine-tune phase. The learned rules: $X_3 \leftarrow X_0 \wedge X_1, X_4 \leftarrow X_2 \wedge X_7, X_5 \leftarrow (X_3 \wedge X_6) \vee X_8 \vee (X_6 \wedge X_7), X_8 \leftarrow X_3 \wedge X_4$.

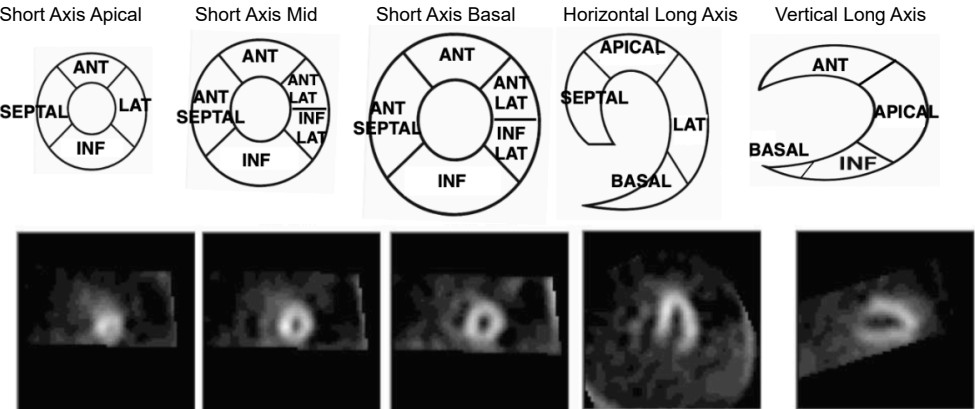

Figure 13: The five slices consists of 22 regions of interest (ROI) for SPECT Diagnosis. The slices are chosen according to the following: Three slices for short axis view-one slice near heart's apex, one in middle of the LV and one near the heart base; One slice corresponds to the center of the LV cavity for horizontal long axis view; One slice corresponds to the center of the LV cavity for vertical long axis view (Kurgan et al., 2001).

and mid-anterior show infarction, it indicates a more extensive problem within the LAD territory, affecting both the apical and mid-portions of the anterior wall.

- R2: The apical lateral wall (typically LCX territory) and the apical inferior wall (typically RCA or LCX territory) are adjacent. Infarction in both suggests a problem in this combined region.

- R3: If both apical septal and apical septal show infarction, it indicates a more extensive problem in the LAD territory, involving ischemia in multiple myocardial segments.

- R4: If apical lateral and apical lateral show infarction, it indicates a more extensive ischemic problem in the Left Circumflex artery (LCX) territory.

- R5: The apical anterior (ANT) and apical septal (SEPTAL) regions are primarily supplied by the Left Anterior Descending artery (LAD); the apical lateral (LAT) region is primarily supplied by the Left Circumflex artery (LCX); the apical inferior (INF) region is primarily supplied by the Right Coronary Artery (RCA), but can sometimes be supplied by the LCX, depending on the coronary artery dominance pattern.

Refer to Figure 13, we can give some explanations of rules learned in Table 27 based on the domain knowledge R1 to R5. For example,

- $F_5 \leftarrow F_1 \wedge F_2$: $F_1$ and $F_2$ are features from the first slice near the heart's apex, while $F_5$ is from the second slice at the middle of the left ventricle (LV). According to clinical

knowledge R1 and R5, the anterior and septal regions are primarily supplied by the Left Anterior Descending (LAD) artery. Therefore, this rule is clinically plausible: if partial diagnosis (labeled as 1) is present in both $F_1$ and $F_2$, it strongly suggests an LAD artery problem. Since $F_5$ is in the mid-anterior region, also supplied by the LAD, it has a high probability of being affected as well.

- $Diagnosis \leftarrow F_5 \wedge F_6$: From R1, we know that the anterior wall and the septum of the left ventricle are adjacent. $F_5$ and $F_6$ both from middle of the LV (left ventricular), and they are adjacent. Thus, if both these adjacent mid-ventricular regions ($F_5$ and $F_6$) show signs of infarction, it significantly increases the likelihood of an overall positive diagnosis.

Table 27: Example rules learned by NS-FCN for SPECT feature imputation and diagnosis.

| **Selected Feature Imputation Rules Learned by NS-FCN** |
| --- |
| $F_5 \leftarrow F_1 \wedge F_2$: partial diagnosis of segment 1 and 2 causes the partial diagnosis of segment 5. |
| $F_6 \leftarrow F_{11} \wedge F_{19}$ |
| $F_{13} \leftarrow F_{22} \wedge F_{12}$ |
| **Learned Diagnosis Rule Structure** |
| $Diagnosis \leftarrow (F_5 \wedge F_6) \vee (F_2 \wedge F_{11}) \vee (F_4 \wedge F_{13})$ |

As detailed in Table 28, the learned rules for diagnosing cardiac abnormalities correspond closely with established domain knowledge from cardiovascular surgery experts. For instance, the model identified that infarcts in adjacent regions like $F_1$ and $F_2$ are indicative of an issue in the Left Anterior Descending (LAD) artery territory. Furthermore, the model learned a composite rule for the final diagnosis, logically aggregating signals from multiple infarcted regions across different coronary artery territories (LAD, LCX, RCA). This ability to synthesize information from disparate features into a coherent diagnostic rule highlights the model's capacity for complex reasoning. The clinical relevance of these rules was further validated by a Large Language Model (LLM), which confirmed their consistency with expert knowledge on ischemia propagation patterns.

Table 28: Analysis of learned rules for the SPECT dataset, evaluated by human experts and LLM.

| Rules | Evaluation with Human Expert Knowledge | LLM Evaluation |
| --- | --- | --- |
| $F_6 \leftarrow F_1 \wedge F_2$ | Matches R1 & R5: $F_1$ and $F_2$ are in LAD territory. Infarction in both suggests LAD issue affecting apical and mid-anterior LV. | **Plausible:** Both regions are LAD-supplied and adjacent; mid-anterior ($F_3$) likely also affected if $F_1$ & $F_2$ show infarction. Clinically consistent. |
| $F_0 \leftarrow F_{11} \wedge F_{19}$ | Related to R2 & R4: $F_{11}$ and $F_{19}$ are adjacent. Infarction implies LCX or RCA/LCX combined territory issue. | **Valid:** Matches adjacency and vascular territory logic (LCX-lateral, RCA-inferior). Supports ischemia propagation in midventricular slices. |
| $F_{13} \leftarrow F_{22} \wedge F_{12}$ | Partial link to R3 & R5: Likely involves basal/apical septal ($F_{22}$) and adjacent basal regions. Indicates LAD or multi-segment ischemia. | **Reasonable:** Suggests ischemia spread in basal-septal regions (LAD) adjacent to basal/anterior. Fits multi-segment LAD pathology. |
| $Diagnosis \leftarrow (F_1 \wedge F_0) \vee (F_2 \wedge F_{11}) \vee (F_6 \wedge F_{13})$ | Consistent with R1 & R4. Combines LAD ($F_0$), LCX/RCA ($F_6$), and adjacent mixed regions. Multiple adjacent infarct pairs increase diagnosis likelihood. | **Strong:** Logical aggregation of adjacent infarcted regions across LAD, LCX, RCA territories. Matches expert ischemia propagation patterns. |

**Performance with varying missing ratios.** To assess the model's robustness under different levels of data scarcity, we evaluated its performance on the SPECT dataset while varying the observation ratio from 0.3 to 0.9. As shown in Table 18, the model's accuracy remains strong and improves

consistently as more data becomes available. Notably, even with only 30% of the data observed (a 70% missing ratio), the model maintains a high F1 score of 0.751, demonstrating its capability to learn meaningful diagnostic rules from highly incomplete datasets.

### K.2 HEART DISEASE

#### K.2.1 ASSESSMENT OF LEARNED RULES

For feature imputation, as shown in Table 29, our model discovers rules with clinically relevant numerical thresholds by directly modeling continuous data. For instance, it learns to impute resting blood pressure (trestbps) based on conditions like age $> 60$ and chol $> 250$. Similarly, it links high cholesterol to factors like age $> 55$ in males or very high blood pressure (trestbps $> 150$). The learned rule for ST depression (oldpeak) combines the slope of the ST segment with a maximum heart rate threshold (thalach $< 150$), demonstrating the model's ability to capture complex, non-linear relationships within the data.

Beyond imputation, NS-FCN learns interpretable rules for the final diagnosis, classifying patients into low-risk or high-risk categories.

Table 30 presents several of these diagnostic rules. For example, the model learns that a combination of factors such as an upsloping ST segment (slope_upsloping), a fixed thallium defect (thal_fixed_defect), and exercise-induced angina (exang_yes) is strongly indicative of high risk. Conversely, it identifies that factors like the absence of exercise-induced angina (exang_no) and a flat ST slope (slope_flat) in female patients suggest a low risk of coronary artery disease. These diagnostic rules were also evaluated by an LLM and deemed "Excellent" or "Strong," underscoring their consistency with clinical practice.

Table 29: Learned rules for feature imputation on the Heart dataset, with LLM assessments.

| Feature | Imputation Acc. | Learned Rule Example | LLM Assessment |
|---------|-----------------|----------------------|----------------|
| trestbps | 0.86 | $trestbps\_high \leftarrow$ (age $> 60$) $\wedge$ (chol $> 250$) | **Excellent:** This rule captures the well-established link between age, high cholesterol, and hypertension. Both are primary risk factors for cardiovascular disease and often co-occur. |
| chol | 0.85 | $chol\_high \leftarrow$ (sex $= 1 \wedge$ age $> 55$) $\vee$ (trestbps $> 150$) | **Excellent:** The rule correctly identifies two key risk profiles for high cholesterol: middle-aged to elderly males, and individuals with significant hypertension. This aligns perfectly with clinical understanding of metabolic syndrome. |
| thalach | 0.90 | $hr\_high \leftarrow$ (trestbps $> 145$) $\vee$ (age $> 57 \wedge$ cp $= 3$) | **Strong:** This rule insightfully links factors that limit exercise capacity to the maximum heart rate achieved. Both hypertension and severe asymptomatic coronary disease can prevent a patient from reaching a higher peak heart rate. |
| oldpeak | 0.76 | $st\_severe \leftarrow$ (slope $= 2$) $\wedge$ (thalach $< 150$) | **Excellent:** This rule identifies a classic high-risk pattern. A downsloping ST segment is a strong positive finding, and its occurrence at a sub-maximal heart rate indicates ischemia at a low workload, a sign of severe coronary artery disease. |

### K.3 HYPER-PARAMETERS SETTING AND COMPUTING RESOURCE

For NS-FCN (Ours):

- Rule embedding optimizer: Adam with learning rate of 0.01.
- Fine-tune optimizer: Adam with learning rate of 0.01.

Table 30: Learned rules examples for disease prediction on the Heart dataset, with LLM assessments.

| Learned Rule Example | LLM Assessment |
|---|---|
| *high_risk* ← *restecg_stt_abnormality* ∧ *ca* = 3 ∧ *oldpeak* > 1.49 | **Excellent:** This rule identifies a high-risk profile by combining three critical indicators of severe coronary artery disease: significant ST depression, an abnormal resting ECG, and extensive vessel blockage. |
| *high_risk* ← *slope_downsloping* ∧ *restecg_normal* ∧ *trestbps* > 145.68 | **Strong:** A downsloping ST segment is a powerful predictor of ischemia. Combining this with hypertension identifies patients at high risk, even if their resting ECG appears normal, highlighting the importance of stress-test indicators. |
| *high_risk* ← *slope_flat* ∧ *oldpeak* > 1.49 ∧ *restecg_hypertrophy* | **Excellent:** This rule effectively combines signs of acute ischemia (a flat ST slope with significant depression) with evidence of chronic cardiac stress (left ventricular hypertrophy). This profile is strongly indicative of advanced coronary artery disease. |

- Temperature of softmin and softmax: 0.1 (for Eq. 3) and 10.0 for (Eq. 4).
- Our model can run efficiently on a CPU environment with the PyTorch package.

Baselines:

- **BRCG** (Dash et al., 2018), **LEN** (Barbiero et al., 2022), **DR-NET** (Qiao et al., 2021), **RRL** (Wang et al., 2021) are trained with the default hyperparameter settings specified in the original paper.
- **MICE** (Van Buuren & Groothuis-Oudshoorn, 2011): We use $m = 5$ imputations and $maxit = 5$ iterations with the default imputation methods in the `mice` R package.
- **MissForest** (Stekhoven & Bühlmann, 2012): We use the default hyperparameter settings in the `missForest` R package.
- **MLP**: We train a 3-layer fully connected network (input-128-128-output) with batch size 32, learning rate 0.001, and 100 epochs using Adam optimizer.
- **VAE** (Veldkamp et al., 2025): We use a variational autoencoder with latent dimension 16, encoder architecture (input×2-128-64-latent), decoder architecture (latent-64-128-output), batch size 32, learning rate 0.001, and 100 epochs.
- **DAE (mDAE)** (Dupuy et al., 2024): We use a denoising autoencoder with bottleneck dimension 16, encoder architecture (input-128-64-bottleneck), decoder architecture (bottleneck-64-128-output), corruption rate $\rho = 0.2$, batch size 32, learning rate 0.001, and 100 epochs.
- **GAIN** (Yoon et al., 2018): We use mini-batch size 128, hint rate $p_{hint} = 0.9$, MSE loss weight $\alpha = 100.0$, cross-entropy loss weight $\mu = 100.0$, learning rate 0.001, and 1000 epochs.
- **MissDiff** (Ouyang et al., 2023): We use 1000 diffusion timesteps with $\mu_{start} = 10^{-4}$ and $\mu_{end} = 0.02$, batch size 32, learning rate 0.001, and 100 epochs.
- All baseline models can run efficiently on CPU environment with PyTorch package (for deep learning methods) or R packages (for statistical methods).

## L    LIMITATION

While our model shows promising performance, the ethical implications, such as potential over-reliance or misuse for inferring sensitive information, require careful consideration.

Despite its strengths, NS-FCN has limitations. While effective, the asynchronous coordinate gradient descent optimization can be computationally intensive. Besides, the negative predicates are not well explored (we consider negative predicates as an independent predicate from positive predicates). Furthermore, while our model can derive predicates from continuous features, the current implementation learns a single threshold per feature, which may not capture more complex relation-

ships (e.g., intervals). Extending the framework to learn more expressive predicates from continuous data is a promising direction for future work.

# M    USE OF LLMS

In this paper, LLMs were used solely for writing polishing. The key idea, the model design, research study, and all substantive writing are completed by human authors.

In the assessment of discovered rules, we use LLM to write the evaluation of rule quality, which we have mentioned in the paper.

