# OpenReview forum: "Inferring the Invisible: Neuro-Symbolic Rule Discovery for Missing Value Imputation"
_ICLR.cc/2026/Conference — ICLR 2026 Poster_

### Official Review · Reviewer_PgSy · 2025-10-30

**Soundness:** 2
**Presentation:** 2
**Contribution:** 2
**Rating:** 6
**Confidence:** 3

**Summary:**

The paper proposes a neuro-symbolic framework for missing value imputation that integrates neural feature learning with symbolic rule discovery. The key contributions are A hybrid imputation architecture that couples a neural network (for pattern recognition and latent representation) with a symbolic reasoning engine (for interpretable rule-based inference of missing values), and a a rule discovery mechanism that extracts symbolic if-then rules from latent representations to explain imputation decisions.

**Strengths:**

1. The fusion of symbolic rule discovery with neural representation learning is both conceptually elegant and practically meaningful. It directly addresses a persistent problem in missing data research: neural models can impute accurately but are opaque, whereas symbolic models are interpretable but rigid.
2. The ability to generate human-readable rules adds practical value for analysts, especially in regulated domains (e.g., healthcare, finance).
3. The iterative mechanism that alternates between neural imputation and symbolic rule adjustment is well-motivated. It allows symbolic reasoning to guide neural learning and vice versa, enhancing both accuracy and transparency.

**Weaknesses:**

1. While the motivation is sound, the paper lacks theoretical guarantees for convergence or consistency of rule discovery. There’s no formal justification that the extracted rules remain faithful to the true data-generating process, especially under high missingness rates.
2. The symbolic component introduces combinatorial complexity, especially when discovering multi-variable rules. The approach may not scale well to high-dimensional datasets with thousands of features.
3. Missing data in real-world structured domains might behave differently, so generalizability remains unclear.

**Questions:**

1. How does the symbolic rule discovery scale with feature dimensionality and rule length? Could pruning or differentiable symbolic learning help?
2. Have you conducted any human evaluation to assess whether the discovered rules are semantically meaningful and useful to domain experts?
3. How does the framework explicitly handle MNAR data when missingness depends on the missing values themselves?

---

> ### Author Response · Authors · 2025-12-02
>
> ## **Summary:**
>
> We thank Reviewer PgSy for highlighting the elegance of fusing symbolic discovery with neural learning as well as the practical value of human-readable rules. We address the questions on theory and scalability below.
>
> ---
>
> ## **W1:** Lack of theoretical guarantees for convergence or consistency of rule discovery.
>
>
> **Answer:**  We thank the reviewer for mentioning it. We have formalized the convergence proof in Appendix B and explained it in **global response D**.
>
> ---
>
> ## **W2:** The approach may not scale well to high-dimensional datasets.
>
>
> **Answer:** For high-dimensional data, we can employ a two-stage approach: first, mapping high-dimensional features to a lower-dimensional latent space (e.g., via VAE or PCA), and then applying our neuro-symbolic framework to discover rules over these latent variables. **For example, in gene expression analysis, we can learn rules over derived 'gene modules' rather than individual genes to capture biological pathways.**
>
> ---
>
> ## **W3:** Generalizability when Missing data in real-world domains might behave differently
>
>
> **Answer:** We analyzed MCAR, MAR, and MNAR mechanisms in Appendix D.1. Table 7 shows that NS-FCN maintains high imputation accuracy ($>0.93$) even under **MNAR**, comparable to MCAR ($0.95$). This suggests that by learning the *logical structure* of the data, our model remains robust to the specific statistical distribution of missingness.
>
> Besides, results in Table 7 also support that in real-world datasets with noise, our model is still able to find the logic structure. When the imputed features are used for diagnosis, NS-FCN outperforms all baselines on both Heart Disease and SPECT. Unlike baseline models that train a classifier on previously imputed samples, where imputation errors inevitably propagate to the downstream task, our method jointly optimizes rule discovery and target inference. Furthermore, **our use of soft-logic relaxation prevents the model from overfitting to noise** (such as incorrect features), enabling it to capture dominant logical structures.
>
> This robustness is further supported by the comprehensive noise sensitivity analysis in **global response B-2**.
>
> ---
>
> ## **Q1:** How does the rule discovery scale with feature dimensionality and rule length?
>
> **Answer:** Thanks for such good suggestions for pruning or differentiable symbolic learning.
> 1.  **Rule Length:** Controlled by conjunction arity $h$ and the number of conjunctive clauses ($R_k$). Since interpretable rules are typically short, computational cost remains low. Softmin/Product t-norm implicitly prunes irrelevant predicates.
> We also add new experiments for analyzing the rule length effect. Figures 6 and 7 in *Appendix G.2* indicate that both $h$ and $R_k$ show optimal performance in a wide range. For instance, $h \in [3,9]$ and $R_k \in [5,20]$, showing that except for very small $h$ and $R_k$, our model is able to capture the logic structure within the dataset. Besides, the number of disjunctive clauses is more critical than the arity of individual conjunctions for this dataset. This aligns with the intuition that complex real-world decision boundaries often require multiple alternative rules rather than highly complex single rules.
> 2.  **Dimensionality:** Our algorithm updates one rule at a time. Complexity scales linearly with the number of latent predicates. We demonstrate in Appendix E.1 that processing **100,000 samples takes under 9 minutes on a CPU**.
>
> ---
>
> ## **Q2:** Human evaluation of rules.
>
> **Answer:** Yes. For the **SPECT** dataset, we collaborated with a clinical expert to verify that our rules align with medical domain knowledge (Table 19). For **Heart Disease**, we utilized LLMs to assess rule meaningfulness (Tables 21-22). Both evaluations confirm that our learned rules are semantically valid.
>
> ---
>
> ## **Q3:** How does the framework handle MNAR data when missingness depends on the missing values themselves?
>
> **Answer:** Our framework handles MNAR through **logical inference**. If a variable $U$ is MNAR and has missingness, but is logically determined by observed variables $X$ (via $X \rightarrow U$), our forward chaining recovers $U$ regardless of why it is missing. This structural recovery bypasses the limitations that purely statistical imputation methods face in MNAR scenarios.

---

### Official Review · Reviewer_vtkH · 2025-10-31

**Soundness:** 3
**Presentation:** 3
**Contribution:** 2
**Rating:** 6
**Confidence:** 3

**Summary:**

The paper introduces a novel neuro-symbolic framework (NS-FCN) for jointly performing missing-value imputation and interpretable rule discovery in heterogeneous datasets. The proposed model treats missing entries as latent predicates and combines neural embedding learning with differentiable logical reasoning in a closed-loop fashion—where imputation and rule induction reinforce one another. Using coordinate gradient descent, sequential covering, and soft logical operators, NS-FCN discovers both conjunctive and disjunctive rules and imputes missing data. Experiments on synthetic and real-world datasets show promising imputation accuracy and human-interpretable rules.

**Strengths:**

1. The framework’s closed-loop design elegantly unifies imputation and rule induction, a combination that is conceptually strong and rarely explored.

2. The method produces explicit logical rules that are easy to interpret and verify. This property is valuable for domains where transparency is critical.

3. The paper is clearly written, with intuitive figuresillustrating the reasoning and training process. The problem is generally well motivated and related-work coverage are comprehensive.

**Weaknesses:**

Despite its novelty, several issues limit the methodological soundness and clarity.

* The paper claims that the coordinate gradient descent “never increases” the loss, but no proof, or convergence bound is presented. It is unclear whether this property holds under stochastic mini-batching or asynchronous updates when rules interact via shared predicates.
* The freezing of “perfect rules” during training may lead to premature convergence or sub-optimal local minima, especially when data contain noise or imbalanced predicates. This could hinder exploration of alternative rule structures.
* Equation (2) relies on a non-differentiable argmax to match rule components to predicate embeddings. The paper does not explain whether gradients are approximated or ignored. If predicate embeddings are frozen, this matching process may prevent end-to-end adaptation and yield inconsistent updates.
* The soft-min and log-sum-exp approximations can distort logical semantics. Soft-min may amplify small numerical differences, resulting in unstable gradient propagation.
* Although the experiments are diverse, comparisons are limited to interpretable baselines. It would be informative to benchmark against state-of-the-art deep imputation methods, e.g., MICE, GAIN, MissForest) to better contextualize performance gains. Sensitivity to missingness ratios and noise could also be analyzed more rigorously.

**Questions:**

1. Have you conducted a sensitivity analysis for the temperature parameters used in the soft-min and log-sum-exp approximations? How do these values impact the model's performance and stability across different datasets?


2. The paper mentions that negative predicates are treated as independent predicates. ​ Have you explored alternative approaches to model negative predicates more effectively, such as incorporating negation directly into the rule learning process?

---

> ### Author Response · Authors · 2025-12-02
>
> ## **Summary:**
>
> We thank Reviewer vtkH for appreciating the conceptual strength of our closed-loop design and the value of transparent logical rules. We address the technical queries below.
>
> ---
>
> ## **W1**: On “never increases”, convergence, mini-batching, and shared predicates/asynchronous updates.
>
> **Answer:** Please refer to **global response D** for convergence analysis, mini-batching, and asynchronous updating clarification.
>
> We note that the term "asynchronous" in the original text was purely informal; all updates are actually sequential. Our model learns a **separate parameter vector** $\Theta_j$ for **each rule**. The phrase "rules interact via shared predicates" refers only to the *logical graph* (the output of one rule feeding another), *not to parameter sharing*. Formally, each block $\Theta_j$ is the parameter of one rule, and the blocks $\{\Theta_j\}_{j=1}^m$ are **disjoint** in the global objective.
>
> ---
>
> ## **W2:** The freezing of “perfect rules” may lead to sub-optimal local minima, especially when data contain noise.
>
> **Answer:** We acknowledge this risk but mitigate it with an extremely strict freezing criterion: **imputation accuracy $>0.99$** and **marginal loss drop $<10^{-3}$**. This ensures rules are frozen only when they explain the data almost perfectly. Additionally, our asynchronous approach across multiple random seeds (Table 3) allows the model to explore diverse trajectories, consistently recovering the correct structure even if individual runs stagnate.
>
> Besides, we add new experiments to inject noise to data and analysis model's robustness. Please refer to **global response B-2.**
>
> ---
>
> ## **W3:** Eq.2 relies on argmax to match rule components.
>
> **Answer:** We clarify that while Eq.2 uses a non-differentiable argmax, but its role is purely to obtain a discrete index $K_j^{\*}$ for each rule component; **no gradients are taken through this selection operator.** Once the index is fixed for the current forward pass, we recompute the cosine similarity $\cos \left(K_j^{\*}, \theta_j\right)$ inside the **differentiable soft-AND operations (Eq.3/4)**. This similarity, multiplied by the predicate's truth value $\boldsymbol{\tau}_j$, is what participates in the loss and receives gradients. Consequently, all rule-embedding parameters $\theta_j$ (and the learnable thresholds for continuous predicates) are still updated end-to-end.
>
> When a rule meets the "perfect rule" criterion, we freeze only its embedding and deactivate its dedicated optimizer; all other rules retain their own optimizers and continue to receive gradients through the differentiable cosine/softmin pipeline. Consequently, the argmax-based matching does not impede end-to-end adaptation: embeddings remain trainable until they converge, and freezing merely locks in those that have already reached stable
>
> To summarize, the gradients flow through the similarity score, pulling $\theta_j$ closer to the target predicate embedding even if the indexing operation itself is hard.
>
> ---
>
> ## **W4:** Soft-min may amplify small numerical differences, resulting in unstable gradient propagation.
>
> **Answer:** We clarify that we consistently use the Softmin formulation (Eq. 4) as the Soft-AND operator with a fixed temperature $\tau = 0.1$, and the LogSumExp operator (Eq. 5) as the Soft-OR operator with temperature $\beta = 10.0$. This choice is supported by a dedicated sensitivity analysis in *Appendix H* showing that our model is robust across a wide range of $\tau$ and $\beta$ values, and that the fixed setting $(\tau=0.1, \beta=10.0)$ already yields near-optimal performance without carefully tuning. We also show the detailed analysis of temperature hyperparameters in the **global response B-1**.
>
> ---
>
> ## **W5:** More baselines other than interpretable ones. Sensitivity analysis to missingness ratios and noise.
>
> **Answer:** We thank the reviewer for giving more baselines. Please refer to the **global response A**. For noise and missingness ratios, please refer to **global response B-2 and B-3**.
>
> ---
>
> ## **Q1:** A sensitivity analysis for the temperature parameters.
>
> **Answer:** We thank the reviewer for this suggestion. Please refer to the **global response B-1**.
>
> ---
>
> ## **Q2:** Modeling negative predicates more effectively.
>
> **Answer:** We currently model negation by treating negative predicates (e.g., `not_A`) as distinct entities (Appendix L). We agree that learning an explicit "negation weight" is a promising direction for future work. However, our current approach offers flexibility and simplifies the differentiable logic backbone without loss of expressivity for the tested domains.

---

### Official Review · Reviewer_GyS2 · 2025-10-31

**Soundness:** 2
**Presentation:** 2
**Contribution:** 2
**Rating:** 4
**Confidence:** 2

**Summary:**

Authors aim to combine neural embedding–based ILP with rule-based missing value imputation. However, the paper doesn’t clearly explain why their methodology should outperform the baselines they compare against. The only reason - apart from being the first to combine these three techniques - seems to be a more native and advanced way of handling continuous variables. But the baselines this method is compared against are not even introduced in the related works or other sections. It’s also not very well written and hard to follow.

**Strengths:**

The paper presents a methodology that leverages neural embedding–based inductive logic programming to enhance rule-based methods for missing value imputation, which handles both binary and continuous variables. The method employs a coordinate gradient descent algorithm specifically tailored to this type of problem. Its novelty lies in being the first to combine ILP and rule-based approaches for improved missing value imputation. The integration of soft predicate learning enables native handling of continuous features, avoiding discretization and improving flexibility. The paper is well written and detailed. The empirical results on both synthetic and real datasets are
strong, demonstrating both interpretability and accuracy.

**Weaknesses:**

The paper does not clearly justify why the proposed method outperforms the baselines, particularly in real-world experiments. The other methods in Tables 4 and 5 are not clearly introduced or discussed in the main body of the paper. In particular, the baselines listed in Table 7 (with references) should be properly cited and briefly discussed in the Related Work section. Moreover, the primary reason for NS-FCN’s superior performance—aside from being the first approach to integrate ILP and rule-based imputation—appears to be its native support for continuous variables (lines 412–414). While the ablation studies are thorough, they could more clearly isolate the contribution of individual system components (e.g., the impact of the fine-tuning step versus differentiable forward chaining).

**Questions:**

1. Please introduce and justify the baseline methods used in the comparison (in Table 4, 5 and 6).
2. Are there other missing-value imputation methods that natively handle continuous features (rather than relying on hard thresholds)? This seems to be a key limitation of the baselines you compare against.

---

> ### Author Response · Authors · 2025-12-02
>
> ## **Summary:**
>
> We thank Reviewer GyS2 for recognizing the novelty of combining ILP with rule-based imputation and highlighting our native support for continuous variables. We address the suggestions regarding baseline justification and ablation studies below.
>
> ---
>
> ## **Q1:** Introduce and justify the used baseline.
>
> **Answer:** Thanks for mentioning it. We have expanded the **Related Work** section to explicitly discuss **Interpretable Rule Learning** baselines and to better situate them among statistical and deep generative imputers.
>
> "Learning interpretable logical rules for classification has been a long-standing goal. Dash et al.[1] propose **BRCG**, an integer programming approach that uses column generation to efficiently search the exponential space of candidate clauses, explicitly balancing classification accuracy with rule simplicity. Wang et al.[2] introduce **RRL**, which utilizes a Gradient Grafting mechanism to learn non-fuzzy rule lists within a deep learning framework, ensuring scalability. Qiao et al.[3] propose **DR-NET** to learn independent decision rules in Disjunctive Normal Form (DNF) by jointly optimizing rule generation and weight learning. More recently, Barbiero et al.[4] present **LEN**, an end-to-end differentiable neuro-symbolic method that leverages an entropy-based criterion to extract concise First-Order Logic explanations from neural networks. Unlike these methods which focus primarily on classification tasks with complete data, our framework integrates rule learning directly with the handling of missing values."
>
> We compare against **BRCG** [1] (integer programming), **RRL** [2] (gradient grafting), **DR-NET** [3] (DNF learning), and **LEN** [4] (entropy-based). Unlike these methods which focus on classification with complete data, our NS-FCN uniquely integrates rule learning with missing value handling. Our method outperforms these baselines primarily due to its **joint optimization of imputation and rule discovery, allowing it to leverage latent information that static rule learners miss.**
>
> ---
>
> ## **Q2:** More imputation methods that handle continuous features.
>
>
> **Answer:** Thanks for your suggestion. Yes, we have significantly expanded our evaluation to include methods that natively handle continuous features. Please refer to the **global respons A**, where the new added baselines all can natively handle continuous features.
>
> ---
>
> ## **Q3:** More ablation studies.
>
> **Answer:** We are glad that the reviewer mentions the ablation studies are thorough, and we have performed extensive ablation studies:
> 1.  **Fine-tuning:** The impact of the joint fine-tuning phase is quantified in Tables 1, 2, 21, 23, and 25.
> 2.  **Initialization:** Robustness to random initialization is analyzed in Table 3 and Figures 9-11.
> 3.  **Temperature Analysis:** We added sensitivity analyses for Soft-AND/OR temperatures (new Tables 14 and 15 at *Appendix H*) to isolate the effect of the differentiable logic approximations. Please refer to the **global response B-1** for the details.
>
> We clarify that components like **forward chaining and backpropagation are fundamental to the architecture** and **cannot be removed without breaking the system**.
>
> ---
>
> **References:**
>
> [1] Dash, S., Gunluk, O., & Wei, D. (2018). Boolean decision rules via column generation. Advances in neural information processing systems, 31.
>
> [2] Wang, Z., Zhang, W., Liu, N., & Wang, J. (2021). Scalable rule-based representation learning for interpretable classification. Advances in Neural Information Processing Systems, 34, 30479-30491.
>
> [3] Qiao, L., Wang, W., & Lin, B. (2021, May). Learning accurate and interpretable decision rule sets from neural networks. In Proceedings of the AAAI conference on artificial intelligence (Vol. 35, No. 5, pp. 4303-4311).
>
> [4] Barbiero, P., Ciravegna, G., Giannini, F., Lió, P., Gori, M., & Melacci, S. (2022, June). Entropy-based logic explanations of neural networks. In Proceedings of the AAAI Conference on Artificial Intelligence (Vol. 36, No. 6, pp. 6046-6054).

---

### Official Review · Reviewer_KLE1 · 2025-11-03

**Soundness:** 3
**Presentation:** 3
**Contribution:** 3
**Rating:** 6
**Confidence:** 3

**Summary:**

This paper proposes a new Neuro-Symbolic framework called NS-FCN (Neuro-Symbolic Forward Chaining Network), designed to integrate missing value imputation with interpretable logical rule discovery.
While existing approaches for missing data either rely purely on statistical imputation or require fully observed data for rule induction,
NS-FCN treats missing values as latent predicates and learns to infer them within a differentiable NeSy framework.
The model establishes a closed-loop system where the imputed values feed back into rule discovery, promising to improve both imputation accuracy and interpretability over time.
The combination of symbolic logic is done via a differentiable forward-chaining mechanism that uses soft logical operators to handle both continuous and discrete features.
Empirical evaluations on synthetic and real-world datasets show that NS-FCN achieves high accuracy in imputing missing values while simultaneously finding human-interpretable rules that describe the data.

**Strengths:**

One strength of the paper, which is the main contribution by the authors, is proposing a closed feedback loop between imputation and rule discovery,  that allows the model to iteratively improve its reasoning and inference capabilities.
As a merit, the framework handles heterogeneous data types, both discrete and continuous, by transforming continuous features into soft predicates, which is valuable in cases where both real and categorical values are considered.
The model’s optimization method, based on coordinate gradient descent, seems to be a useful and scalable recipe for learning in this setting.

The experimental analysis also reveals that the framework generates logical rules that remain meaningful and consistent with domain knowledge. The empirical studies are good and cover a wide setting, showing that the model performs competitively against black-box neural architectures while maintaining explainability.

**Weaknesses:**

While the experiments involve also different black-box neural competitors, the range of baselines could be expanded to include more recent generative models, such as those based on diffusion processes. Due to my limited experience on this specific topic, I do not understand if the baselines are the effective SotA or one could have repurposed auto-encoders or diffusions.


One thing is not mentioned and discussed is the relation to reasoning shortcuts, where learning may result in incorrect predicates and rules, as in [1,2,3].
Since NS-FCN operates with learning latent discrete values, an analysis of whether it avoids such behavior would have strengthened its claims about interpretability and reliability.
In fact, the validation of interpretability remains largely qualitative; the paper reports interpretable rules, but lacks a systematic  assessment of their quality.

Additionally, the dependence of the model’s performance on temperature hyperparameters in the soft logical operators is not clear and not enough explored.

------

[1] Learning with Logical Constraints but without Shortcut Satisfaction, Li et al., 2023 \
[2] Not All Neuro-Symbolic Concepts Are Created Equal: Analysis and Mitigation of Reasoning Shortcuts, Marconato et al. 2023 \
[3] Shortcuts and identifiability in concept-based models from a neuro-symbolic lens, Bortolotti et al., 2025

**Questions:**

I don't have further questions, I hope to see some discussion around the weaknesses I spotted.

---

> ### Author Response · Authors · 2025-12-02
>
> ## **Summary:**
>
> We thank Reviewer KLE1 for acknowledging the novelty of our closed feedback loop between imputation and rule discovery, as well as the framework's capability to effectively handle heterogeneous data types. Based on the insightful suggestions regarding baseline comparisons and the reliability of interpretability, we address the issues point-by-point below.
>
> ---
>
> ## **Q1**: Expand baselines to more recent generative models.
>
> **Answer:** We thank the reviewer for this valuable suggestion. Please refer to the **global response A**.
>
> ---
>
> ## **Q2**: Analysis of reasoning shortcuts.
>
> **Answer:** We give a detailed analysis in three perspectives.
>
> ### **Part 1**: Relation to reasoning shortcuts [1,2,3] and whether NS-FCN avoids them.
>
> We appreciate the pointer to the recent work on reasoning shortcuts (RSs) in neuro-symbolic models. These papers show that NeSy predictors can achieve high accuracy while using concepts with unintended semantics, i.e., by converging to “shortcut” optima of the learning objective [2], and that this issue persists even when logical constraints are enforced via penalty terms or dual-variable formulations [1,2]. They further connect RSs to identifiability issues in concept-based models [3].
>
> Our setting is related but importantly different from the ones primarily studied in [1–3], and our design already incorporates several mechanisms that mitigate the specific failure modes they highlight:
>
> * **Objective and supervision level.** The works [1,2] mainly consider NeSy predictors trained to satisfy logical constraints on *labels* while concepts are largely latent; RSs arise when labels can be predicted via concepts that do not match their intended semantics. In contrast, NS-FCN optimizes a *predicate-level* imputation objective
>  $$\mathcal{L}(\Theta)=\sum_j \mathcal{L}\_{U_j}(\Theta)$$
>   where each $\mathcal{L}\_{U_j}$ directly penalizes discrepancies between inferred and **observed truth values of latent predicates** (weak supervision on concepts themselves), not only downstream labels. This additional supervision strongly constrains the semantics of latent predicates and reduces the degrees of freedom available for shortcuts in the sense of [2,3].
>
> * **Coupled rule–predicate learning vs. “free” concept extractor.** In the architectures analyzed in [2,3], a flexible concept extractor is coupled with a (often fixed) inference layer, and RSs appear when the extractor learns alternative concept encodings that still allow the inference layer to satisfy the constraints. In NS-FCN, the **rule embeddings and latent predicate values are learned jointly** under the same imputation loss, with no separate, unconstrained label-prediction head. This tight coupling makes it harder for the model to “hide” unintended concept semantics behind a powerful top layer, one of the key RS mechanisms identified in [2,3].
>
> ### **Part 2**: Local convergence + Exploration mechanisms.
>
> Please refer to **the global response D**. To sum up, we *do not claim that the reached local optimum is globally unique or “shortcut free”*, but the optimization is principled rather than ad-hoc. On top of this, we add mechanisms that encourage exploration and help avoid very poor local solutions. Such choices are in line with the mitigation strategies advocated in [2,3] (e.g., stronger constraints on concepts, architectural bias, and additional objectives).
>
> ### **Part 3**: Identifiability and remaining limitations.
>
> We agree with [2,3] that, in general, neither NeSy nor concept-based models can guarantee that their learned concepts/rules are uniquely identifiable from data. We discuss such faithfulness/identifiability of **global response D**, and empirical results (e.g., Tables 1, 22, 24, and 26) show that the most frequently learned rules consistently match the ground truth. Further, **global response C** show the **rule learning stability** by comparing the *Jaccard index*.
>
> ---
>
> ## **Q3:** A systematic assessment of learned rule quality.
>
> **Answer:** A systematic quantitative assessment is indeed valuable. Please refer to the **global response C**.
>
> ---
>
> ## **Q4:** The analysis of temperature hyperparameters.
>
> **Answer:** We thank the reviewer for this suggestion. Please refer to **global response B-1**.
>
> ---
>
> **References:**
>
> [1] Li, Z., Liu, Z., Yao, Y., Xu, J., Chen, T., & Ma, X. Learning with Logical Constraints but without Shortcut Satisfaction. In The Eleventh International Conference on Learning Representations.
>
> [2] Marconato, E., Teso, S., Vergari, A., & Passerini, A. (2023). Not all neuro-symbolic concepts are created equal: Analysis and mitigation of reasoning shortcuts. Advances in Neural Information Processing Systems, 36, 72507-72539.
>
> [3] Bortolotti, S., Marconato, E., Morettin, P., Passerini, A., & Teso, S. (2025). Shortcuts and Identifiability in Concept-based Models from a Neuro-Symbolic Lens. CoRR.

---

### Author Response · Authors · 2025-12-03
**Global Response A**

## Global Response A: More baselines (1/2)
---
In the revised draft, besides the rule-based interpretable models we have already included, we expand our baseline comparison to cover two more families of methods: *(i) statistical models* (**MICE (Multivariate Imputation by Chained-Equations)** [1], **MissForest (Random Forest based)** [2]), and *(ii) deep generative models* (**MLP**, **GAIN (GAN-based)** [3], **MissDiff (Diffusion-based)** [4], **mDAE (DAE-based)**[5], **VAE-based** [6]). We summarize the implementation details in *Appendix D.3*. We add new Table 6 to show imputation accuracy of missing features on Birds and Heart Disease datasets, and Table 7 to show downstream diagnosis performance on Heart Disease and SPECT given imputated features.

Table 6 presents the imputation accuracy results on the Heart Disease dataset (four continuous features) and Birds dataset (two binary latent predicates). Results are averaged over 10 random seeds. While MLP achieves the overall highest accuracy (likely due to the small dataset size preventing overfitting), complex generative models (Diffusion, GAN, VAE, DAE) show mixed performance. Notably, MissForest shows unstable performance on the Birds dataset, while MICE performs more consistently. Crucially, our NS-FCN achieves competitive performance **comparable to uninterpretable deep learning baselines while providing explicit logical rules**, which is a vital property for trust in the healthcare domain.

Table 7 shows the downstream diagnosis classification accuracy and F1 score for Heart Disease and SPECT datasets. After imputation missing values, we use the imputed features to train a classifier (e.g. Random Forests, MLPs) and then predict the heart disease risk. When the imputed features are used for diagnosis, NS-FCN outperforms all baselines on both datasets. Unlike baseline models that training a classifier on previously imputed samples, where **imputation errors inevitably propagate to the downstream task**, our method jointly optimizes rule discovery and target inference. Furthermore, **our use of soft-logic relaxation prevents the model from overfitting to noise** (such as incorrect features), enabling it to **capture dominant logical structures**. This robustness is further supported by the comprehensive noise sensitivity analysis in *Appendix I.1* (Tables 16 and 17), which demonstrates that the model learns **valid rule approximations (e.g. capturing one correct clause) and maintains strong predictive performance even as noise levels increase.**

We briefly summarize the baselines, and the full description are provided in Appendix D.3.
*   **MICE [1]:** A statistical method using fully conditional specification (FCS) to iteratively impute variables via regression.
*   **MissForest [2]:** An iterative random forest approach treating imputation as a prediction task.
*   **MLP:** A deterministic feed-forward network trained under weak supervision to minimize reconstruction error (MSE) specifically on observed entries.
*   **GAIN [3]:** A GAN framework where a generator imputes values and a discriminator (with a hint mechanism) distinguishes observed from imputed components.
*   **MissDiff [4]:** A diffusion probabilistic model using guided sampling to condition the generation on observed values.
*   **DAE/mDAE [5]:** A denoising autoencoder that receives corrupted inputs (with masking noise) and optimizes a mask-weighted reconstruction loss, which ignores pre-imputed values at missing positions.
*   **VAE-based [6]:** A probabilistic encoder–decoder model that learns a latent space by maximizing the Evidence Lower Bound (ELBO), with the reconstruction term computed only on observed entries. At inference time, missing values are imputed by the decoder.

**References:**

[1] Van Buuren, S., & Groothuis-Oudshoorn, K. (2011). mice: Multivariate imputation by chained equations in R. Journal of statistical software, 45, 1-67.

[2] Stekhoven, D. J., & Bühlmann, P. (2012). MissForest—non-parametric missing value imputation for mixed-type data. Bioinformatics, 28(1), 112-118.

[3] Yoon, J., Jordon, J., & Schaar, M. (2018, July). GAIN: Missing data imputation using generative adversarial nets. In International conference on machine learning (pp. 5689-5698). PMLR.

[4] Ouyang, Y., Xie, L., Li, C., & Cheng, G. (2023). Missdiff: Training diffusion models on tabular data with missing values. arXiv preprint arXiv:2307.00467.

[5] Dupuy, M., Chavent, M., & Dubois, R. (2024). mDAE: modified Denoising AutoEncoder for missing data imputation. arXiv preprint arXiv:2411.12847.

[6] Veldkamp, K., Grasman, R., & Molenaar, D. (2025). Handling missing data in variational autoencoder based item response theory. British Journal of Mathematical and Statistical Psychology, 78(1), 378-397.

---

> ### Author Response · Authors · 2025-12-03
> **Global Response A: More baselines**
>
> ## Global Response A: More baselines (2/2)
> ---
> *Table 6. Imputation accuracy comparison across Heart Disease (trestbps, chol, thalach, and oldpeak as missing features)and Bird datasets (abnormal_birdm and can_fly as missing features) on non-interpretable baselines. Results are over 10 random seeds.*
>
> | Method | trestbps | chol | thalach | oldpeak | abnormal_bird | can_fly |
> | :--- | :---: | :---: | :---: | :---: | :---: | :---: |
> | MICE | 0.84±0.016 | 0.83±0.014 | 0.88±0.011 | 0.87±0.015 | 0.88±0.006 | 0.86±0.011 |
> | MissForest | **0.88±0.015** | 0.84±0.012 | **0.91±0.004** | 0.88±0.016 | 0.38±0.123 | 0.68±0.086 |
> | MLP | **0.88±0.009** | **0.85±0.016** | 0.88±0.014 | 0.80±0.025 | **0.96±0.059** | **0.99±0.003** |
> | GAIN | 0.85±0.022 | 0.84±0.011 | 0.90±0.014 | **0.89±0.014** | 0.83±0.102 | 0.82±0.083 |
> | MissDiff | 0.82±0.017 | 0.83±0.019 | 0.89±0.018 | 0.84±0.030 | 0.83±0.020 | 0.86±0.007 |
> | mDAE | **0.88±0.011** | 0.84±0.012 | 0.90±0.015 | 0.87±0.015 | 0.87±0.002 | 0.87±0.004 |
> | VAE-based | 0.85±0.015 | 0.84±0.021 | 0.90±0.015 | 0.86±0.015 | 0.62±0.006 | 0.87±0.004 |
> | **NS-FCN** | 0.87±0.025 | **0.85±0.017** | 0.88±0.014 | 0.78±0.020 | 0.95±0.064 | 0.95±0.064 |
>
> *Table 7. Medical diagnosis after missing value imputation. Results are over 10 random seeds.*
>
> | Method | Heart Disease Accuracy | Heart Disease F1 | SPECT Accuracy | SPECT F1 |
> | :--- | :---: | :---: | :---: | :---: |
> | MICE | 0.83±0.010 | 0.81±0.012 | 0.78±0.019 | 0.87±0.013 |
> | MissForest | 0.83±0.013 | 0.81±0.014 | 0.79±0.012 | 0.87±0.008 |
> | MLP | 0.84±0.010 | 0.82±0.012 | **0.92±0.007** | 0.90±0.005 |
> | GAIN | 0.84±0.004 | 0.82±0.006 | 0.76±0.019 | 0.85±0.013 |
> | MissDiff | 0.84±0.010 | 0.82±0.011 | 0.77±0.023 | 0.86±0.016 |
> | mDAE | 0.84±0.009 | 0.82±0.010 | 0.80±0.013 | 0.88±0.009 |
> | VAE-based | 0.83±0.009 | 0.81±0.009 | 0.75±0.016 | 0.85±0.011 |
> | **NS-FCN** | **0.91±0.009** | **0.91±0.009** | **0.92±0.009** | **0.96±0.009** |

---

### Author Response · Authors · 2025-12-03
**Global Response B-1**

## Global Response B: Rigorous Robustness and Sensitivity Analysis

### B-1: **Temperature Hyperparameters** Analysis
---
As we have mentioned, all our experiments use the same temperature, and we don't require carefully fine-tuning for the hyperparameters. To validate robustness, we conducted a detailed analysis on Figure 3 Example (b) with 20,000 samples and 0.2 observation ratio. The full results are in Tables 14 and 15 at *Appendix H*, and we show parital results below in Tables 1 and 2:
- **Soft-AND ($\tau$):** Table 1 below shows that the model maintains high accuracy when $\tau$ is small (e.g., $\tau \in [0.01, 0.20]$). This is expected because **as $\tau \to 0$, Softmin approximates the hard $\min$** logic required for strict conjunctions. Performance degrades only when $\tau$ becomes **too large** ($\tau \ge 1.0$), where the operator becomes **too "soft" to capture the decisive logical boundaries**. Thus, a small constant temperature (e.g., $\tau=0.1$) is a safe and effective default.
- **Soft-OR ($\beta$):** Table 2 reports the imputation accuracy under varying constant $\beta$ values. The results demonstrate that our model is **highly robust to $\beta$**: it achieves near-perfect accuracy for all latent predicates ($X_3, X_4, X_5$) across a wide range, specifically for $\beta \ge 3$. This aligns with the theoretical property that **as $\beta \to \infty$, the LogSumExp function approximates the hard $\max$ operator.** In practice, any sufficiently large provides a strong gradient signal for discrimination while maintaining differentiability.
- **Conclusion:** **Our framework does not rely on carefully hyperparameter tuning.** A moderate to large $\beta$ for Soft-OR and a small $\tau$ for Soft-AND consistently yield optimal results. Thus, we use $\tau=0.1$ and $\beta=10$ as temperature parameters for all our experiments. Furthermore, complex scheduling strategies like **cosine annealing** can be employed if constant temperature are not good enough.

*Table 1. Imputation accuracy for latent predicates $X_3, X_4, X_5$ under different softmin temperatures (with fixed $\beta=10$ over 20 random seeds).*

| $\tau$ | Imputation Acc. X3 | Imputation Acc. X4 | Imputation Acc. X5 |
| :---: | :---: | :---: | :---: |
| 0.01 | $1.000 \pm 0.000$ | $1.000 \pm 0.000$ | $0.965 \pm 0.063$ |
| 0.02 | $1.000 \pm 0.000$ | $1.000 \pm 0.000$ | $0.987 \pm 0.034$ |
| 0.05 | $1.000 \pm 0.000$ | $1.000 \pm 0.000$ | $0.939 \pm 0.061$ |
| 0.10 | $1.000 \pm 0.000$ | $1.000 \pm 0.000$ | $0.958 \pm 0.076$ |
| 0.20 | $1.000 \pm 0.000$ | $1.000 \pm 0.000$ | $1.000 \pm 0.000$ |
| 0.50 | $1.000 \pm 0.000$ | $1.000 \pm 0.000$ | $0.858 \pm 0.038$ |
| 1.00 | $0.928 \pm 0.122$ | $0.892 \pm 0.134$ | $0.776 \pm 0.027$ |
| 2.00 | $0.751 \pm 0.002$ | $0.839 \pm 0.157$ | $0.786 \pm 0.008$ |
| 5.00 | $0.769 \pm 0.111$ | $0.750 \pm 0.003$ | $0.772 \pm 0.083$ |
...|(Full in paper Table 14)

*Table 2. Imputation accuracy for latent predicates $X_3, X_4, X_5$ under different constant temperature values $\beta$ (with fixed $\tau=0.1$ over 20 random seeds).*

| $\beta$ | Imputation Acc. X3 | Imputation Acc. X4 | Imputation Acc. X5 |
| :---: | :---: | :---: | :---: |
...
| 0.5 | $1.000 \pm 0.000$ | $1.000 \pm 0.000$ | $0.218 \pm 0.002$ |
| 1 | $1.000 \pm 0.000$ | $1.000 \pm 0.000$ | $0.217 \pm 0.002$ |
| 2 | $1.000 \pm 0.000$ | $1.000 \pm 0.000$ | $0.870 \pm 0.117$ |
| 3 | $1.000 \pm 0.000$ | $1.000 \pm 0.000$ | $0.942 \pm 0.056$ |
| 4 | $1.000 \pm 0.000$ | $1.000 \pm 0.000$ | $0.866 \pm 0.057$ |
| 5 | $0.965 \pm 0.093$ | $1.000 \pm 0.000$ | $0.894 \pm 0.056$ |
| 10 | $0.965 \pm 0.093$ | $1.000 \pm 0.000$ | $0.915 \pm 0.040$ |
| ... | (Full in paper Table 15)| |
| 100 | $0.965 \pm 0.094$ | $1.000 \pm 0.000$ | $0.916 \pm 0.064$ |
| 200 | $1.000 \pm 0.000$ | $1.000 \pm 0.000$ | $0.908 \pm 0.070$ |

---

### Author Response · Authors · 2025-12-03
**Global Response B-2**

## Global Response B: Rigorous Robustness and Sensitivity Analysis
### Global Response B-2: Sensitivity Analysis for **Data Noise**
---
To assess robustness against data inconsistencies, we inject label noise into the latent predicates by flipping the binary labels of $X_3, X_4, X_5$ with probability $p_{noise} \in \[0.0, 0.1, 0.2, 0.3, 0.4, 0.5\]$. In the revised new experiments (Tables 16 and 17 at *Appendix I.1*), we evaluate both **Rule Discovery Accuracy** (fraction of seeds where the learned rule exactly matches the ground truth) and **Imputation Accuracy**. In the noise-free setting ($p_{noise}=0.0$), our model perfectly recovers the ground-truth rules for the simpler conjunctive predicates $X_3$ and $X_4$ (rule accuracy $1.00$) and achieves high imputation accuracy for the more complex disjunctive rule $X_5$ (around $0.96$ after fine-tuning). For low–moderate noise ($p_{noise}\le 0.3$), the ground-truth rules for $X_3$ and $X_4$ remain dominant (rule accuracy close to or equal to $1.00$), and $X_5$ still attains strong imputation accuracy (roughly $0.82$–$0.95$) even when the exact structure is not always recovered. When the noise level becomes very high ($p_{noise}\ge 0.4$), rule recovery degrades substantially (rule accuracy near zero for $X_5$ at $p_{noise}=0.5$), but imputation accuracy for all predicates remains above $0.70$, indicating that **the soft-logic relaxation and fine-tuning step enable the model to learn useful approximations rather than overfitting to noisy labels.**

*Table 16. Impact of label noise on rule learning and missing value imputation performance. Results are over 20 random seeds.*

| Noise Ratio | Predicate | Imputation Accu. (Before FT) | Imputation Accu. (After FT) | Train Loss (Before FT) | Train Loss (After FT) |
| :---: | :---: | :---: | :---: | :---: | :---: |
| 0.0 | $X_3$ | $1.000 \pm 0.000$ | / | $0.001 \pm 0.000$ | / |
|     | $X_4$ | $1.000 \pm 0.000$ | / | $0.001 \pm 0.000$ | / |
|     | $X_5$ | $0.907 \pm 0.050$ | $0.955 \pm 0.049$ | $0.089 \pm 0.035$ | $0.067 \pm 0.031$ |
| 0.1 | $X_3$ | $1.000 \pm 0.000$ | / | $0.098 \pm 0.003$ | / |
|     | $X_4$ | $1.000 \pm 0.000$ | / | $0.099 \pm 0.004$ | / |
|     | $X_5$ | $0.948 \pm 0.042$ | $0.946 \pm 0.038$ | $0.168 \pm 0.026$ | $0.123 \pm 0.021$ |
| 0.2 | $X_3$ | $0.975 \pm 0.076$ | / | $0.193 \pm 0.005$ | / |
|     | $X_4$ | $0.987 \pm 0.057$ | / | $0.193 \pm 0.007$ | / |
|     | $X_5$ | $0.894 \pm 0.050$ | $0.902 \pm 0.046$ | $0.260 \pm 0.008$ | $0.204 \pm 0.012$ |
...(Full in paper Table 16)
| 0.5 | $X_3$ | $0.745 \pm 0.085$ | / | $0.418 \pm 0.007$ | / |
|     | $X_4$ | $0.725 \pm 0.077$ | / | $0.421 \pm 0.007$ | / |
|     | $X_5$ | $0.761 \pm 0.057$ | $0.767 \pm 0.072$ | $0.421 \pm 0.007$ | $0.349 \pm 0.007$ |

*Table 17. Learned rule structures under label noise. Ground truth rules are indicated with underlines. Results are over 20 random seeds.*

| Noise Ratio | Predicate | Learned Rule Structure (most frequent rules shown) | Rule Accu. |
| :---: | :---: | :--- | :---: |
| 0.0 | $X_3$ | $\underline{X_0 \wedge X_1}$ | $1.00$ |
|     | $X_4$ | $\underline{X_2 \wedge X_7}$ | $1.00$ |
|     | $X_5$ | $(\underline{X_0 \wedge X_4 \vee (X_3 \wedge X_6)}$, ... | $0.50$ |
| 0.1 | $X_3$ | $\underline{X_0 \wedge X_1}$ | $1.00$ |
|     | $X_4$ | $\underline{X_2 \wedge X_7}$ | $1.00$ |
|     | $X_5$ | $(\underline{X_0 \wedge X_4 \vee (X_3 \wedge X_6)}$, ... | $0.55$ |
| 0.2 | $X_3$ | $\underline{X_0 \wedge X_1}$ | $1.00$ |
|     | $X_4$ | $\underline{X_2 \wedge X_7}$ | $1.00$ |
|     | $X_5$ | $(\underline{X_0 \wedge X_4 \vee (X_3 \wedge X_6)}$, ... | $0.60$ |
...(Full in paper Table 17)
| 0.5 | $X_3$ | $X_1 \wedge X_7$, $X_2 \wedge X_2$, ..., $\underline{X_0 \wedge X_1}$  | $0.10$ |
|     | $X_4$ | $X_6 \wedge X_6$, $X_0 \wedge X_6$, ... (no Ground Truth rule in Top 5) | $0.05$ |
|     | $X_5$ | $(X_0 \wedge X_3) \vee (X_1 \wedge X_6)$, ... (no Ground Truth rule in Top 5) | $0.00$ |

---

### Author Response · Authors · 2025-12-03
**Global Response B-3**

## Global Response B: Rigorous Robustness and Sensitivity Analysis

### B-3: Sensitivity Analysis for **Missingness Ratios**
---
In three synthetic datasets, we have varied the missing ratio in {0.7, 0.8, 0.9}, and we also explore three missingness patterns (i.e. MCAR, MAR, and MNAR) in Appendix E. In real-world datasets, we use different missingness ratios. We have provided the results of SPECT in the inital submission (Table 18). In the revised draft, we add more results for Heart Disease and Birds dataset (Table 19 and 20 at *Appendix I.2*).

Specifically for the **Heart Disease** dataset, we varied the missing ratio as shown below. Results show that with larger missing ratios, the imputation quality remains acceptable.

*Table 19: Imputation accuracy under different observation ratios for Heart Disease dataset.*

| Missing Ratio | Overall | trestbps | chol | thalach | oldpeak |
| :---: | :---: | :---: | :---: | :---: | :---: |
| 0.9 | 0.5602 | 0.6535 | 0.5677 | 0.6502 | 0.3696 |
| 0.7 | 0.6444 | 0.7129 | 0.6304 | 0.7393 | 0.4950 |
| 0.5 | 0.7434 | 0.8053 | 0.7558 | 0.7954 | 0.6172 |
| 0.3 | 0.8432 | 0.8647 | 0.8482 | 0.9043 | 0.7558 |
| 0.1 | 0.9439 | 0.9439 | 0.9307 | 0.9769 | 0.9241 |

For the **Birds Dataset**, we fix the observation ratio as 0.1 (i.e. 90% missingness) and show results over different number of training samples. Results show that a few hundred samples are sufficient for the model to converge to the correct logical truth.

*Table 20: Impact of training sample size on the imputation accuracy of latent predicates (abnormal_bird, can_fly) in the Birds domain.*

| # Samples | Acc. Abnormal Bird | Acc. Can Fly |
| :--- | :---: | :---: |
| 100 | $0.896 \pm 0.058$ | $0.845 \pm 0.148$ |
| 500 | $0.976 \pm 0.054$ | $0.928 \pm 0.066$ |
| 1000 | $0.951 \pm 0.067$ | $0.928 \pm 0.066$ |
| 1500 | $0.949 \pm 0.070$ | $0.952 \pm 0.066$ |
| 2000 | $0.951 \pm 0.067$ | $0.928 \pm 0.066$ |

---

### Author Response · Authors · 2025-12-03
**Global Response C**

## Global Response C: Assessment of learned rule quality.
---
In our inital draft, we have shown the **Rule Discovery Accuracy** (i.e., the proportion of runs where the learned rule structure exactly matches the ground-truth rule) of synthetic (Tables 1, 22, 24, 26, 28) and Birds dataset (Table 4), and show the **Imputation Accuracy** of all datasets to incidate the learned rules qualtiy. Further, for the real-world datasets (SPECT and Heart Disease) without known ground-truth rules, we combined **expert and LLM-based assessment** (Tables 30-32), which shows that the learned rules are meaningful and helpful for human doctors.

In the revised manuscript, we have added a new section in **Appendix G** to further analysis rule quality.

1. We quantify **Structural Stability** by using the *Jaccard index* (i.e., the intersection over union of two predicate sets) to measure the similarity between rule structures across random seeds. All the datasets achieve high Jaccard score (Tables 11-13), supporting that learned rules are structurally stable, even with the real-world dataset with noise. Below we show Table 11 as an example. If we **add noise into synthetic datasets,** the new results in Tables 16 and 17 at Appendix I.1 show that our model can still stably recover the correct rules under noise ratio with 0.3, which support **our model the soft-logic relaxation effectively prevents the model from overfitting to noise.**

*Table 11. Jaccard index of learned rule predicates on synthetic data under different observation probabilities. Example (b) of Figure 3 with 50,000 samples over 20 seeds.*

| Obs. Ratio | $X_3$ | $X_4$ | $X_5$ |
| :---: | :---: | :---: | :---: |
| 0.1 | $1.0000 \pm 0.0000$ | $1.0000 \pm 0.0000$ | $0.7572 \pm 0.2526$ |
| 0.2 | $1.0000 \pm 0.0000$ | $1.0000 \pm 0.0000$ | $0.5987 \pm 0.2837$ |
| 0.3 | $1.0000 \pm 0.0000$ | $1.0000 \pm 0.0000$ | $0.6726 \pm 0.2385$ |

2. We analysis **Rule Complexity**, tracking the arity of conjunction ($h$) and the number of conjunctive clauses ($R_k$). New results in Figures 6 and 7 show that in a wide range except for very small values, our model can get optimal performance (e.g., $h \in [3,9]$ and $R_k \in [5,20]$). This aligns with the intuition that complex real-world decision boundaries often require multiple alternative rules.

---

### Author Response · Authors · 2025-12-03
**Global Response D**

## Global Response D: Theoretical Clarifications on Convergence
---
As detailed in our new *Appendix B* (Convergence Analysis), our learning algorithm is formally standard *Gauss-Seidel Block Coordinate Gradient Method* on the **smooth global objective**:

$$\mathcal{L}(\Theta)=\sum_{j=1}^m \mathcal{L}_{U_j}(\Theta), $$

where $\mathcal{L}\_{U\_j}$ is defined in Eq.6 which minimizes the discrepancy between inferred and observed latent predicates. As all forward-chaining operators (due to the sigmoid, softmin, log-sum-exp approximations) are **continuously differentiable**, so each block gradient is Lipschitz-continuous:
$$
\left\|\nabla_{\Theta_j} \mathcal{L}\left(\Theta+h_j\right)-\nabla_{\Theta_j} \mathcal{L}(\Theta)\right\| \leq L_j\left\|h_j\right\| .
$$
By standard results (Tseng, 2001[1]; Bertsekas, 1999[2]; Nesterov, 2004[3]), in the **full-batch setting** with **sufficiently small steps**, a standard descent lemma for block coordinate descent implies that for step size $\eta \leq 1 / L_{j_t}$,
$$
\mathcal{L}\left(\Theta^{t+1}\right) \leq \mathcal{L}\left(\Theta^t\right)-\frac{\eta}{2}\left\|\nabla_{\Theta_{j_t}} \mathcal{L}\left(\Theta^t\right)\right\|^2,
$$
so the loss is monotonically non-increasing, and every limit point $\Theta^{\star}$ is block-stationary (a local optimum w.r.t. rule-wise perturbations):
$$
\nabla_{\Theta_j} \mathcal{L}\left(\Theta^{\star}\right)=0 \quad \forall j .
$$
*We do **not** claim global optimality, but we now provide a principled and explicit **local convergence justification***.

- Regarding the practical **Adam with mini-batches** updates used by us, our optimization becomes a stochastic block coordinate scheme that decreases the loss **in expectation** (Richtárik \& Takáč, 2014[4]; Wright, 2015[5]. Details can be found in Appendix B). Our model learns a separate parameter vector $\Theta_j$ for each rule, and all updates are actually sequential (what we mentioned "asynchronous" in the original text).

- To **reduce the chance of landing in poor local minima**, which is  an intrinsic challenge in rule learning, we combine the above framework with several exploration: (i) *multiple random initializations*, (ii) sequential covering to grow OR-of-AND structures followed by subsequent *joint fine-tuning*, and (iii) conservative *freezing* of only near-deterministic rules (later unfrozen for refinement). These mechanisms allow the optimizer to explore more of the parameter space and avoid extremely suboptimal basins.

- As for **faithfulness/identifiability**, we acknowledge that exact rule recovery is **NP-hard** (set cover problem) even without missingness and distinct rule sets can be observationally equivalent. Under high missingness, identifiability fundamentally requires additional structural assumptions. Therefore, we do **not** claim global uniqueness. Our goal is instead a principled **local convergence** method that, in practice, **reliably recovers ground-truth rules** on synthetic benchmarks (e.g., Tables 1, 22, 24, and 26) and yields **stable, interpretable rules** on real datasets (e.g., Tables 4, 5, 30-32). **High Jaccard Index** in Tables 11-13 further support the model consistently recovers truth (if known) and stable rules.


**References**:

[1] Tseng, P. *Convergence of a Block Coordinate Descent Method for Nondifferentiable Minimization*. Journal of Optimization Theory and Applications, 109(3):475–494, 2001.

[2] Bertsekas, D. P. *Nonlinear Programming*. 2nd Edition. Athena Scientific, 1999.

[3] Nesterov, Y. *Introductory Lectures on Convex Optimization: A Basic Course*. Applied Optimization, Vol. 87. Springer, 2004.

[4] Richtárik, P., and Takáč, M. *Iteration Complexity of Randomized Block-Coordinate Descent Methods for Minimizing a Composite Function*. Mathematical Programming, 144(1–2):1–38, 2014.

[5] Wright, S. J. *Coordinate Descent Algorithms*. Mathematical Programming, 151(1):3–34, 2015.

---

### Author Response · Authors · 2025-12-03
**Summary of our Response and Revision**

# Summary of our Response and Revision
---

Dear Area Chair,

We sincerely thank all the reviewers for recognizing the **novelty**, **conceptual elegance**, **empirical strength**, **partical value**, and **presentation** of our work. We appreciate that Reviewer **KLE1** commends the paper for proposing a *"closed feedback loop between imputation and rule discovery"*; Reviewer **GyS2** acknowledges *"integration of soft predicate learning enables native handling of continuous features"*; Reviewer **vtkH** highlights *"an elegant combination of imputation and rule induction is conceptually strong and rarely explored"*; and Reviewer **PgSy** praises the fusion of symbolic and neural learning as *"both conceptually elegant and practically meaningful"*.

Regarding evaluation and presentation, Reviewer **KLE1** notes that *"the empirical studies are good and cover a wide setting"*; Reviewer **GyS2** finds the paper *"well written and detailed"* with *"strong results demonstrating both interpretability and accuracy"*; Reviewer **vtkH** adds that the problem is *"well motivated"* and the *"related-work coverage are comprehensive"*; and Reviewer **PgSy** agrees that *"ability to generate human-readable rules adds practical value for analysts"*.

Guided by the reviewers' constructive advice, we have revised the manuscript to add a **convergence analysis and proof**(Appendix B), **new baselines** (Tables 6-7, Appendix D.3), a **rule quality assessment** (Appendix G), **temperature hyperparameter analysis** (Appendix H), and a **sensitivity analysis regarding noise and missingness** (Appendix I). Additionally, we have expanded the **related work** (lines 87-97, 727-733, 739-750) and updated the **experimental findings** (lines 476-480, 495-501).

We highlight the major improvements below:

### **1. More Baselines** (reviewers KLE1, GyS2, and vtkH)

Except for interpretable models we included, we have expanded our evaluation to statistical and deep generative baselines. Please refer to ***Global Response A*** for details. Briefly:

- **Statistical Models**: **MICE** (Chained Equations) and **MissForest** (Random Forest).

- **Deep Generative Models**: **MLP**, **GAIN** (GAN-based), **MissDiff** (Diffusion-based), **mDAE** (DAE-based), and **VAE-based** methods.

Our updated results show that NS-FCN achieves imputation accuracy comparable to complex deep learning baselines while outperforming them in downstream diagnosis, all while retaining interpretability. This stems from our **joint optimization of imputation and rule discovery and soft-logic relaxation**, which **prevents overfitting to noise** and captures **dominant logical structures**.

### **2. Rigorous Robustness and Sensitivity Analysis** (reviewers KLE1, vtkH, and PgSy)

We have added extensive analyses to validate system stability (see ***Global Response B***):

- Noise and Missing Ratio: We introduced label noise and varying missingness ratios. Results demonstrate that the model learns **valid rule approximations** (e.g., capturing one correct clause) and maintains strong predictive performance even under difficult conditions.

- Hyperparameter Stability: Sensitivity analyses on Soft-AND ($\tau$) and Soft-OR ($\beta$) temperatures support our initial claim that the model is robust and **does not require careful tuning**.

### **3. Systematic Assessment of Rule Quality** (reviewers KLE1 and PgSy)

Besides the semantic validation of clinical experts and LLM which were included in the initial draft, we have addressed concerns regarding rule stability and quality (see ***Global Response C***):


- Structural Stability: We introduce the **Jaccard index** to quantify the consistency of learned rules across random seeds, yielding high stability scores.

- Rule Length Analysis: Consistent performance across varying conjunction arity ($h$) and clause counts ($R_k$) confirms the model sustainably learns meaningful rules.


### **4. Theoretical Clarifications** (reviewers KLE1, vtkH and PgSy)

We have clarified the theoretical underpinnings of our approach, specifically:

- **Convergence**: We clarified that our optimization follows Block Coordinate Descent, guaranteeing principled local convergence (details in ***Global Response D***).

- **Differentiability**: We detailed how gradients propagate effectively through cosine similarity scores despite the use of discrete indexing for rule components.

We believe our comprehensive response addresses all reviewer concerns and further strengthens the paper's contributions. Our sincere gratitude to all reviewers and Area Chairs for their time again.

---

### Meta-Review · Area_Chair_Cwzk · 2026-01-06

**Summary:**

This submission presents a neuro-symbolic framework, integrating missing-value imputation with rule discovery.
The reviewers appreciated the elegant formulation, connecting imputation and rule discovery with a feedback loop that enables the model to improve. They brought up the benefit of generating rules that remain consistent with domain knowledge.

The reviewer brought up a variety of points of diverse importance, such as limited baselines or the lack of clarity on the drivers of success of the method (some that seem addressed).

The authors replied with very thorough reply but that overly length and hard to read, with elements of the discussion referring to 8 (!) global comments (to circumvent the size limitation of comments). This was probably counterproductive on the reviewers, which did not reply. This is counterproductive on the AC which feels that there is a lot of information spread out and does not get clear messages.

**Reviewer Concerns:**

Various review concerns, some more minors than others, were addressed: minor concerns on local optima, monotonicity of convergence, more major on increasing baselines of clarifying the drivers of success of the method (the ablations in the rebuttal suggest that forward chaining and back-propagation are important).

The reviewers did not reply, and the crowded aspect of the rebuttal may be partly a reason.
I believe that some of the concerns may be addressed, though the reviewers did not conclude this, and thus is raises the difficult question of what should be assessed by whom.

**Reviewer Scores:**

Had the reviewer been able to participate in the discussion and engaged, they might have increased a bit their score.

---

### Decision · Program_Chairs · 2026-01-26

Accept (Poster)